# Patching open-vocabulary models by interpolating weights

**Gabriel Ilharco**[*1]    **Mitchell Wortsman**[*1]    **Samir Yitzhak Gadre**[*2]    **Shuran Song**[2]
**Hannaneh Hajishirzi**[1,3]    **Simon Kornblith**[4]    **Ali Farhadi**[1]    **Ludwig Schmidt**[1,3]
[1]University of Washington  [2]Columbia University  [3]AI2  [4]Google Research, Brain Team

## Abstract

Open-vocabulary models like CLIP achieve high accuracy across many image classification tasks. However, there are still settings where their zero-shot performance is far from optimal. We study *model patching*, where the goal is to improve accuracy on specific tasks without degrading accuracy on tasks where performance is already adequate. Towards this goal, we introduce PAINT, a patching method that uses interpolations between the weights of a model before fine-tuning and the weights after fine-tuning on a task to be patched. On nine tasks where zero-shot CLIP performs poorly, PAINT increases accuracy by 15 to 60 percentage points while preserving accuracy on ImageNet within one percentage point of the zero-shot model. PAINT also allows a single model to be patched on multiple tasks and improves with model scale. Furthermore, we identify cases of *broad transfer*, where patching on one task increases accuracy on other tasks even when the tasks have disjoint classes. Finally, we investigate applications beyond common benchmarks such as counting or reducing the impact of typographic attacks on CLIP. Our findings demonstrate that it is possible to expand the set of tasks on which open-vocabulary models achieve high accuracy without re-training them from scratch.

## 1  Introduction

Open-vocabulary models are characterized by their ability to perform any image classification task based on text descriptions of the classes [56]. Thanks to advances in large-scale pre-training, recent examples of open-vocabulary models such as CLIP and BASIC have reached parity with or surpassed important task-specific baselines, even when the open-vocabulary models are not fine-tuned on task-specific data (i.e., in a zero-shot setting) [57, 31, 56, 88, 1, 86]. For instance, the largest CLIP model from Radford et al. [57] used in a zero-shot setting matches the ImageNet accuracy of a ResNet-50 trained on 1.2 million ImageNet images [14, 24].

Nevertheless, current open-vocabulary models still face challenges. The same CLIP model that matches a ResNet-50 on ImageNet has lower MNIST accuracy than simple logistic regression in pixel space [57]. Moreover, even when zero-shot models achieve good performance, they are usually still worse than models trained or fine-tuned on specific downstream tasks.

To address these issues, several authors have proposed methods for adapting zero-shot models to a task of interest using labeled data [82, 91, 21, 89, 37, 73]. A common practice is to fine-tune the zero-shot model on the task of interest [82, 56]. However, fine-tuned models can suffer from catastrophic forgetting [48, 76, 20, 33], performing poorly on tasks where the zero-shot model initially performed well [2, 82, 56]. Additionally, fine-tuning typically produces a task-specific classification head, sacrificing the flexible text-based API that makes open-vocabulary models so appealing. Whereas an open-vocabulary model can perform any classification task in a zero-shot fashion, a fine-tuned

---

[*]Equal contribution. Code available at https://github.com/mlfoundations/patching. Correspondance to {gamaga,mitchnw,schmidt}@cs.washington.edu, sy@cs.columbia.edu.

36th Conference on Neural Information Processing Systems (NeurIPS 2022).

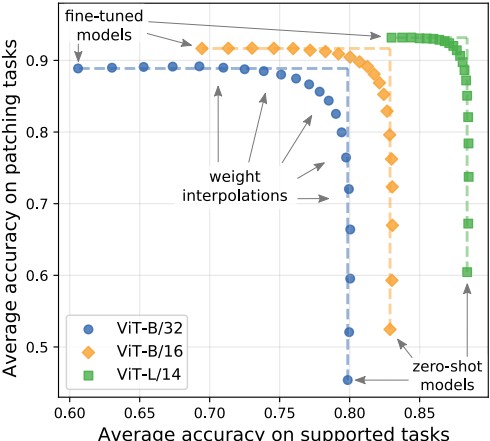

Figure 1: **Patching open-vocabulary models by linearly interpolating weights.** We wish to improve accuracy on tasks where a model performs poorly (*patching tasks*), without degrading performance on tasks where accuracy is already adequate (*supported tasks*). When interpolating weights of fine-tuned models and zero-shot (unpatched) models, there are intermediate solutions where accuracy improves on the patching task without reducing accuracy on supported tasks. Results are shown for CLIP models [57], averaged over nine patching tasks (Stanford Cars, DTD, EuroSAT, GTSRB, KITTI distance, MNIST, RESISC45, SUN397 and SVHN [35, 11, 25, 71, 22, 39, 7, 12, 84, 53]) and five supported tasks (ImageNet, CIFAR-10, CIFAR-100, STL-10 and Food101 [14, 36, 12, 5]). We apply PAINT separately on each patching task and average results across experiments. The dashed lines illustrate vertical movement from the unpatched models and horizontal movement from the fine-tuned models.

model with a task-specific head can only process the specific task that it was fine-tuned on. This specialization can prevent knowledge obtained by fine-tuning on one task from transferring to other related tasks with different classes.

Another approach to adapting zero-shot models would be to add data from the downstream task to the pre-training dataset and train a new open-vocabulary model from scratch. The resulting model could still perform any classification task, and zero-shot performance may improve on related tasks. However, training large image-text models from scratch can require hundreds of thousands of GPU hours [57, 56, 86], which makes this approach practically infeasible in most settings.

In this paper, we study *patching* open-vocabulary models, where the goal is to increase accuracy on new target tasks while maintaining the flexibility of the model and its accuracy on other tasks.[1] Patching aims to combine the benefits of fine-tuning and re-training from scratch: improved performance on the task of interest, maintaining the flexibility of an open vocabulary, transfer between tasks, and fast adaptation time. Motivated by these goals, we extend existing fine-tuning techniques [82] to open-vocabulary settings, where the class space is not fixed. We introduce Patching with Interpolation (PAINT), a simple, two-step procedure for patching models: first, fine-tune the model on the patching task without introducing any task-specific parameters; then, linearly interpolate between the weights of the model before and after fine-tuning. Linearly interpolating neural network weights [52, 19, 54] has been previously used to improve accuracy on a single task [28, 81] or robustness to distribution shift [82]. Indeed, averaging network weights has been explored in continual learning contexts, although for closed-vocabulary models [40].

With PAINT, accuracy can improve on new tasks without degrading accuracy on unrelated tasks, as illustrated in Figure 1. For instance, applying PAINT to a CLIP ViT-L/14 [57] independently on nine image classification tasks [35, 11, 25, 71, 22, 39, 7, 84, 53] improves accuracy by 15 to 60 percentage points compared to the unpatched model, while accuracy on ImageNet [14] decreases by less than one percentage point. We also observe a promising trend: patching becomes more effective with model scale (Section 4.1).

Beyond single tasks, we show that models can be patched on multiple tasks (Section 5). When patching on nine image classification tasks simultaneously, a single CLIP ViT-L/14 model is competitive with using one specialized model for each task—the average accuracy difference is less than 0.5 percentage points.

Moreover, PAINT enables *broad transfer* (Section 6): accuracy on related tasks can increase, even when the class space changes. For instance, we partition EuroSAT [25], a satellite image dataset, into two halves with disjoint labels. Patching a ViT-L/14 model on the first half improves accuracy on the second half by 7.3 percentage points, even though the classes are unseen during patching.

---

[1]The term *patching* is borrowed from software development terminology, drawing inspiration from recent work which conceptualizes developing machine learning models like open-source software [58, 62, 47, 72].

Finally, we investigate PAINT on case studies including typographic attacks [23], counting [32], and visual question answering [4] (Section 7). For instance, applying PAINT using synthetic typographic attacks leads to a model that is less susceptible to typographic attacks in the real world, improving its accuracy by 41 percentage points.

In summary:

- Even the best pre-trained models are not perfect. We introduce PAINT, a method designed to improve accuracy on new tasks without harming accuracy elsewhere.
- PAINT incurs no extra computational cost compared to standard fine-tuning, neither during fine-tuning itself nor at inference time.
- PAINT can also be applied with multiple tasks, providing a single model that is competitive with many specialized models.
- Applying PAINT with one task can improve accuracy on a related task, even when they do not share the same classes.
- PAINT improves with model scale, indicating a promising trend for future models.

## 2   Patching with interpolation (PAINT)

This section details our method for patching models on a single and multiple tasks.

**Patching on a single task.** Given an open-vocabulary model with weights $\theta_{zs}$ and a patching task $\mathcal{D}_{patch}$, our goal is to produce a new model $\theta_{patch}$ which achieves high accuracy on $\mathcal{D}_{patch}$ without decreasing model performance on tasks where accuracy is already acceptable. We let $\mathcal{D}_{supp}$ denote a representative supported task where model performance is adequate, and later show that the method is stable under different choices of $\mathcal{D}_{supp}$ (Section 4.2). The two-step procedure we explore for producing $\theta_{patch}$ is given below.

> **Step 1.** Fine-tune $\theta_{zs}$ on training data from $\mathcal{D}_{patch}$ to produce a model with weights $\theta_{ft}$.
> **Step 2.** For mixing coefficient $\alpha \in [0, 1]$, linearly interpolate between $\theta_{zs}$ and $\theta_{ft}$ to produce $\theta_{patch} = (1 - \alpha) \cdot \theta_{zs} + \alpha \cdot \theta_{ft}$. The mixing coefficient is determined via held-out validation sets for $\mathcal{D}_{supp}$ and $\mathcal{D}_{patch}$. We refer to the resulting model as $\theta_{patch}$.

In our experiments, we do not introduce any additional task-specific parameters when fine-tuning, as discussed in Section 3 and Appendices B and C.

**Patching on a multiple tasks.** In practice, we often want to improve model accuracy on multiple patching tasks $\mathcal{D}_{patch}^{(1)}, ..., \mathcal{D}_{patch}^{(k)}$, which can be accomplished with straightforward modifications to the procedure above. We explore three alternatives and examine their relative trade-offs in Section 5:

- *Joint patching*, where we merge all the patching tasks $\mathcal{D}_{patch}^{(i)}$ into a single task $\mathcal{D}_{patch}$ before running the patching procedure;
- *Sequential patching*, where we iteratively repeat the patching procedure above on each new task $\mathcal{D}_{patch}^{(i)}$ and let $\theta_{zs} \leftarrow \theta_{patch}$ after each completed iteration;
- *Parallel patching*, where we apply the first step on each task in parallel to produce fine-tuned models with weights $\theta_{ft}^{(1)}, ..., \theta_{ft}^{(k)}$. Then, we search for mixing coefficients $\alpha_i$ to produce $\theta_{patch} = (1 - \sum_{i=1}^{k} \alpha_i) \cdot \theta_{zs} + \sum_{i=1}^{k} \alpha_i \cdot \theta_{ft}^{(i)}$.

For joint and parallel patching we assume access to held-out validation sets for all tasks, while in sequential patching we only assume access to held-out validation sets from the tasks seen so far. Unless mentioned otherwise, we pick the mixing coefficient $\alpha$ that optimizes average accuracy on the held-out validation sets from the supported and patching tasks.

## 3   Experimental setup

**Tasks.** We consider a diverse set of image classification tasks from Radford et al. [57]. In most experiments, we use ImageNet [14] as a representative supported task, although we explore other supported tasks in Section 4.2. We categorize tasks into patching tasks or supported tasks based on the accuracy difference between the zero-shot model and a model specialized to the task. A large

accuracy difference indicates that the task is a relevant target for patching because the zero-shot model is still far from optimal. Specifically, we consider a subset tasks from Radford et al. [57], categorizing tasks where the linear probes outperform the zero-shot model by over 10 percentage points as patching tasks: Cars [35], DTD [11], EuroSAT [25], GTSRB [71], KITTI [22], MNIST [39], RESISC45 [7], SUN397 [84], and SVHN [53]. We use the remaining tasks as supported tasks: CIFAR10 [36], CIFAR100 [36], Food101 [5], ImageNet [14], and STL10 [12]. We investigate additional patching tasks as case studies in Section 7 and provide further details in Appendix A.

**Models.** We primarily use CLIP [57] pre-trained vision transformer (ViT) models [15]. Unless otherwise mentioned our experiments are with the ViT-L/14 model, while Section 4.2 studies ResNets [24].

**Fine-tuning on patching tasks.** Unless otherwise mentioned, we fine-tune with a batch size of 128 for 2000 iterations using learning rate 1e-5 with 200 warm-up steps with a cosine annealing learning rate schedule and the AdamW optimizer [43, 55] (weight decay 0.1). When fine-tuning, we use the frozen final classification layer output by CLIP's text tower so that we do not introduce additional learnable parameters. This design decision keeps the model open-vocabulary and does not harm accuracy, as discussed in in Appendices B and C.

**Evaluation.** We use accuracy as the evaluation metric unless otherwise stated. We refer to the average of the mean accuracy on the patching tasks and the mean accuracy on the supported tasks as *combined accuracy*.[2]

## 4 Patching models on a single new task

As shown in Figure 1, when patching a model on a single task, we interpolate the weights of the zero-shot and fine-tuned model, producing a model that achieves high accuracy on both the patching task and the supported task. On the nine tasks, PAINT improves the accuracy of ViT-L/14 by 15 to 60 percentage points, while accuracy on ImageNet decreases by less than one percentage point. PAINT also allows practitioners to control the accuracy trade-off on the patching and supported tasks without re-training a new model, by varying the mixing coefficient $\alpha$.

### 4.1 The effect of scale

We consistently observe that PAINT is more effective for larger models. Our findings are aligned with those of Ramasesh et al. [59], who observed that larger models are less susceptible to catastrophic forgetting. This section formalizes and provides insights for these observations.

**Measuring the effectiveness of patching.** We measure the effectiveness of patching via the accuracy difference between the single patched model and two specialized models with the same architecture and initialization. For both the supported task and patching task, we take specialized models that maximize performance on the task, considering the set of all interpolations between the zero-shot and fine-tuned models. We refer to this measure as *accuracy distance to optimal*. Formally, accuracy distance to optimal is given by

$$\frac{1}{2}\left[\max_{\alpha}\mathsf{Acc}(\theta_{\alpha},\mathcal{D}_{\mathrm{supp}}) + \max_{\alpha}\mathsf{Acc}(\theta_{\alpha},\mathcal{D}_{\mathrm{patch}})\right] - \frac{1}{2}\max_{\alpha}\left[\mathsf{Acc}(\theta_{\alpha},\mathcal{D}_{\mathrm{supp}}) + \mathsf{Acc}(\theta_{\alpha},\mathcal{D}_{\mathrm{patch}})\right], \quad (1)$$

where $\mathsf{Acc}(\theta,\mathcal{D})$ represents the accuracy of model $\theta$ on task $\mathcal{D}$. In Figure 2 (left), we show that accuracy distance to optimal decreases with scale, indicating that patching becomes more effective for larger models.

**Model similarity.** Fine-tuning modifies overparameterized models less [9], which provides insights on why larger models are easier to patch: less movement is required to fit new data. We demonstrate this by evaluating representational similarity using Centered Kernel Alignment (CKA) [34] (see Appendix D for details). As shown in Figure 2 (center), the representations of the unpatched and fine-tuned models become more similar as models grow larger, indicated by larger CKA values. Moreover, Figure 2 (right) shows that the cosine similarity between the weights of the unpatched and fine-tuned models, $\cos(\theta_{\mathrm{zs}},\theta_{\mathrm{ft}}) = \langle\theta_{\mathrm{zs}},\theta_{\mathrm{ft}}\rangle/(||\theta_{\mathrm{zs}}||\,||\theta_{\mathrm{ft}}||)$, increases with scale.

---

[2]In other words, $(\mathbb{E}_{\mathcal{D}_{\mathrm{supp}}}[\mathsf{Acc}(\theta,\mathcal{D}_{\mathrm{supp}})] + \mathbb{E}_{\mathcal{D}_{\mathrm{patch}}}[\mathsf{Acc}(\theta,\mathcal{D}_{\mathrm{patch}})])/2$, where $\mathsf{Acc}(\theta,\mathcal{D}_{\mathrm{supp}})$ and $\mathsf{Acc}(\theta,\mathcal{D}_{\mathrm{patch}})$ are accuracies on supported tasks and patching tasks, respectively.

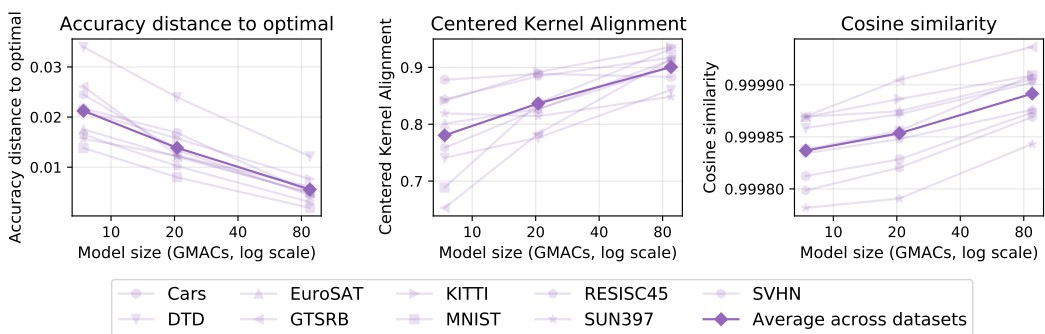

Figure 2: **Larger models are easier to patch** (left). For larger models, the unpatched and fine-tuned model are more similar with respect to their representations (center) and weights (right). Model scale is measured in Giga Multiply-Accumulate operations (GMACs).

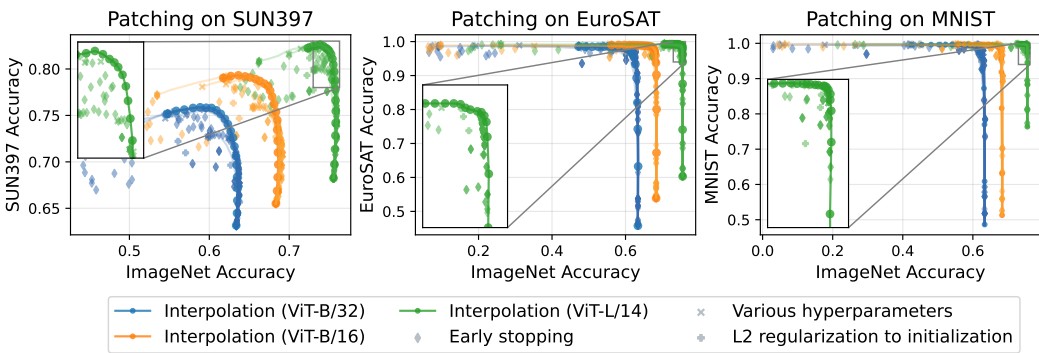

Figure 3: **The frontier of accuracy trade-offs can be recovered by linearly interpolating weights.** Interpolating the unpatched and fine-tuned models recovers the accuracy trade-off of early stopping, regularization, and changes in hyperparameters. Additional details and comparisons can be found in Appendix E.

## 4.2 Baselines and ablations

**Baselines.** There are many alternatives which enable a trade-off between accuracy on the supported and patching tasks. These methods include early stopping during fine-tuning, applying a regularization term which penalizes movement from initialization, or training with different hyperparameters including a smaller learning rate. Unlike interpolation, these methods do not enable navigating the accuracy trade-off without fine-tuning the model again many times. Moreover, Figure 3 demonstrates that the accuracy trade-off frontier for early stopping, regularization, or varying hyperparameters can be recovered by interpolating weights with different mixing coefficients. Appendix E provides additional baselines and discussion, including EMA [74], EWC [33], LwF [41], re-training a model with data from the patching task, and mixing the pre-training and fine-tuning objectives.

**Additional supported tasks.** In Figure 1, we use ImageNet as a representative supported task. This section demonstrates that PAINT is stable under different choices of the supported task. Instead of ImageNet, we use CIFAR10, CIFAR100, Food101 and STL10. Figure 4 displays representative results, where performance is averaged over the nine patching tasks (see Appendix F for additional results). We observe consistent results across supported tasks, and that the optimal mixing coefficients are stable across different choices of supported tasks (Figure 4, right).

**Additional models.** In addition to the CLIP ViTs used in the majority of our experiments, we study four ResNet models [24] from Radford et al. [57] in Appendix G. We find that patching is less effective for ResNets compared to ViTs of similar size, which corroborates the findings of Ramasesh et al. [59] that ResNets are generally more susceptible to catastrophic forgetting. However, similarly to ViTs, we still observe improvements with scale. Finally, we show that patching is also effective for closed-vocabulary models in Appendix H.

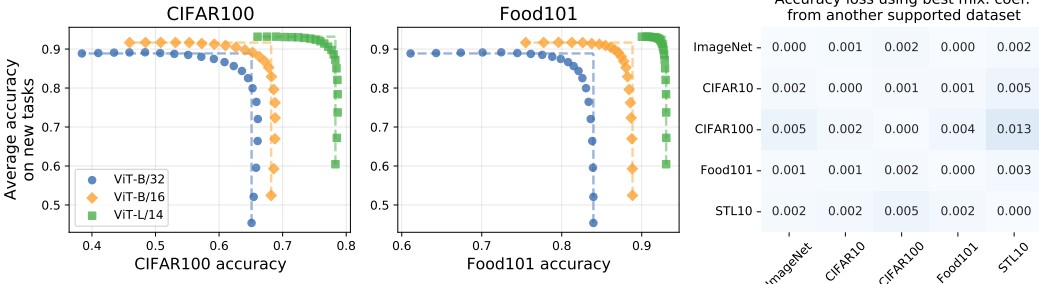

Figure 4: **Results are consistent across supported tasks.** For multiple supported tasks, we observe similar accuracy improvements on patching tasks, without substantially decreasing supported task accuracy. Additional results for the supported tasks Food101, STL10 and ImageNet are in Appendix F. Moreover, choosing the mixing coefficients using a different supported task does not substantially decrease combined accuracy on patching and supported tasks (right).

## 5 Patching models on multiple tasks

This section details experimental results for patching on multiple datasets. Recall from Section 2 that there are various strategies for extending PAINT to multiple datasets, which we briefly revisit. For *joint* patching we merge all the datasets into a single fine-tuning task and apply our patching procedure as before. For *sequential* patching we iteratively perform our procedure once per task, using the patched model at each step as the initialization for the next step.[3] We also explore *parallel* patching, for which we have an unpatched model $\theta_{zs}$ and independently fine-tune on each of the tasks in parallel. We then search for mixing coefficients to combine the resulting models. For tasks $1, ..., k$, let $\theta_{ft}^{(1)}, ..., \theta_{ft}^{(k)}$ denote the fine-tuned models for each task. Since it is impractical to exhaustively search over each $\alpha_i$, we instead search over a one-dimensional scalar $\alpha \in [0, 1]$, which interpolates between $\theta_{zs}$ and the average of all fine-tuned solutions $\frac{1}{k} \sum_{i=1}^{k} \theta_{ft}^{(i)}$.[4] Appendix J provides further experimental details.

These methods have various trade-offs and may be applicable for different scenarios. Joint patching is only possible when data from all tasks you wish to patch is available. On the other hand, sequential patching is appropriate when the tasks are observed one after another. Finally, parallel patching can leverage distributed hardware.

Figure 5 displays experimental results when patching on all nine tasks from Section 4. We observe that joint patching is the best-performing method on average. This is perhaps unsurprising since joint patching has simultaneous access to all patching datasets, unlike other patching strategies. Nevertheless, it is still interesting that for ViT-L/14, joint patching yields a *single* model with only 0.5 percentage points worse combined accuracy than using *multiple* specialized models.[5] Joint patching also achieves a 15.8 percentage points improvement over the unpatched model. Moreover, patching a ViT-B/32 model with the joint strategy achieves a combined accuracy 6.1 percentage points higher than a ViT-L/14 unpatched model, which requires 12x more GMACs.

The accuracy of sequential patching approaches that of joint patching, especially for larger models. Note that, unlike in joint patching, forgetting can compound since the patching procedure is applied multiple times in sequence. In sequential patching, weight interpolations do not completely eradicate forgetting, but greatly mitigate it. This is most noticeable for smaller models: sequentially fine-tuning a ViT-B/32 without interpolation *reduces* the combined accuracy by 4.6 percentage points compared to the unpatched model, as shown in Appendix J. This is compared to a combined accuracy *increase* of 11 percentage points when using sequential patching. Additional results, including experiments on SplitCIFAR [61], can be found in Appendix J.

Finally, parallel patching underperforms other patching strategies. Like sequential patching, parallel patching is in the challenging setting where data from all patching tasks is not available simultaneously.

---

[3]The results are averaged over three random seeds that control the order in which tasks are seen.

[4]We also explored adaptive black-box optimization algorithms to choose the mixing coefficients $\alpha_i$ [60], but observed little improvement (0.3 to 0.4 percentage points on average).

[5]Recall from Section 3 that combined accuracy weight patching and supported tasks equally.

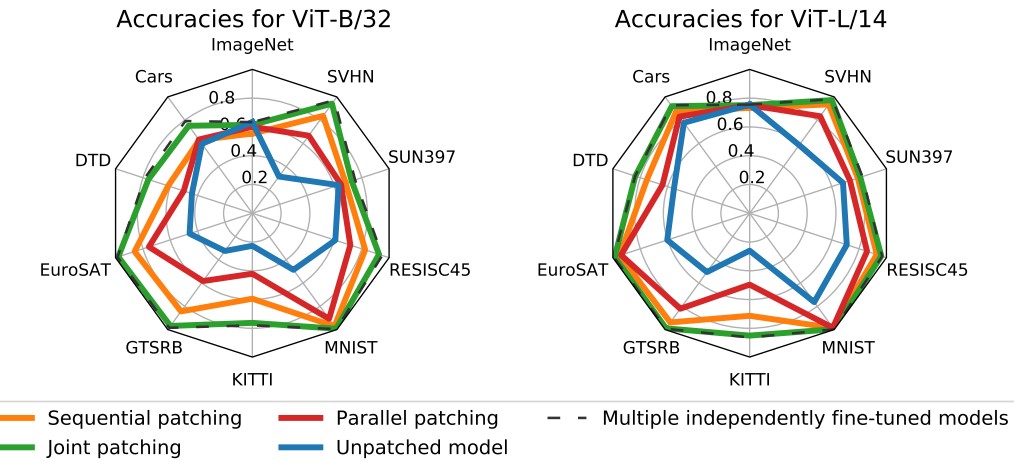

Figure 5: **Contrasting various strategies for patching on multiple tasks.** On all experiments, ImageNet is used as the supported task while the other nine datasets are used for patching. When data from all patching tasks is available, joint patching yields a single model that is competitive with using ten different specialized models. Weight interpolations greatly mitigate catastrophic forgetting on the sequential case, but do not completely eradicate it. Finally, parallel patching underperforms other patching strategies, but still provides improvements over the unpatched model.

| | Cars | DTD | EuroSAT | GTSRB | KITTI | MNIST | RESISC45 | SUN397 | SVHN |
|---|---|---|---|---|---|---|---|---|---|
| Unpatched accuracy | 86.2 | 64.9 | 79.9 | 51.7 | 43.4 | 82.6 | 73.4 | 76.9 | 72.8 |
| Patched accuracy | 87.0 (+0.8) | 66.1 (+1.2) | 87.2 (+7.3) | 71.1 (+19.4) | 60.4 (+17.0) | 91.3 (+8.7) | 74.2 (+0.8) | 79.3 (+2.4) | 88.9 (+16.1) |

Table 1: **PAINT can generalize to unseen classes.** We randomly partition each dataset into tasks $A$ and $B$ with disjoint class spaces of roughly equal size. This table reports how patching on task $A$ affects accuracy on task $B$ for the ViT-L/14 model. In all cases, accuracy on task $B$ improves when patching on task $A$ even though the classes are *unseen* during patching.

Moreover, unlike in joint or sequential patching, no model is optimized on data from all patching tasks. Using a black box optimization algorithm for finding the mixing coefficients did not yield large improvements over using the same mixing coefficient for all models. However, it is possible that more sophisticated search methods could yield better results. In Appendix J, we present additional experiments for a subset of the tasks where exhaustively searching the space of mixing coefficients is tractable, finding headroom for improvement in most cases.

## 6 Broad transfer

An alternative to our patching approach is to introduce parameters which are specific to each new task. By contrast, PAINT always maintains a single model. This section describes an additional advantage of the single model approach: patching the model on task $A$ can improve accuracy on task $B$, even when task $A$ and $B$ do not share the same classes. We refer to this phenomenon as *broad transfer*. Note that we are able to study this phenomenon because the single patched model remains open-vocabulary throughout the patching procedure. This is a key advantage of PAINT compared to maintaining a collection of task-specific models.

We now describe two experiments to measure the effects on a task $B$ when patching the model on a task $A$. First, we explore broad transfer by randomly partitioning datasets into disjoint sets with no class overlap. For a dataset $\mathcal{D}$ we partition the class space $\mathcal{Y}$ into two disjoint sets of roughly equal size $\mathcal{Y}_A$ and $\mathcal{Y}_B$. We build task $A$ with the examples $(x, y) \in \mathcal{D}$ where $y$ belongs to $\mathcal{Y}_A$, and task $B$ with examples $(x, y)$ where $y$ belongs to $\mathcal{Y}_B$. Table 1 shows how patching a model on task $A$ affects the accuracy on task $B$ for nine datasets $\mathcal{D}$. The accuracy improvements on task $B$ range from 0.8 to 19.4 percentage points, even though the classes from task $B$ are not seen during patching.

To further understand transfer, we consider additional task pairs $A$ and $B$, which are now different datasets. While some pairs $A$, $B$ share classes, there are still instances of broad transfer. Concretely,

| Task $A$ | MNIST | SVHN | EuroSAT | RESISC45 | MNIST | FashionMNIST | GTSRB | MTSD |
| Task $B$ | SVHN | MNIST | RESISC45 | EuroSAT | FashionMNIST | MNIST | MTSD | GTSRB |
|---|---|---|---|---|---|---|---|---|
| Unpatched accuracy | 58.6 | 76.4 | 71.0 | 60.2 | 67.7 | 76.4 | 19.3 | 50.6 |
| Patched accuracy | 68.9 (+10.3) | 93.2 (+16.8) | 69.7 (-1.3) | 70.4 (+10.2) | 70.8 (+3.1) | 77.5 (+1.1) | 30.8 (+11.5) | 69.8 (+19.2) |

Table 2: **Patching on task $A$ can improve accuracy on a related task $B$.** For a pair of tasks $A$ and $B$, we report accuracy of the ViT-L/14 on task $B$, after patching on task $A$, finding improvements on seven out of eight cases.

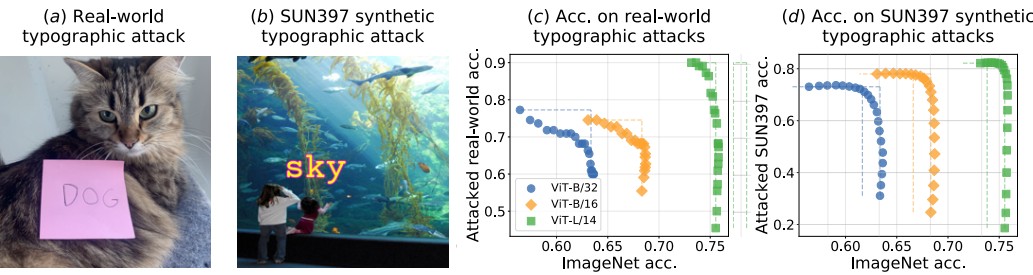

Figure 6: **Guarding against real-world typographic attacks by patching on synthetic data.** *(a)* A sample from our real-world typographic attacks test set. A CLIP ViT-L/14 is "tricked" into classifying this image as a dog instead of a cat. *(b)* Sample of synthetic typographic attack data. *(c)* Performance on real-world data with unseen classes after patching on *only* synthetic typographic attacks (curves produced by interpolating between the unpatched and fine-tuned model). *(d)* Analogous curves for the test set of the synthetic data used for patching.

Table 2 examines i) MNIST and SVHN, two digit recognition tasks with shared classes; ii) EuroSAT and RESISC45, two satellite imagery recognition tasks where there are unshared classes but some overlap; iii) GTSRB and MTSD [17], two traffic sign recognition datasets where there are unshared classes but some overlap; and iv) MNIST and FashionMNIST [83], which do not share any classes but appear visually similar. In seven out of eight experiments, patching on task $A$ improves accuracy by 1.1 to 19.2 percentage points on task $B$. The exception is when $A$ is EuroSAT and $B$ is RESISC45, where accuracy decreases by 1.3 percentage points.

In all experiments, when patching on task $A$ we choose the mixing coefficient $\alpha$ by optimizing the held-out validation accuracy on task $A$ and a supported task (in this experiment we use ImageNet). While it is possible for a method that introduces new parameters for each task to exhibit broad transfer to new data, this also requires knowing which parameters to apply for the new data. This is not necessary in the single model approach.

# 7 Case studies

We further examine the performance of PAINT in three additional settings, which highlight weaknesses of the zero-shot CLIP model and showcase broad transfer (Section 6).

**Typographic attacks.** Goh et al. [23] find that CLIP models are susceptible to *typographic attacks*, where text superimposed on an image leads to misclassification. For example, in Figure 6 *(a)*, the text on the pink note saying "dog" leads a CLIP to misclassify the image of a cat as a dog. To fix this vulnerability, we procedurally generate typographic attack data by adding text with incorrect class names to SUN397 [84], as seen in Figure 6 *(b)*. We then collect a test set of 110 real world images by placing notes on objects and taking photos.[6] After applying PAINT using the synthetic data, we evaluate on the real-world images (Figure 6 *(c)*) and synthetic test set (Figure 6 *(d)*). We observe that while larger models are more susceptible to typographic attacks, they are also more amenable to patching. Furthermore, we see an example of broad transfer between the synthetic and real-world data: when patching ViT-L/14 on synthetic data, its accuracy on real-world typographic attacks improves 41 percentage points even though the real-world classes are unseen. The cost is

---

[6]Data available at https://github.com/mlfoundations/patching.

a reduction of less than 1 percentage point on ImageNet. We present details on the task and data collection in Appendix K.

**Counting.** Radford et al. [57] find that CLIP models struggle to count the number of objects in CLEVR [32]. Here, the task is to choose an integer between 3 and 10 for each image, corresponding to the number of visible objects. While a straightforward way to patch such a task is to fine-tune on it directly, we investigate if applying PAINT using a subset of the classes allows the patched model to generalize to other numbers. Specifically, we patch on images with 4, 5, 6, 8, or 9 objects. To evaluate broad transfer, we test on images with 3, 7, and 10 objects (7 for understanding interpolation and 3 and 10 for extrapolation). We find that PAINT improves accuracy from 59% to over 99% on unseen classes with less than half a percentage point decrease in ImageNet accuracy. For more details see Appendix L.

**Visual question answering.** As shown by Shen et al. [68], zero-shot CLIP models perform poorly on visual question answering [4]. Using CLIP for VQA typically involves additional parameters—for instance, Shen et al. [68] trains a transformer [77] on CLIP features. In contrast, our procedure for patching CLIP on VQA does not introduce new parameters. Following Shen et al. [68], we contrast images with a series of text prompts, where each prompt corresponds to an option in multiple-choice VQA, formed by both the question and a candidate answer using the following template: "Question: [question text] Answer: [answer text]". We evaluate on multiple-choice VQA v1 [4], where each question is associated with 18 candidate answers. Our results, further detailed in Appendix M, show that patching is effective for visual question answering: PAINT improves the accuracy of a ViT-L/14 model by 18 percentage points, while accuracy drops by less than one percentage point on ImageNet.

## 8 Related work

**Continual learning and catastrophic forgetting.** Learning tasks sequentially remains a challenge for neural networks. When a neural network learns a new task, the accuracy on other tasks often decreases, a phenomenon known as *catastrophic forgetting* [48, 76, 20, 33]. While forgetting in neural networks may actually aid learning [90], researchers have proposed various approaches for alleviating catastrophic forgetting, including: i) Regularization-based approaches such as elastic weight consolidation (EWC) [33] and synaptic intelligence (SI) [87] which penalize the movement of parameters and are related to weight-interpolation by Lubana et al. [44]; ii) Replay methods [61, 69, 42, 6, 64, 50], which incorporate data or gradient information from previous tasks when learning a new task; and iii) Introducing task-specific parameters [65, 85, 46, 8, 78, 80].

In contrast to these approaches, PAINT requires no modification to the standard fine-tuning process besides the later weight interpolation step. Moreover, unlike regularization or replay based methods, PAINT requires no extra computational cost during training. In contrast to methods with task specific parameters, we maintain a single model. Having a single model is beneficial when there is new data which is similar to one of the tasks which have already been patched. Even without explicitly knowing which task the new data is similar to, we can observe accuracy improvements (see Section 6).

Similar to our work is that of Mirzadeh et al. [50], who observe high accuracy on task A on the linear path between a model which achieves high accuracy on task A and a model which is fine-tuned jointly on task A and B. Moreover, they observe high accuracy on task B on the linear path between a model fine-tuned on task B, and the jointly fine-tuned model. Therefore, there exists a path between a model which achieves good performance on task A and a model fine-tuned on task B along which accuracy is high on both tasks. However, in Mirzadeh et al. [50] this combined path can be non-linear, leading them to propose a regularization and replay based method. In our work, we find that examining models on a linear path between the unpatched model (which has high accuracy on task A) and the model fine-tuned on task B is often sufficient for obtaining a model which achieves high accuracy on both tasks (Figure 1). We speculate that this is due to scale and model architecture: in contrast to Mirzadeh et al. [50], we initialize with a model pre-trained on a large dataset consisting of 400 million images [57], and primarily use vision transformers [15]. As shown in Section 4.2, our method performs substantially worse with ResNets [24], which are used by Mirzadeh et al. [50].

Finally, Ramasesh et al. [59] and Mehta et al. [49] also observed that catastrophic forgetting is less problematic for large and pre-trained models. In addition, Ramasesh et al. [59] found—similar to our results—that vision transformers are less susceptible to forgetting than ResNets of the same size.

**Linear mode connectivity and robust fine-tuning.** Linearly interpolating neural network weights is a key step in PAINT. Because of the many nonlinear activations in a neural network, it is not clear

a priori that linearly interpolating between two sets of weights can result in a high accuracy solution. However, researchers have observed that interpolating neural network weights can achieve high accuracy when training on MNIST from a common initialization [52] or when part of the optimization trajectory is shared [19, 28, 54, 18, 82, 47, 16, 81, 10]. The term linear mode connectivity was coined by Frankle et al. [19]: two networks exhibit linearly mode connectivity if the accuracy does not decrease when using weights on the linear path between them [52, 19]. Weight averaging for continual learning has also been studied by Lee et al. [40] for closed-vocabulary models.

While Nagarajan and Kolter [52] and Frankle et al. [19] focused on accuracy on a single task, Wortsman et al. [82] use linear mode connectivity to fine-tune models while preserving their robustness to natural distribution shifts. By interpolating the weights of a zero-shot and fine-tuned model, they find a solution which performs well both on the fine-tuning task and under distribution shift. In contrast to Wortsman et al. [82], we do not modify any task-specific parameters when fine-tuning, preserving the open-vocabulary nature of the models we patch. Unlike Wortsman et al. [82], we examine accuracy trade-offs across different tasks with little or no class overlap and adapt a model to multiple tasks.

In addition, closely related to our work is that of Matena and Raffel [47], who use Fisher-weighted averaging of language models before and after fine-tuning on downstream tasks. Unlike Fisher-weighted averaging of Matena and Raffel [47], we do not use different mixing coefficients for each parameter, and thus require no extra compute when patching. Moreover, we explore new strategies for patching on multiple tasks (see Section 5), and focus on open-vocabulary image classifiers.

**Interventions to change the behavior of a trained model.** Several authors have studied the problem of updating a model to locally alter its behavior on certain inputs without external disruptions on other inputs [70, 13, 51, 66, 63, 62]. Previous literature uses various terms to refer to this process, including model editing, patching or debugging. A popular use case is to update trained language models to reflect changes in the world (for instance, facts like who is the current president of Brazil) [29, 45, 38, 30]. Moreover, inspired by software engineering practice, previous work explored "debugging" language models through user interaction [63, 62], including providing corrective feedback to the models via natural language [3]. Mitchell et al. [51], De Cao et al. [13] propose training auxiliary networks to perform local edits on pre-trained models. Santurkar et al. [66] introduce a method for rewriting the prediction rules of a classifier, focusing on specific failure modes such as reliance on spurious correlations. In contrast with previous literature, our work explores patching models at the *task* level, aiming to systemically improve accuracy on a dataset—for instance, enabling a model to recognize dozens of satellite imagery classes with a single patch.

## 9 Limitations and conclusion

**Limitations.** When applying PAINT, accuracy on supported tasks can still decrease, especially for smaller models. This limitation is perhaps best reflected in the case of sequential patching: patched models underperform using multiple specialized models when many tasks are added sequentially. Using larger models and weight interpolations can alleviate this issue, but do not completely resolve it. Finally, better understanding on which datasets patching is more effective is an exciting direction for future research.

**Conclusion.** In this work, we explore several techniques for patching open-vocabulary models with the goal of improving accuracy on new tasks without decreasing accuracy elsewhere. PAINT is effective in several scenarios, ranging from classifying digits to defending against typographic attacks. PAINT becomes more effective with scale, and can be applied on multiple tasks sequentially or simultaneously. Our findings demonstrate that in many circumstances it is possible to expand the set of tasks on which models achieve high accuracy, without introducing new parameters, without re-training them from scratch, and without catastrophic forgetting.

## Acknowledgments

We thank Akari Asai, Alex Fang, David Fleet, Huy Ha, Ari Holtzman, Pieter-Jan Kindermans, Marco Tulio Ribeiro, Ofir Press, Sarah Pratt, Sewon Min, Thao Nguyen and Tim Dettmers for helpful discussions and feedback, and Hyak at UW for computing support. This work is in part supported by the NSF AI Institute for Foundations of Machine Learning (IFML), Open Philanthropy, NSF IIS 1652052, NSF IIS 17303166, NSF IIS 2044660, NSF IIS 2132519, ONR N00014-18-1-2826, DARPA N66001-19-2-4031, DARPA W911NF-15-1-0543, the Sloan Fellowship and gifts from Allen Institute for AI.

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
