| | Size of the set used for | | | |
| Dataset | Training | Validation | Testing | Number of classes |
|---|---|---|---|---|
| Cars [35] | 7,330 | 814 | 8041 | 196 |
| DTD [11] | 3,384 | 376 | 1,880 | 47 |
| EuroSAT [25] | 21,600 | 2,700 | 2,700 | 10 |
| GTSRB [71] | 23,976 | 2,664 | 12,630 | 43 |
| KITTI [22] | 6,347 | 423 | 711 | 4 |
| MNIST [39] | 55,000 | 5,000 | 10,000 | 10 |
| RESISC45 [7] | 17,010 | 1,890 | 6,300 | 45 |
| SUN397 [84] | 17,865 | 1,985 | 19,850 | 397 |
| SVHN [53] | 68,257 | 5,000 | 26,032 | 10 |
| FashionMNIST [83] | 55,000 | 5,000 | 10,000 | 10 |
| MTSD [17] | 55,078 | 5,000 | 8,737 | 235 |
| CIFAR10 [36] | 45,000 | 5,000 | 10,000 | 10 |
| CIFAR100 [36] | 45,000 | 5,000 | 10,000 | 100 |
| Food101 [5] | 70,750 | 5,000 | 25,250 | 101 |
| STL10 [12] | 4,500 | 500 | 8,000 | 10 |
| ImageNet [14] | 1,255,167 | 26,000 | 50,000 | 1,000 |

Table 3: **Dataset statistics** for patching and supported tasks. When a set for validation is not available we use held-out data from the official training set for validation purposes. In cases like ImageNet we use the official validation set for testing.

## A  Dataset details

In Table 3, we present the number of classes and the size of the training, validation and test sets we use for the each patching and supported tasks: Stanford Cars [35], Describable Textures (DTD) [11], EuroSAT [25], German Traffic Sign Recognition Benchmark (GTSRB) [71], KITTI distance [22], MNIST [39], RESISC45 [7], SUN397 [84], SVHN [53] ImageNet [14], FashionMNIST [83], MTSD [17], CIFAR10 [36], CIFAR100 [36], Food101 [5], STL10 [12] and ImageNet [14]. For datasets that did not have publicly available, labeled test sets, we use the validation set as the test set, and split the training set into training and validation sets in a stratified fashion. For all these datasets, we use accuracy as our evaluation metric.

## B  Background on open-vocabulary models

In contrast to typical image classifiers, open-vocabulary models are not constrained to a fixed classification space. Instead, they are able to perform any image classification task, by using textual descriptions of the class names. Recently, many open-vocabulary models have been proposed [57, 31, 56, 88, 86, 1].

A popular class of open-vocabulary models are contrastive image-text models like CLIP, BASIC and ALIGN. Following Radford et al. [57] we use the term CLIP to refer to any contrastive image-text model. This class of models are the focus of this work, although PAINT can naturally be extended to other open vocabulary models like Flamingo [1]. CLIP models are trained to contrast images and textual descriptions, and are usually pre-trained on large, heterogeneous data collected from the web, ranging from hundreds of millions to billions of pairs of images and captions. This class of models consists of a vision encoder $f$ which processes images and a text encoder $g$ which processes text. Given a set of image-caption pairs $\{(x_1, y_1), ..., (x_k, y_k)\}$, the model is optimized to maximize the similarity of aligned pairs $\langle f(x_i), g(y_i) \rangle$ relative to unaligned pairs.

When performing a classification task, a set of captions $\{y_1, ..., y_k\}$ are procedurally generated based on text descriptions of the classes and some template. For instance, for distinguishing between cats and dogs, the set of candidate captions could be {"an image of a dog", "an image of a cat"}. Then, given an image $x$, the chosen class is selected based on the caption that maximizes feature similarity with the image ($\arg\max_i \langle f(x), g(y_i) \rangle$). A common practice is to generate multiple candidate captions for each class (for instance, "an image of a dog" and "a photo of a dog") and average their representations before computing the similarities. This practice, introduced by Radford et al. [57], is known as prompt ensembling.

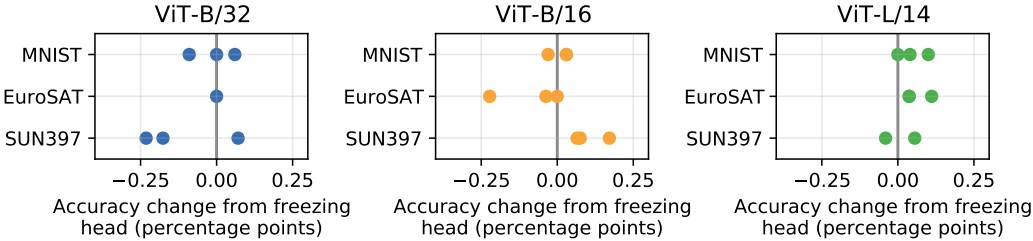

Figure 7: **Freezing the classification layer output by CLIP's text encoder has minimal effect on downstream accuracy.** Each experiment is repeated for three random seeds, shown as different points.

Adapting open-vocabulary models to downstream tasks is typically done by introducing task-specific parameters, mapping visual representations to a fixed class space determined by the downstream task. The model can then be fine-tuned end-to-end, where both the visual encoder and the new classification head are updated, or adapted via training linear probe on top of the visual representations. By contrast, we do not introduce or modify any task-specific parameters. To do so, we adapt only the weights of the visual encoder, maintaining the text encoder without updating its weights. Freezing the text encoder allows for faster fine-tuning, since the text features can be pre-computed once per task. Moreover, in Appendix C, we show that the decision of freezing the text encoder does not substantially harm performance. Importantly, this fine-tuning procedure allows us to adapt the model without loosing its ability to perform any image classification task.

## C Frozen CLIP heads for fine-tuning

Instead of introducing a learnable classification layer, we use the frozen output of CLIP's text encoder to map image features to the space of classes. Figure 7 shows that this modification has a negligible effect on downstream accuracy for MNIST, EuroSAT, and SUN397.

## D The effect of scale on patching

This section introduces additional metrics for the effectiveness of patching and how similar the unpatched and fine-tuned models are. Results are shown in Figure 8.

### D.1 Measuring the effectiveness of patching

In addition to *accuracy distance to optimal* presented in Section 4.1, we present two other metrics to measure the effectiveness of patching, *accuracy distance to endpoints* and *path correction cost*, which are summarized in Table 4. For simplicity, let $x_\alpha$ denote the accuracy of model $\theta_\alpha$ on the supported task and $y_\alpha$ on the patching task. For all metrics, lower values indicate that patching is more effective.

*Accuracy distance to endpoints* contrasts the accuracy of a single model with using two specialized models, the fine-tuned model for the patching task and the unpatched model for the supported task. Recall that $x_\alpha$ denotes the accuracy of model $\theta_\alpha$ on the supported task and $y_\alpha$ on the patching task. Accuracy distance to endpoints is given by $(x_0 + y_1)/2 - \max_\alpha(x_\alpha + y_\alpha)/2$.

*Path correction cost* measures how the curve traced by varying the mixing coefficient compares to a curve without accuracy trade-offs. Its value is the average cost needed to move points in the traced curve to the ideal curve $\mathcal{I}$, where $\mathcal{I} = \{x, y \in [0, 1] \times [0, 1] \mid x = x_{zs} \vee y = y_{ft}\}$ is the set of points on the scatter plots where either accuracy on the supported task is equal to that of the unpatched model or accuracy on the patching task is equal to that of the fine-tuned model. More precisely, patch correction cost is given by, $\mathbb{E}_\alpha[\delta((x_\alpha, y_\alpha), \mathcal{I}) \mathbb{1}[x_\alpha < x_0 \wedge y_\alpha < y_1]]$, where $\delta((x, y), \mathcal{I}) = \min_{x', y' \in \mathcal{I}} ||(x - x', y - y')||_2$ is the distance between $(x, y)$ and $\mathcal{I}$, and $\mathbb{1}[]$ is the indicator function. The indicator term $\mathbb{1}[]$ ensures costs are only computed for points where both accuracy on the supported task is smaller than that of the unpatched model and accuracy on the patching task is smaller than that of the fine-tuned model.

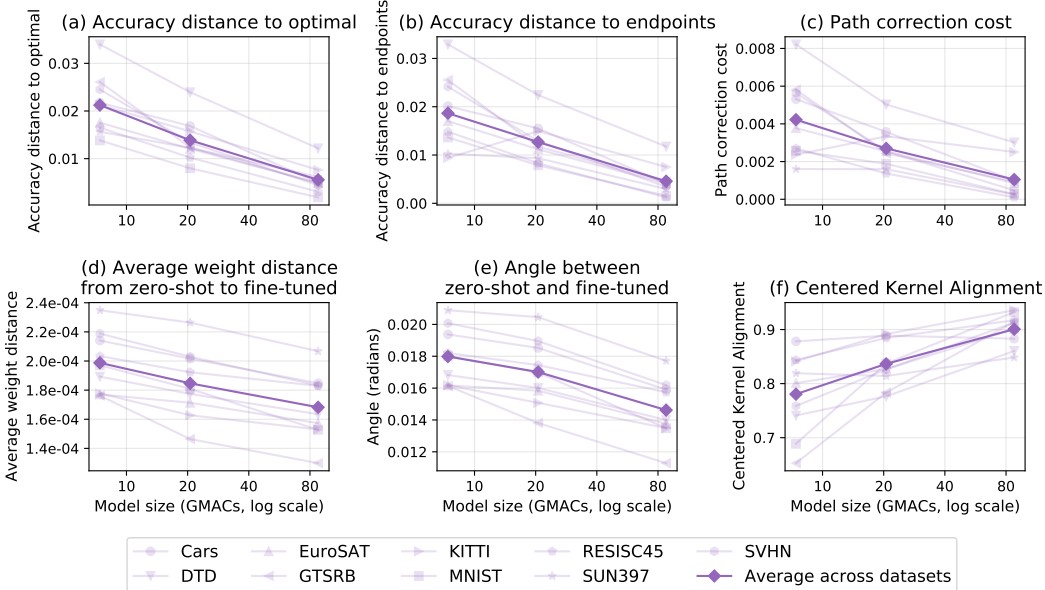

Figure 8: **The effect of scale on model patching**. (a-c) Patching is more effective for larger models. (d, e) Unpatched and fine-tuned models have more similar weights at scale. (f) For larger models, the unpatched and fine-tuned model are more similar with respect to their representations.

| Acc. distance to endpoints | Acc. distance to optimal | Path correction cost |
|---|---|---|
| $\frac{x_0+y_1}{2} - \max_\alpha\left(\frac{x_\alpha+y_\alpha}{2}\right)$ | $\frac{\max_\alpha x_\alpha + \max_\alpha y_\alpha}{2} - \max_\alpha\left(\frac{x_\alpha+y_\alpha}{2}\right)$ | $\mathbb{E}_\alpha[\delta\left((x_\alpha, y_\alpha), \mathcal{I}\right)\mathbb{1}[x_\alpha{<}x_0 \wedge y_\alpha{<}y_1]]$ |

Table 4: **Measures of patching effectiveness.** For all metrics, lower values indicate that patching is more effective. $x_\alpha$ denotes the accuracy of model $\theta_\alpha$ on the supported task and $y_\alpha$ on the patching task.

## D.2 Model similarity

In addition to the angle between the unpatched and fine-tuned models, we also measure the average absolute difference $||\theta_{\text{ft}} - \theta_{\text{zs}}||_1/n$ between the weights $\theta_{\text{zs}}, \theta_{\text{ft}} \in \mathbb{R}^n$.

Representational similarity is measured through CKA [34], which is shown in Equation 2. In short, CKA can be used to compare the similarity between two models by inspecting their representations on a given set of inputs. Formally, let $\Theta_{\text{zs}}, \Theta_{\text{ft}} \in \mathbb{R}^{n \times d}$ denote the $d$-dimensional features of the unpatched and fine-tuned model on a dataset with $n$ samples, pre-processed to center the columns. CKA is then given by:

$$CKA = \frac{||\Theta_{\text{ft}}^\top \Theta_{\text{zs}}||_F^2}{||\Theta_{\text{zs}}^\top \Theta_{\text{zs}}||_F ||\Theta_{\text{ft}}^\top \Theta_{\text{ft}}||_F}, \tag{2}$$

where $||A||_F$ indicates the Frobenious norm of a matrix $A$. Larger values of CKA indicate more similar representations. When reporting CKA values, we compute the last layer features using samples from the supported task.

## E Additional baselines and comparisons

This section details additional baselines for our patching procedure in the context of patching a single task. We discuss exponential moving averages (Section E.1), elastic weight consolidation (Section E.2), learning without forgetting (Section E.3, re-training (Section E.4), and finally mixing the pre-training and fine-tuning objective (Section E.5).

### E.1 Exponential moving averages (EMA)

In Figure 3 we show that interpolating the weights of the unpatched and fine-tuned model can recover the "forgetting frontier" [59]. The forgetting frontier is formed by fine-tuning with various

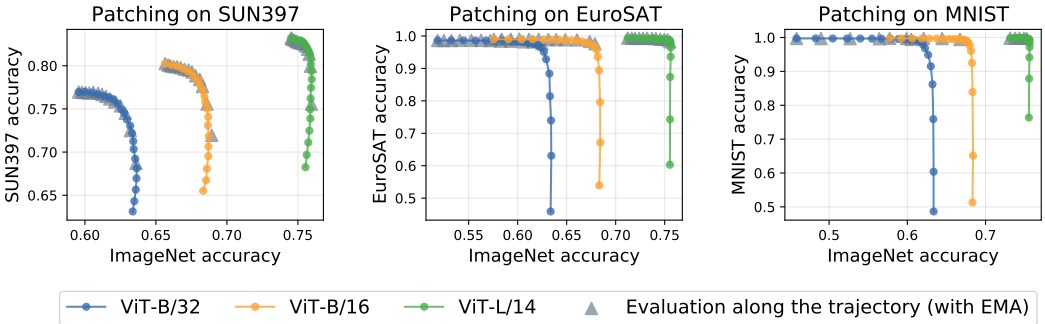

Figure 9: **Comparisons with EMA.** When fine-tuning with a constant learning rate and using EMA [74], interpolating the unpatched and fine-tuned model recovers a similar accuracy trade-off as terminating training early.

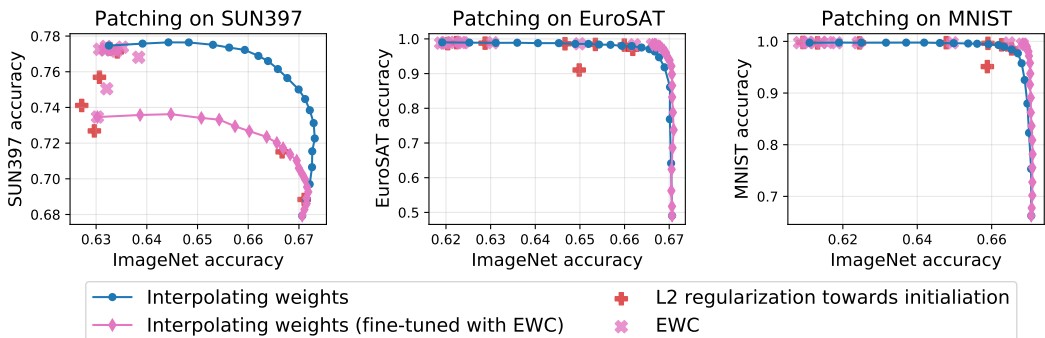

Figure 10: **Comparisons with EWC and regularization towards initialization.** When data is available from the pre-training set it is possible to augment standard fine-tuning with EWC [33]. EWC provides a solution with a good accuracy trade-off when patching on MNIST and EuroSAT, but not SUN397. We also show interpolations between the weights of the unpatched model and a model fine-tuned with EWC. As pre-training data is required to compute the fisher information matrix for EWC, these experiments use a ViT-B/16 model from a CLIP reproduction OpenCLIP [27] pre-trained on LAION 400M [67].

hyperparameters. In particular, we show that interpolating the unpatched and fine-tuned models can recover a solution with similar or better accuracies than early termination. However, these early termination solutions can potentially be suboptimal because learning rate has not yet decayed to zero. As such, we recreate this comparison in Figure 9 where we fine-tune with a constant learning rate and EMA [74].

### E.2 Elastic weight consolidation (EWC)

Elastic weight consolidation [33] (EWC) is a method which penalizes the movement of parameters which are deemed important for solving previous tasks. However, access to the pre-training data is required to investigate which parameters are important. Therefore, we could not use this method for Figure 3 since the pre-training data for the CLIP models of Radford et al. [57] is private. To examine the performance of EWC, we use a reproduction of CLIP from the OpenCLIP repository [27], which is pre-trained on the open source LAION 400M dataset [67]. We use 2,000 iterations of pre-training to compute the fisher information matrix required for EWC. The results are illustrated in Figure 10 which show EWC solutions corresponding to different coefficients from the EWC loss ($\{0.001, 0.01, 0.1, 1, 10, 100, 1000, 10000, 100000, 1000000\}$). We also interpolate the weights of the unpatched model and the EWC solution fine-tuned with coefficient 1000000. On MNIST and EuroSAT, an EWC solution exhibits slightly better accuracy trade-offs than any solution on the interpolation between the unpatched model and model fine-tuned without EWC. However, on SUN397, interpolating the unpatched and fine-tuned model provides a better trade-off than EWC.

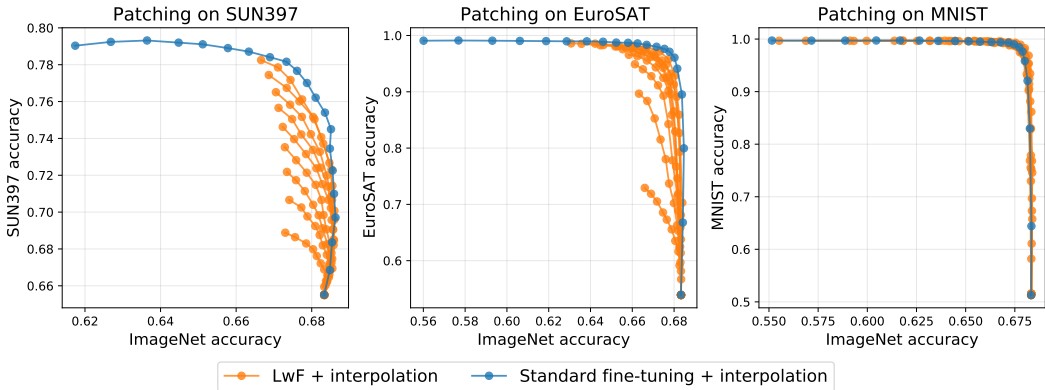

Figure 11: **Comparisons with LwF.** LwF provides a solution with a good accuracy trade-off when patching on MNIST and EuroSAT, but not SUN397. We also show interpolations between the weights of the unpatched model and a model fine-tuned with LwF, showing that PAINT can be complementary with LwF.

### E.3 Learning without forgetting (LwF)

Learning without forgetting [41](LwF) adds an additional regularization term when fine-tuning based on knowledge distillation [26]. We contrast PAINT with LwF in Figure 11. When fine-tuning with LwF, we use multiple loss balance weights $(0.1, 0.2, ..., 0.9)$, and leave the remaining hyper-parameters unchanged. As in Section E.2, we use SUN397, MNIST and EuroSAT as our patching tasks, ImageNet as our supported task, and a patch CLIP ViT-B/16 model. As shown in Figure 11, PAINT is competitive or better than LwF on all tasks. Moreover, weight interpolations can further improve on LwF, showing that PAINT and LwF are complementary, rather than mutually exclusive alternatives.

### E.4 Re-training with data from the patching task

We further contrast patching with re-training a model from scratch, adding data from the patching task to the pre-training dataset. For such, we use a ViT-B/32 model, training for 32 epochs with cosine learning rate schedule with lienar warmup of 5000 steps and learning rate of 0.001, AdamW optimizer with weight decay of 0.1 and global batch size of 1024, using the open-source library OpenCLIP [27]. We train both on data from YFCC-15M [57] alone, and with both YFCC-15M and MNIST data, without upsampling.

Results are shown in Figure 12. We find that re-training with data from the patching task is highly effective at improving accuracy on that task, and only slightly decreases zero-shot accuracy on ImageNet. However, we note that pre-training is substantially more expensive than patching. For instance, considering the ViT-L/14 models trained by Radford et al. [57], pre-training takes around 10,000 times more compute than our patching procedure, which makes re-training impractical in most scenarios.

### E.5 Objective mixing

When data is available from the pre-training dataset it becomes possible mix the pre-training and fine-tuning objectives. This baseline, which we refer to as *objective mixing*, is similar to replay methods in continual learning [61, 69, 42, 6, 64, 50]. However, objective mixing is only possible when the pre-training data is available, which is not the case for the official CLIP models of Radford et al. [57]. Therefore, we use models from a CLIP reproduction, OpenCLIP [27].

First, we use an OpenCLIP ViT-B/32 model which is trained on the publicly available LAION 400M dataset [67] and showcase results in Figure 13 (left). Like CLIP models from Radford et al. [57], this model is trained with a batch size of approximately 32,000. We use a single machine and try batch sizes 128, 256, 512, and 1024 for the pre-training objective. Overall the loss is given by $(1 - \beta) \cdot \ell_{\text{pre-training}} + \beta \cdot \ell_{\text{fine-tuning}}$ for $\beta \in \{0, 0.001, 0.01, 0.1, 0.2, ..., 0.9, 1.0\}$ where $\ell_{\text{pre-training}}$ and $\ell_{\text{fine-tuning}}$ are the pre-training and fine-tuning loss, respectively. We notice and interesting phenomenon: even when $\beta = 0$, the zero-shot accuracy on ImageNet still drops from the pre-trained

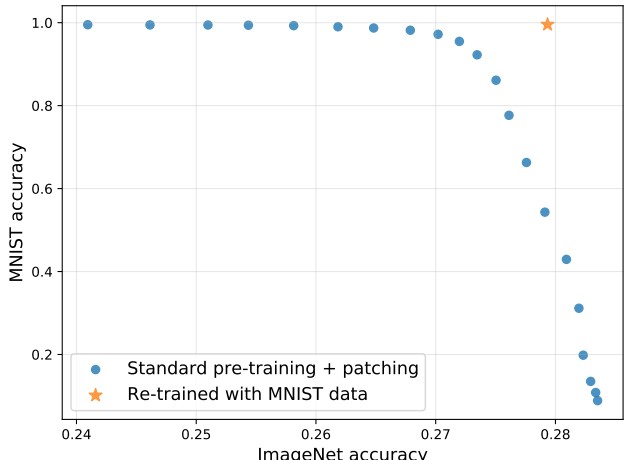

Figure 12: **Comparisons with re-training.** Re-training with data from the patching task is effective at improving accuracy on that task. However, this approach is orders of magnitude more expensive than patching.

model (Figure 13). We hypothesize that the drop in zero-shot accuracy is likely due to a reduction in batch size and note that the importance of large batches for CLIP objectives has been studied by Pham et al. [56], Radford et al. [57]. Because of this drop in accuracy, the patching procedure we study matches or provides better accuracy trade-offs in this setting.

Next, we pre-train our own ViT-B/32 model on a single machine using OpenCLIP [27]. using a batch size of 512 for the pre-training objective. We also use the smaller pre-training dataset YFCC-15M [75, 57]. In Figure 13 (right) we perform objective mixing with coefficients $\beta \in \{0, 0.001, 0.01, 0.1, 0.2, ..., 0.9, 1.0\}$ and batch size 512 for the pre-training objective. In this setting, zero-shot accuracy does not drop for $\beta = 0$ and objective mixing performs better than the patching procedure we propose. We believe this is because scale helps our method (see Section 4.1), and this is the smallest scale model we examine in terms of pre-training set size. In terms of absolute accuracy, applying PAINT to the LAION pre-trained ViT-B/32 model is the best option.

In conclusion, the findings of these experiments are that:

1. Objective mixing is difficult because it requires a large batch size.

2. Objective mixing with a small batch size performs well when applied to CLIP models trained with small batch sizes, but these models are worse overall.

## F  Additional plots for patching on a single task

Breakdowns for each supported task and patching task are shown in Figures 16 to 19.

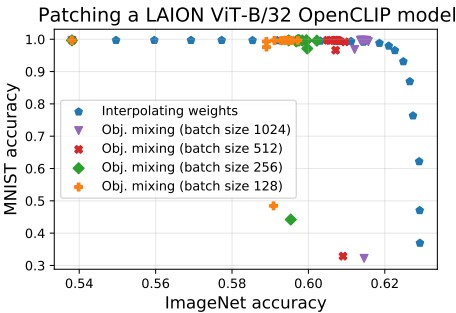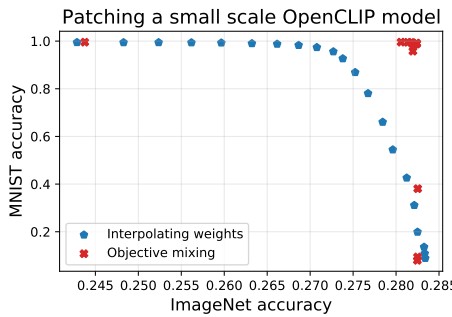

Figure 13: **Comparisons with objective mixing.** Objective mixing mixes together the pre-training objective and fine-tuning objective. Since the pre-training objective requires access to pre-training data, we experiment with a ViT-B/32 model which is pre-trained on LAION 400M [67] from the CLIP reproduction OpenCLIP [27] (left) and a ViT-B/32 model pre-trained by us with a small batch (512) on YFCC-15m [75, 57]. The left plot shows that objective mixing is difficult because it requires a large batch size. Pre-training on LAION 400M uses batch size 32k, and zero-shot accuracy drops when we continue pre-training with a smaller batch size. In this setting, the patching procedure we propose (denoted by "interpolating weights") offers the better accuracy trade-off. The right plot shows that, while objective mixing can be successfully applied to models pre-trained with small batch sizes, these models perform worse overall. Details are in Section E.5.

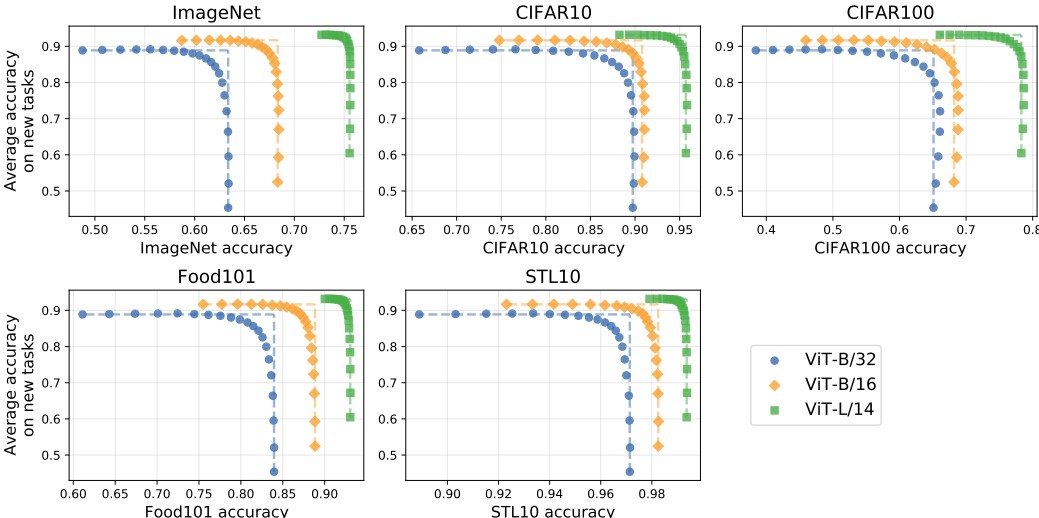

Figure 14: **Average patching results for various supported tasks** For multiple supported tasks, we observe similar accuracy improvements on patching tasks, without substantially decreasing supported task accuracy.

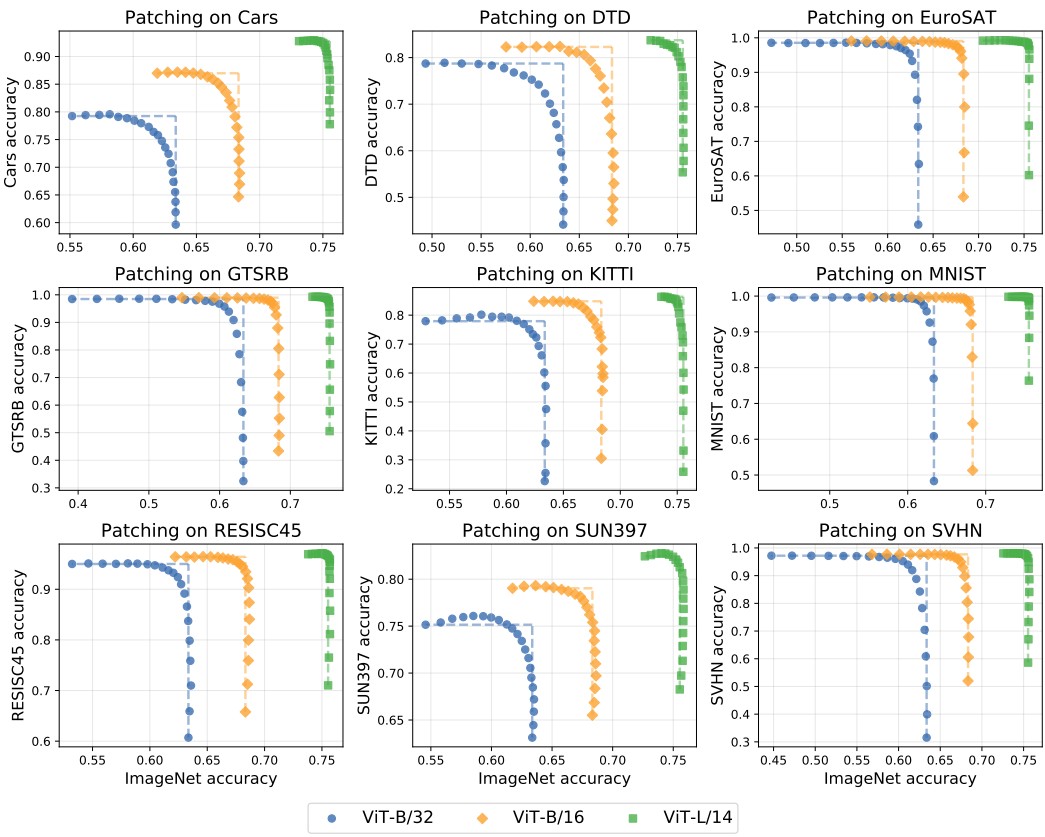

Figure 15: Patching results for ImageNet as the supported task. Results are shown for nine patching tasks.

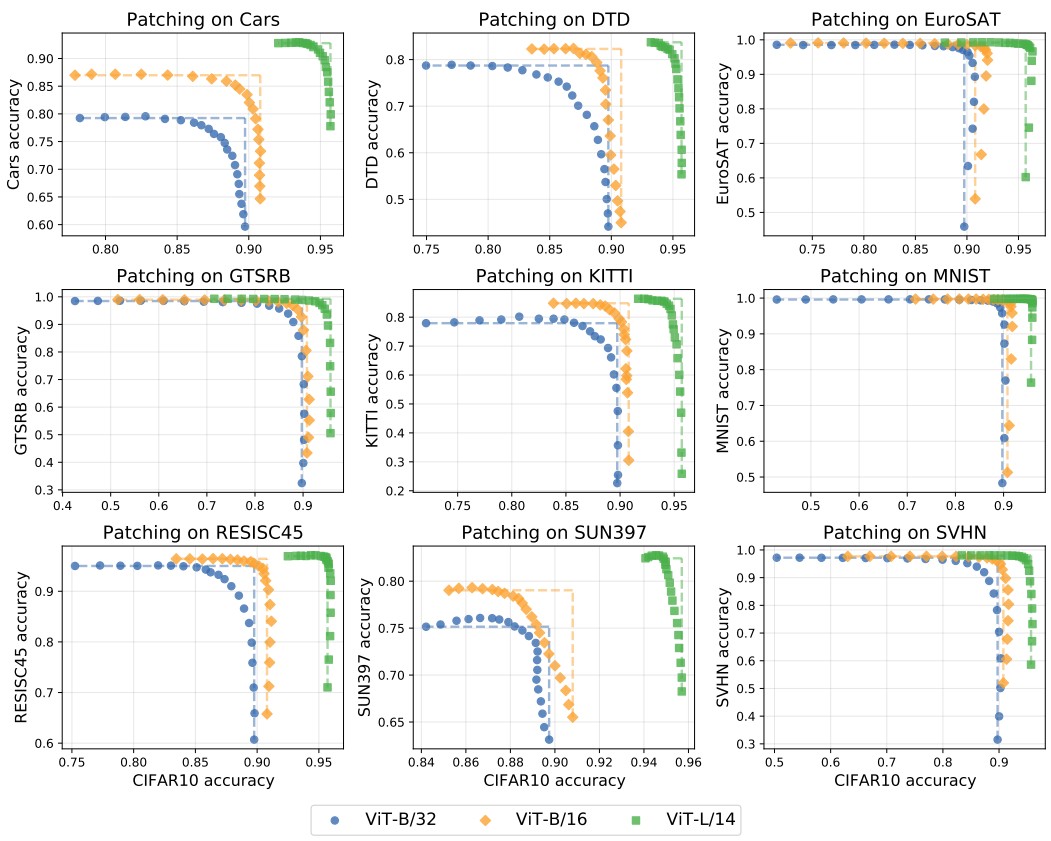

Figure 16: **Patching results for CIFAR10 as the supported task.** Results are shown for nine patching tasks.

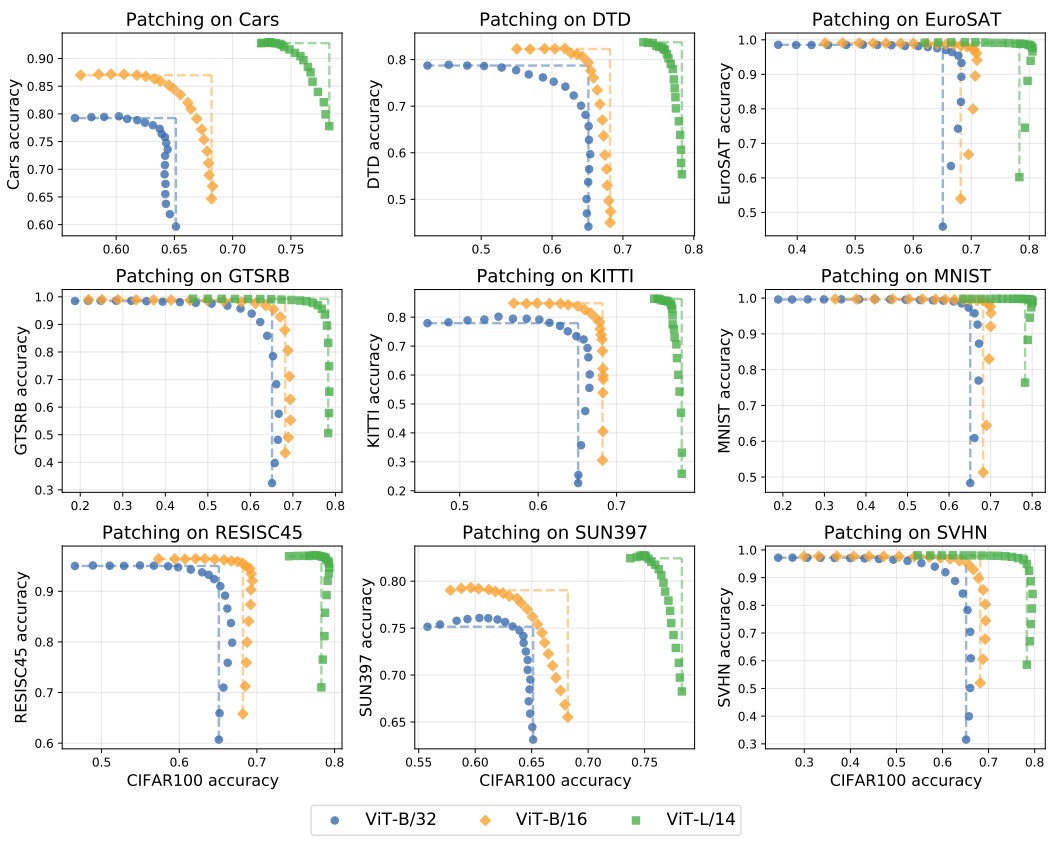

Figure 17: **Patching results for CIFAR100 as the supported task.** Results are shown for nine patching tasks.

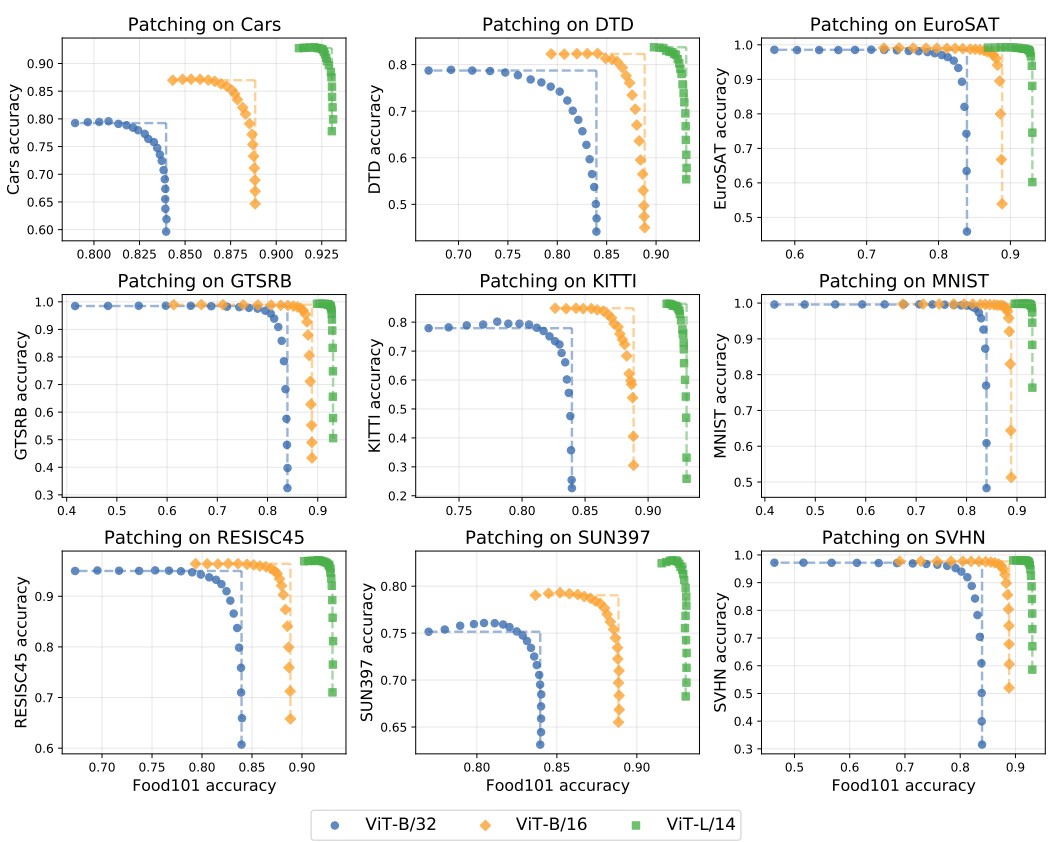

Figure 18: **Patching results for Food101 as the supported task.** Results are shown for nine patching tasks.

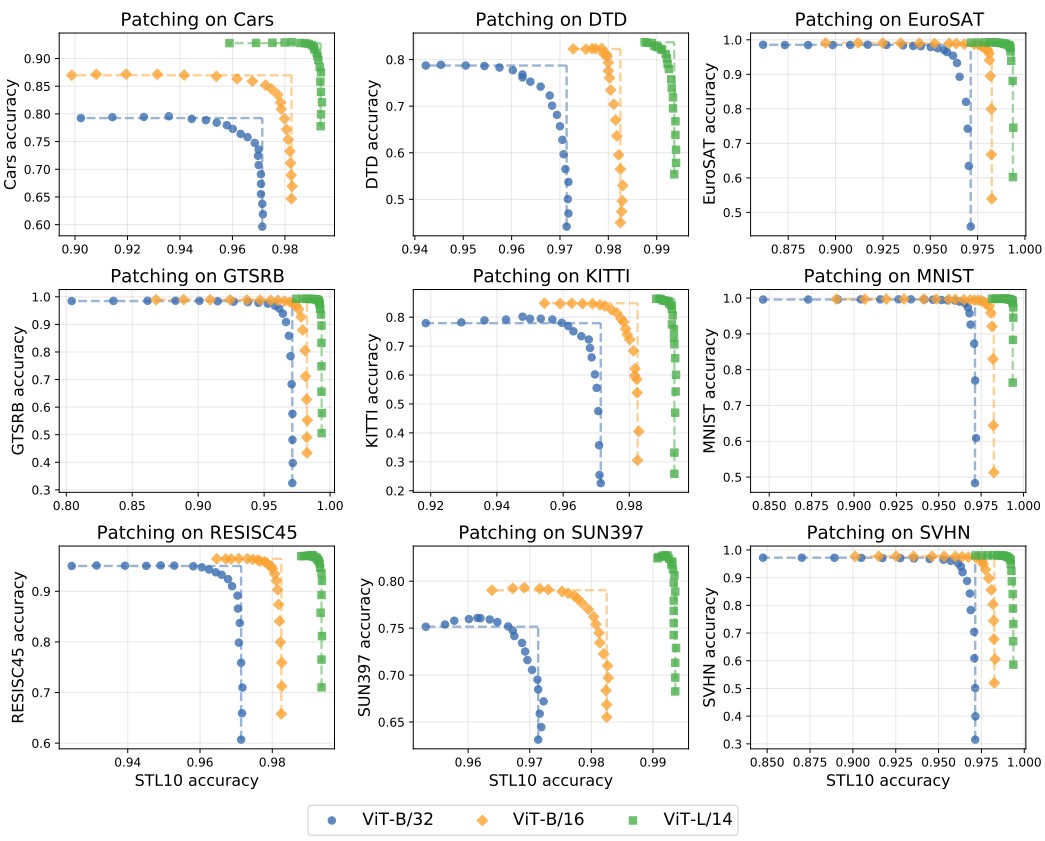

Figure 19: **Patching results for STL10 as the supported task.** Results are shown for nine patching tasks.

# G   Additional models

In addition to ViT models, we measure the effectiveness of patching for four ResNet models [24]. Specifically, we examine ResNet-50, ResNet-101, and two wider networks, ResNet-50x4, ResNet-50x16 [24, 57]. Results are shown in Figure in 20.

# H   Patching closed-vocabulary models

Beyond open-vocabulary models, we show that PAINT is also effective for closed-vocabulary image classifiers. Our experimental setting is as follows: we start with a (closed-vocabulary) model trained on ImageNet from scratch, from the Pytorch ImageNet Models library [79]. Our goal is to expand the set of categories known by the model to improve its performance on MNIST, without hurting accuracy on ImageNet. In other words, we wish to build a model that is competent at classifying an image both amongst the 1000 categories from ImageNet, and amongst 10 digit categories from MNIST. For such, we expand the classification head from the original model by adding 10 new classes, and initialize the corresponding weights and biases to zero. We then fine-tune the model on MNIST without any frozen weights, and interpolate with the model before fine-tuning.

Figure 21 shows results for patching closed-vocabulary ViT-B/32, ViT-B/16 and ViT-L/16 models that are trained from scratch on ImageNet. For all models, accuracy on MNIST improves to over 99%, while accuracy on ImageNet decreases by less than one percentage point. These experiments show that patching is effective beyond open-vocabulary models.

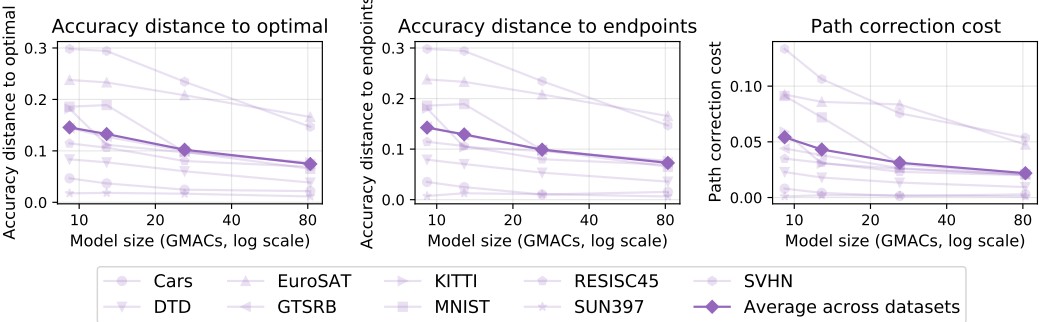

Figure 20: **The effect of model scale in patching ResNets.** Compared to Vision Transformers (ViTs), patching is less effective for ResNets, corroborating findings of Ramasesh et al. [59]. Similarly to ViTs, patching is more effective for larger models.

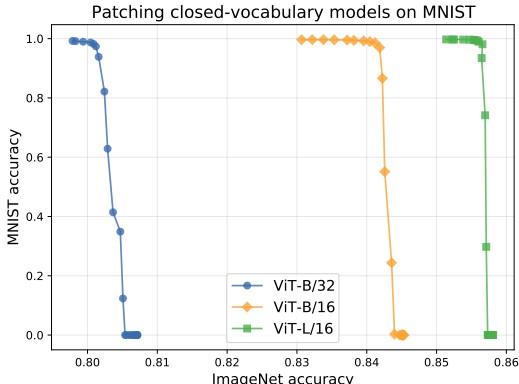

Figure 21: **PAINT is also effective for closed-vocabulary models** For ViT models trained from scratch on ImageNet, PAINT improves accuracy on MNIST to over 99%, while accuracy on ImageNet decreases by less than one percentage point.

# I   Broad Transfer

This section provides additional results for the broad transfer experiments in Section 6, as well as new experiments where we fine-tune on ImageNet.

In Section 6, Tables 1 and 2 only show results for the single mixing coefficient that out procedure chooses. To supplement these tables, we show how accuracy changes for various mixing coefficients $\alpha \in \{0, 0.05, 0.1, ..., 1\}$. For Table 1 the corresponding figure with all mixing coefficient information is Figure 22. Similarly, Table 2 is expanded in Figures 23 and 24.

Finally, we measure broad transfer on 13 datasets when patching on ImageNet. In Figure 25 we fine-tune on ImageNet then interpolate with the unpatched model. Surprisingly, for the ViT-B/16 model, fine-tuning on ImageNet improves accuracy on KITTI accuracy by more than 10 percentage points, and MNIST accuracy by more than 20 percentage points, even without patching. More investigation into broad transfer is required to understand when and how it applies.

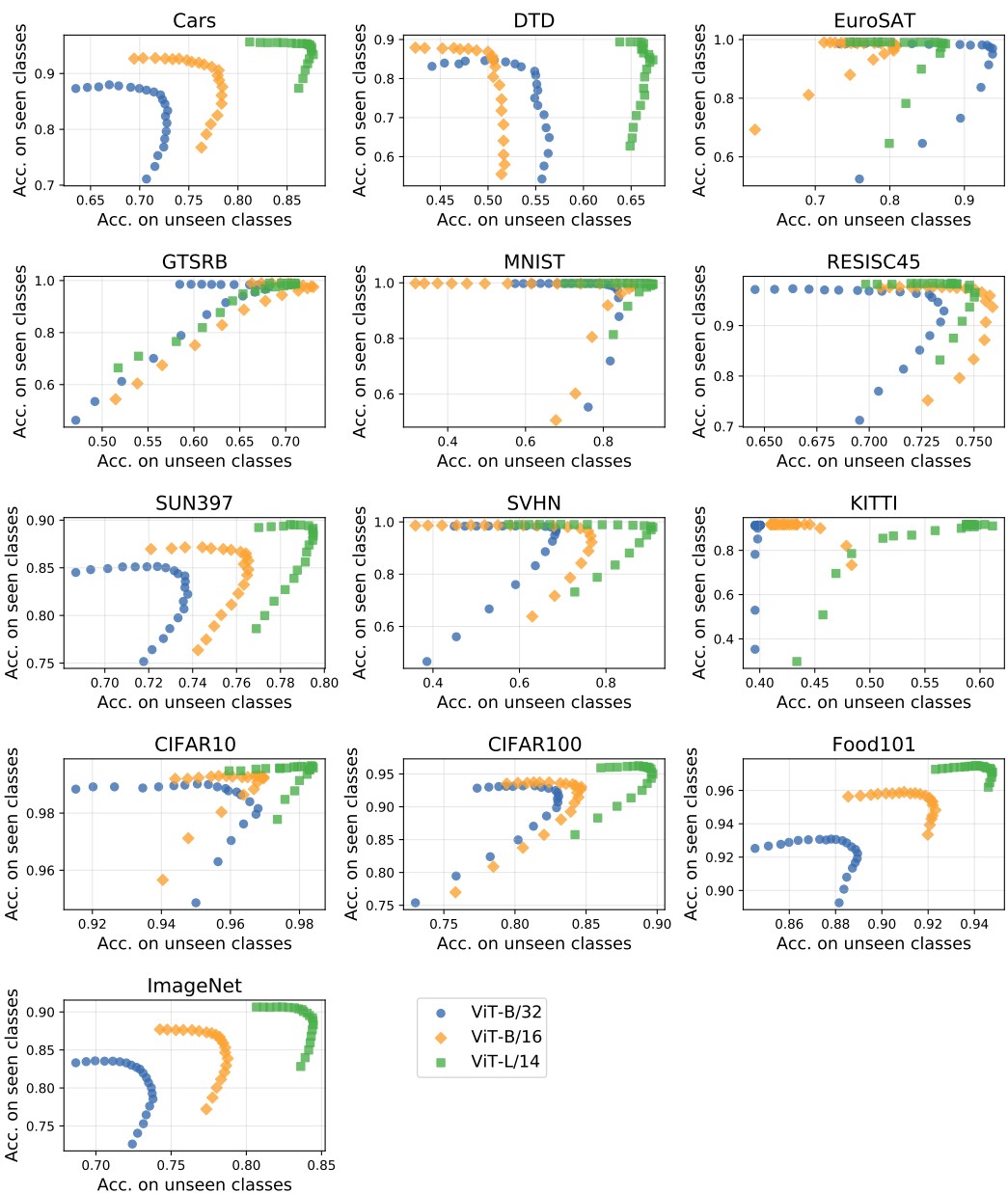

Figure 22: **The effect of patching on unseen classes.** We randomly partition each dataset into tasks $A$ and $B$ with disjoint class spaces of roughly equal size. In this Figure we fine-tune on task $A$ then interpolate the fine-tuned model with the unpatched model. We measure accuracy on task $A$ ($y$-axis) and task $B$ ($x$-axis). This Figure supplements Table 1.

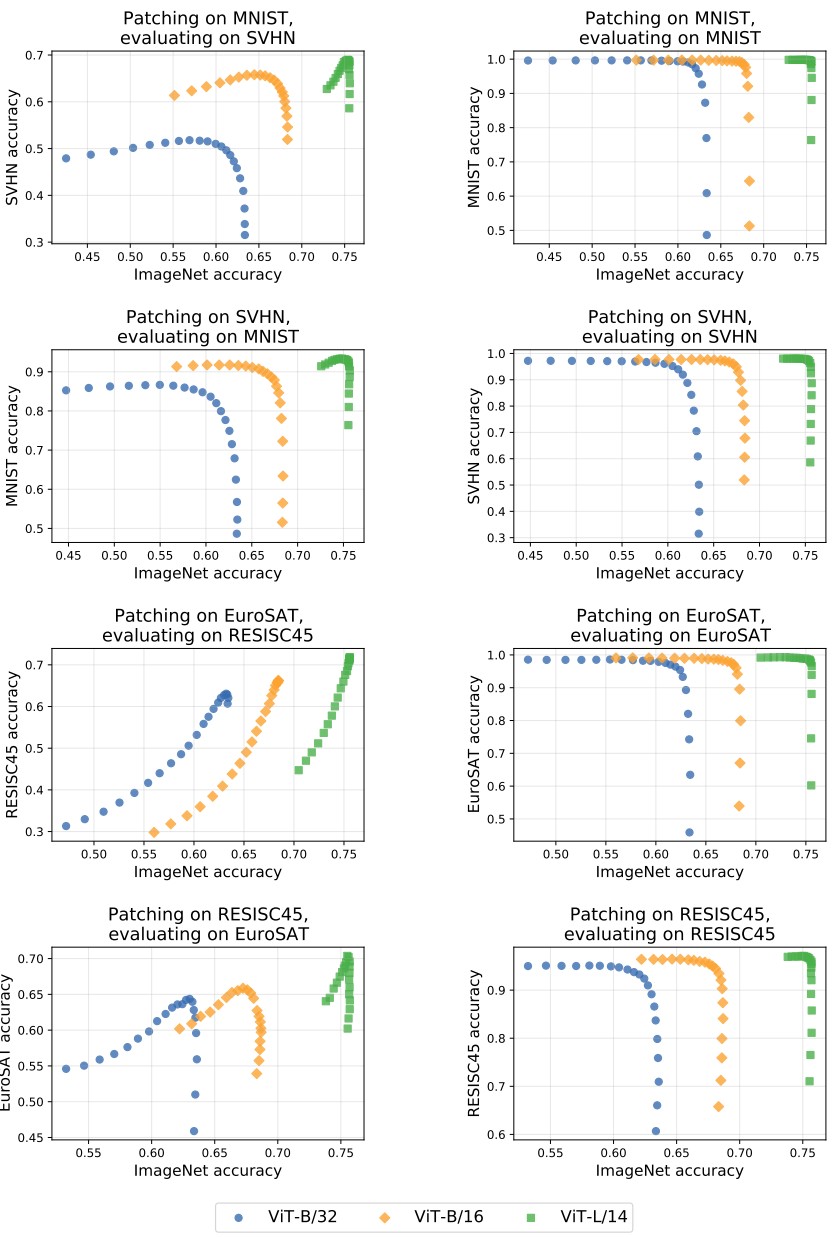

Figure 23: **The effect of patching on related tasks.** In this figure we fine-tune on task $A$ then interpolate the weights of the fine-tuned and unpatched model. In addition to measuring accuracy of task $A$ and a supported task ImageNet, we also measure accuracy on a different task $B$. This figure along with Figure 24 supplement Table 2.

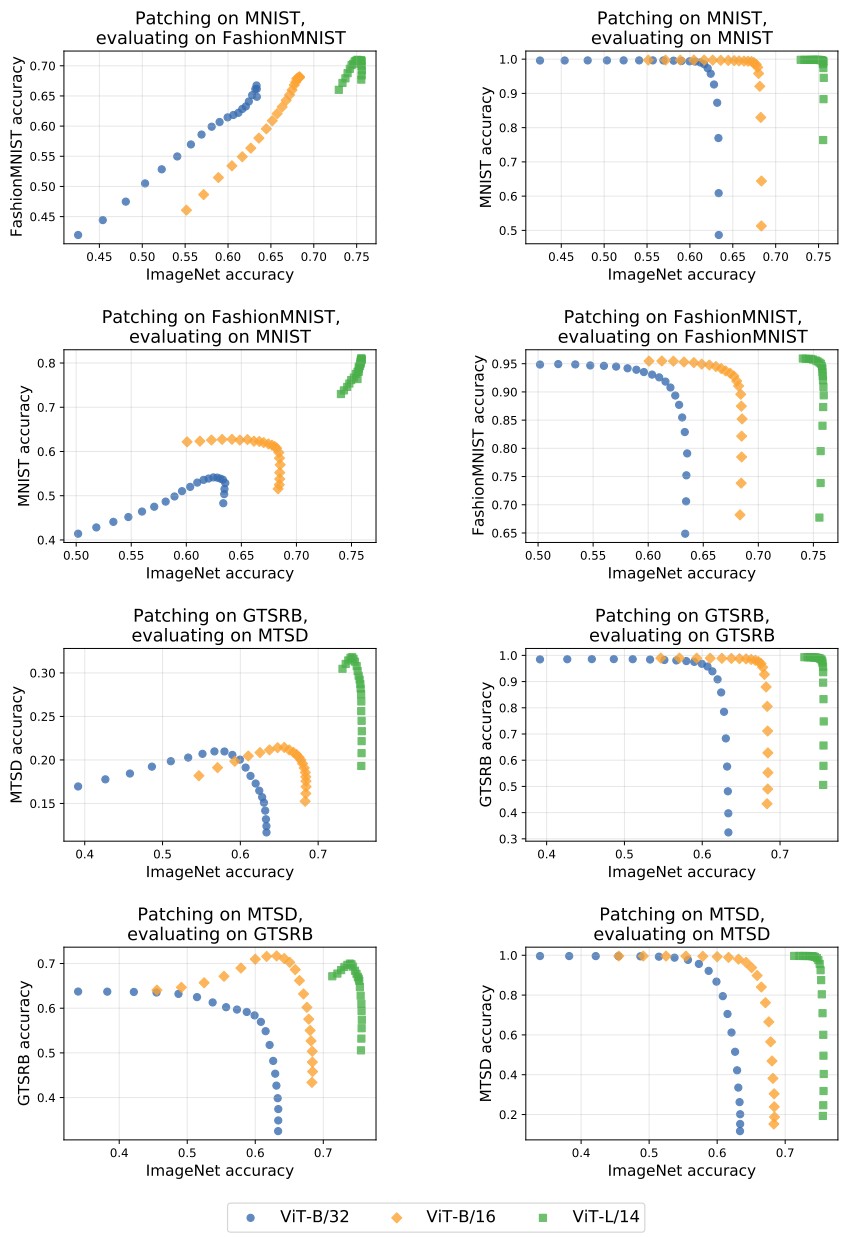

Figure 24: **The effect of patching on related tasks.** In this figure we fine-tune on task $A$ then interpolate the weights of the fine-tuned and unpatched model. In addition to measuring accuracy of task $A$ and a supported task ImageNet, we also measure accuracy on a different task $B$. This figure along with Figure 23 supplements Table 2.

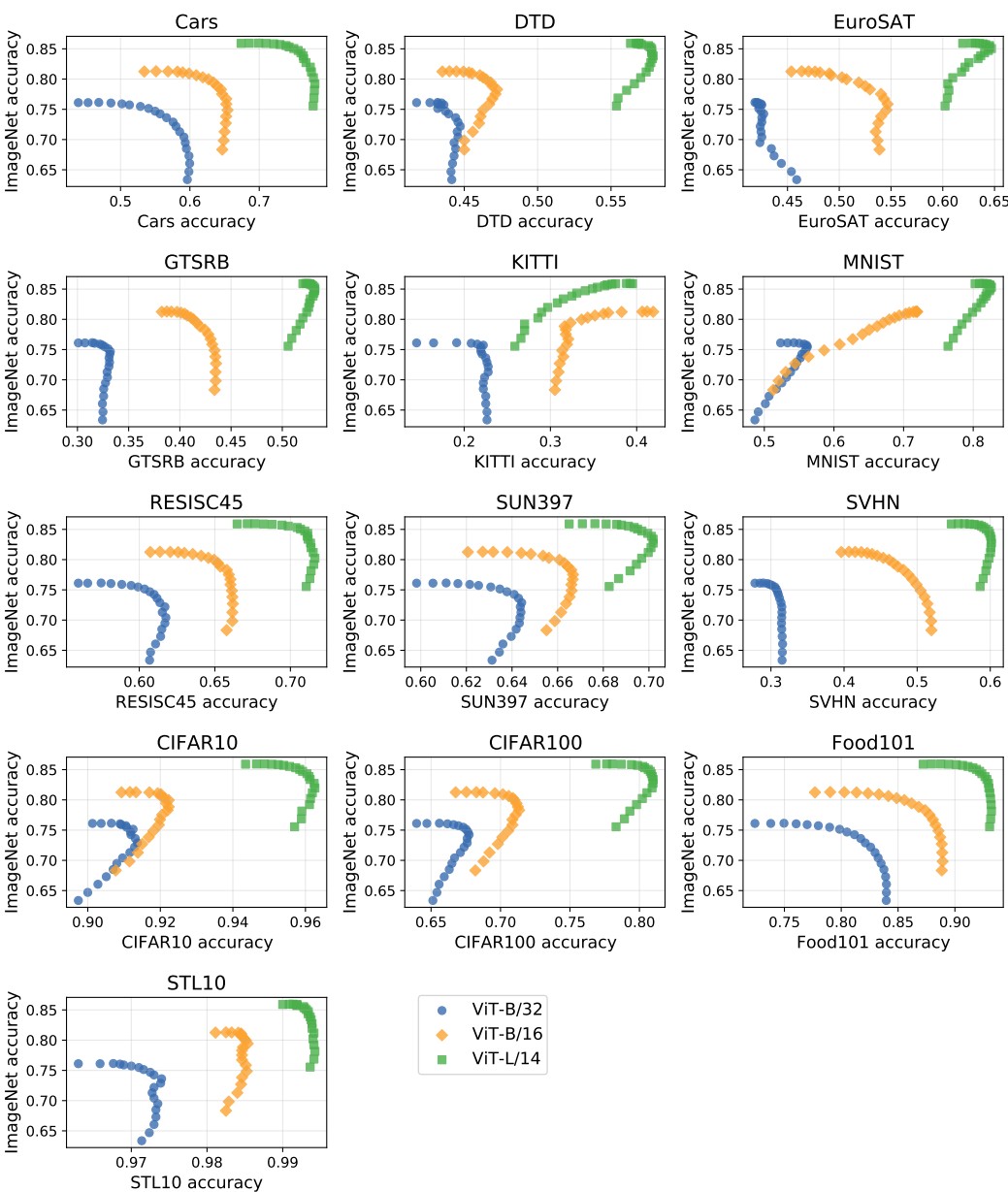

Figure 25: **Measuring broad transfer to 13 image classification datasets when fine-tuning on ImageNet.** We fine-tune on ImageNet then interpolate with the unpatched model with mixing coefficients $\alpha \in \{0, 0.05, ..., 1.0\}$. In addition to measuring ImageNet accuracy, we also measure open-vocabulary accuracy on 13 tasks.

## J   Patching models on multiple tasks

This section expands the experiments and results on patching models on multiple tasks from Section 5. Figure 26 illustrates the different patching strategies.

We first show an exhaustive search over the mixing coefficients for parallel patching on pairs of datasets. Next, we provide additional results for sequential patching, including additional orders in which the datasets are seen and how the number of datasets affects the quality of patching. Finally, we present results on SplitCIFAR [61].

### J.1   Exhaustive parallel search

Recall from Section 5 that exhaustively searching the mixing coefficients is prohibitively expensive when many patching tasks are used. Here, we study patching on two patching tasks, for which an exhaustive search is still feasible, and contrast it with other search strategies. More specifically, we examine three pairs of datasets: i) MNIST and EuroSAT; ii) MNIST and DTD; and iii) Cars and DTD. The results are shown in Figures 27 to 29. For two patching tasks $\mathcal{D}_1$ and $\mathcal{D}_2$, let $\theta_{\text{ft}}^{(1)}$ and $\theta_{\text{ft}}^{(2)}$ be the models fine-tuned on them. We then measure accuracy of models $\theta = (1 - \alpha_1 - \alpha_2) \cdot \theta_{\text{zs}} + \alpha_1 \cdot \theta_{\text{ft}}^{(1)} + \alpha_2 \cdot \theta_{\text{ft}}^{(2)}$ for $\alpha_1, \alpha_2 \in [0, 1]$. In most cases, there exists some values of the mixing coefficients such that accuracy is high on all the three tasks. For instance, when patching a ViT-L/14 on MNIST and EuroSAT, when $\alpha_1 = 0.35$ and $\alpha_2 = 0.45$, accuracy on MNIST and EuroSAT is 39 and 23 percentage points higher compared to the unpatched model, while accuracy on ImageNet decreases by less than 1 percentage point. Note that the area of high average accuracy typically increases with scale, supporting findings of Section 4.1.

In Table 5 we contrast the average accuracy obtained via exhaustive search with using other search strategies, *uniform search* and *black-box optimization*. Recall that these methods optimize for average accuracy on the validation sets. The uniform search strategy, also described in Section 5, consists of searching over a single scalar $\beta \in [0, 1]$, inspecting the models $\theta = (1-\beta) \cdot \theta_{\text{zs}} + \alpha/2 \cdot \theta_{\text{ft}}^{(1)} + \alpha/2 \cdot \theta_{\text{ft}}^{(2)}$. For black-box optimization, we explore an adaptive black box optimization algorithm, Nevergrad [60][7] over coefficients $\alpha^{(1)}, \alpha^{(2)} \in [0, 1]$ for the model $\theta = (1-\alpha_1-\alpha_2) \cdot \theta_{\text{zs}} + \alpha_1 \cdot \theta_{\text{ft}}^{(1)} + \alpha_2 \cdot \theta_{\text{ft}}^{(2)}$. We initialize all $\alpha^{(1)}$ and $\alpha^{(2)}$ with $1/2$, and run for 50 optimization steps. As shown in Table 5, uniform search and black-box optimization are comparable in performance, and substantially outperform the unpatched model. However, both strategies still lag behind exhaustive search, indicating headroom for more sophisticated search strategies.

---

[7]https://facebookresearch.github.io/nevergrad/

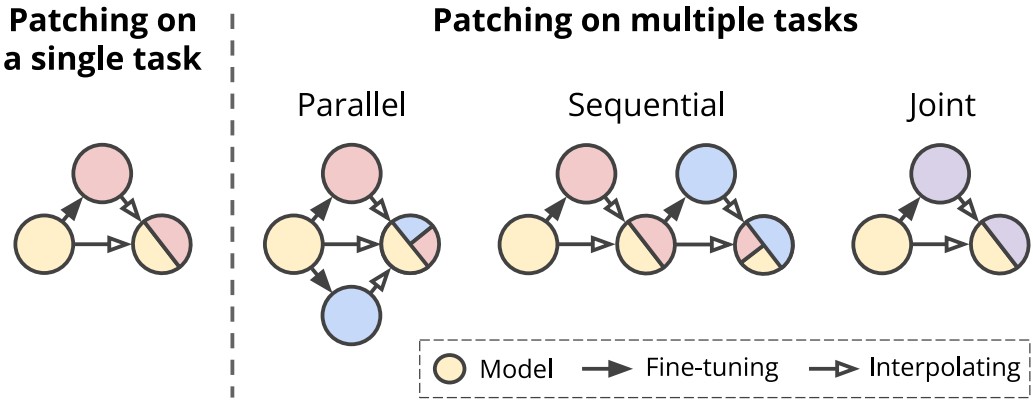

Figure 26: **An illustration of different patching strategies.** The area proportions on the circles that represent patched models are merely illustrative. In practice, the mixing coefficients are determined based on held-out validation sets. These diagrams are inspired by Matena and Raffel [47].

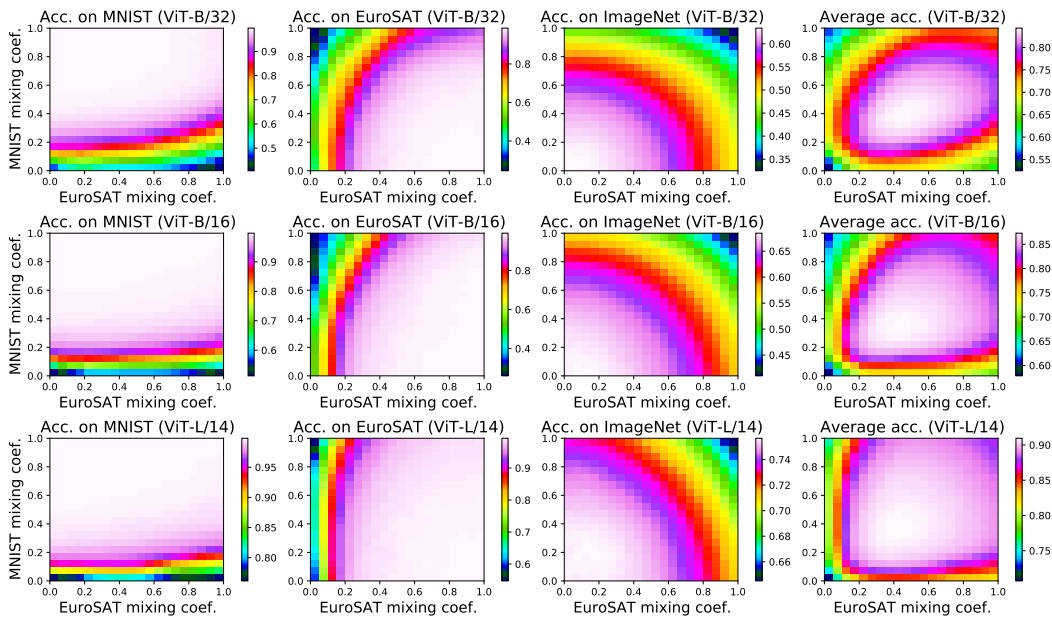

Figure 27: Exhaustive search for parallel patching on MNIST and EuroSAT.

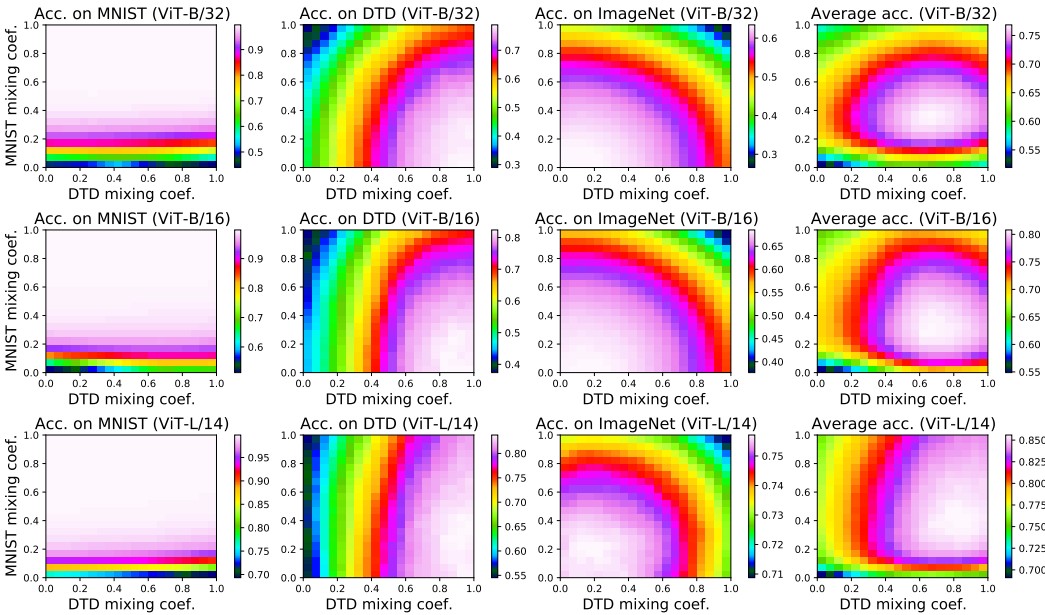

Figure 28: Exhaustive search for parallel patching on MNIST and DTD.

| | MNIST/EuroSAT | | | | MNIST/DTD | | | | Cars /DTD | | | |
|---|---|---|---|---|---|---|---|---|---|---|---|---|
| | ZS | U | BB | E | ZS | U | BB | E | ZS | U | BB | E |
| ViT-B/32 | 52.5 | 78.1 | 78.1 | 84.3 | 51.9 | 70.7 | 72.0 | 76.8 | 55.7 | 65.6 | 65.7 | 68.3 |
| ViT-B/16 | 57.8 | 81.8 | 81.9 | 87.3 | 54.9 | 75.1 | 76.5 | 80.8 | 59.3 | 72.0 | 71.9 | 74.7 |
| ViT-L/14 | 70.7 | 86.9 | 87.0 | 90.9 | 69.1 | 81.8 | 82.6 | 85.6 | 69.6 | 79.6 | 79.8 | 82.3 |

Table 5: **Contrasting multiple search strategies for parallel patching.** *ZS*: zero-shot; *U*: uniform search; *BB*: black-box optimization; *E*: exhaustive search.

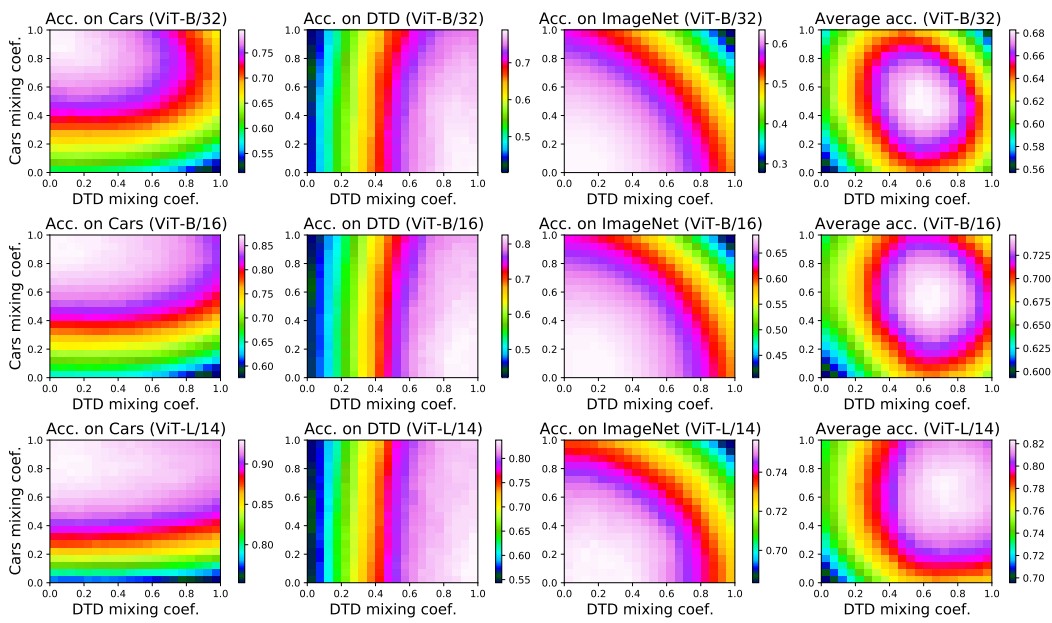

Figure 29: Exhaustive search for parallel patching on Cars and DTD.

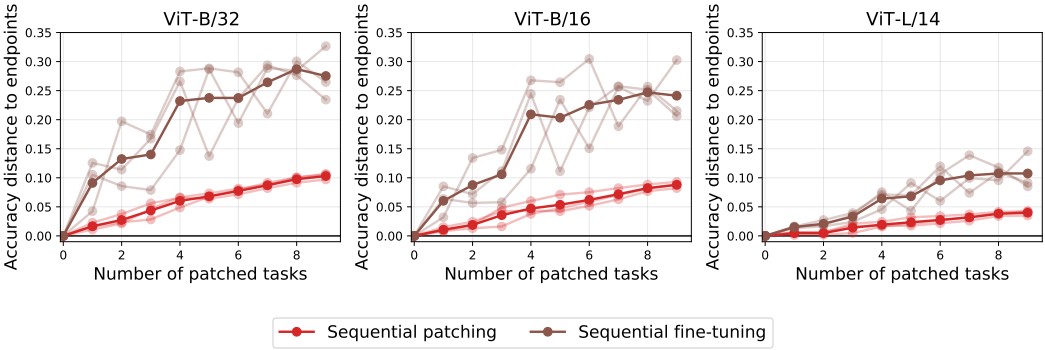

Figure 30: **The evolution of sequential patching as more tasks are added.** We show how a single model patched with different strategies compares to using one specialized model per task, as more patching tasks are added—the difference in average accuracy is shown in the $y$-axis, and referred to as accuracy distance to endpoints (Section D). When sequentially patching, the distance in accuracy to using specialized models increases when more tasks are added, although substantially less than when using sequential fine-tuning (without weight interpolation). Results are shown for nine patching tasks, for three different random seeds that control the order in which datasets are seen. The average across random seeds is highlighted.

### J.2 Sequential patching

In Figure 30, we show the evolution of sequential patching as more tasks are added. The accuracy distance of using a single, patched model to using multiple specialized models increases with with the number of patched tasks, leaving headroom for future work on more sophisticated sequential strategies for patching. Interestingly, sequential patching outperforms sequential fine-tuning (where no interpolation is used) by a large margin.

### J.3 SplitCIFAR

Figure 31 compares the patching methods described in Section 2 when patching on ten tasks from SplitCIFAR100 [61]. We split CIFAR100 randomly into ten different 10-way classification problems which are either learned jointly (as in joint fine-tuning or joint patching) or independently. In Figure 31 we show how the number of tasks learned affects accuracy on i) ImageNet, the supported task used

for this experiment (first row), ii) the patching tasks (second row) and iii) average accuracy on the patching tasks averaged with ImageNet accuracy (third row). We also display accuracy for two additional tasks, Food101 and STL10, in rows four and five, respectively.

While this experiment explores PAINT in a more conventional continual learning setting, there is a key difference: we also examine model accuracy on other tasks like ImageNet and Food101. To choose the mixing coefficients we optimize for performance on the held-out validation set of all patching tasks seen so far (i.e, the SplitCIFAR100 tasks) and the supported task ImageNet with equal weight. This is the reason for also examining accuracy on Food101 and STL10—we want to make sure that we are not overfitting to the representative supported task, ImageNet.

As intended, PAINT show less catastrophic forgetting than the alternative approaches on ImageNet, Food101, and STL10. All methods are fine-tuned with a total of 2,000 iterations. As such, we only use 200 iterations per-task when fine-tuning independently.

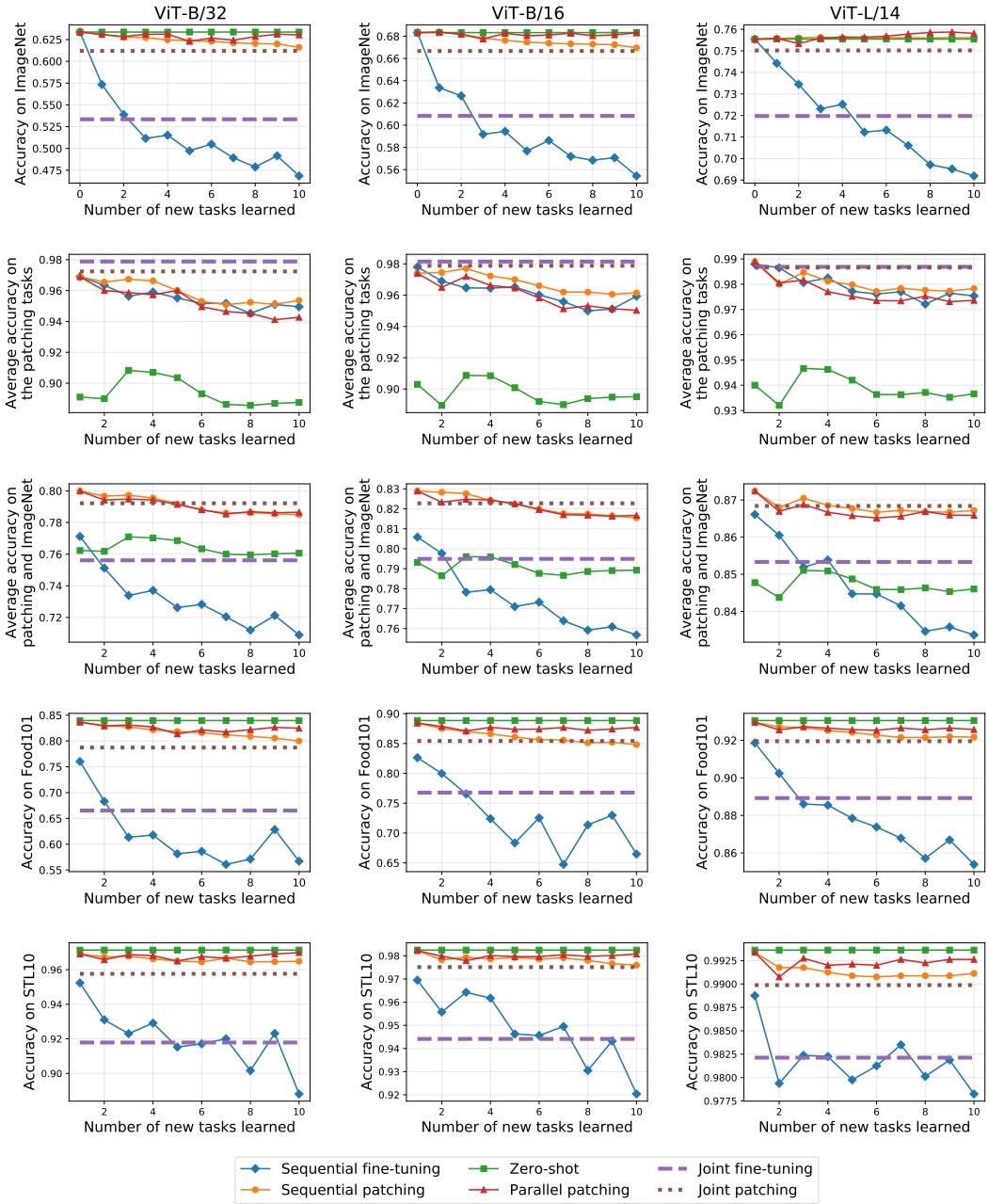

Figure 31: Comparing the methods from Section 2 when patching on ten tasks from SplitCIFAR [61]. When choosing the mixing coefficient we optimize for average accuracy on the held-out validation sets of the supported task ImageNet and the new patching tasks. We also show performance on additional supported tasks Food101 and STL10 to ensure PAINT does not overfit to the supported task used for choosing $\alpha$. PAINT show less catastrophic forgetting than the alternative approaches on ImageNet, Food101, and STL10.

# K  Typographic attacks

In this section we present more details on our typographic attacks experiments presented in Section 7. We first discuss our procedure for creating synthetic typographic attacks. We then outline our real world data collection scheme. Finally we present additional experimental details.

**Synthetic dataset details.**  Starting with the SUN397 [84] dataset, we procedurally add text to images as seen in Figure 32. We resize the shorter dimension to 224 pixels using bicubic interpolation and take a 224 pixel by 224 pixel center crop, which is the standard CLIP resize and crop augmentation [57]. We randomize over three fonts: Courier, Helvetica and Times. For font size, we randomly sample over the range 20 to 40 points. We consider eight colors, and sample uniformly from them: red, green, blue, cyan, magenta, yellow, white, and black. To make sure font is visible, we outline text with a 1 point shadow that is a different color than the main font color. The text is randomly placed in the image with checks to ensure that the text fits completely on the image. The text specifies an incorrect class label, chosen at random. For instance, if the ground truth label from an image is "raceway", the text content will be uniformly sampled from the $397 - 1 = 396$ other options. We apply this typographic attack procedure on the originally SUN397 train and test sets to create attacked train and test sets.

**Real dataset details.**  To test if patching on synthetic typographic attacks transfers to real-world use-cases, we generate a real-world test set of typographic attacks. We provide the following instructions to in-house annotators: 1) Write the name of an object on a sticky note, then place the sticky note on an object that is not the one that you have written. For instance, you can write "desk" on the sticky note then put it on a mug. The objects should be reasonable categories (e.g., common objects, animals, etc.). 2) Take a picture—make sure that the underlying object is centered and the text is visible. We acquire consent from each annotator to use their images for research purposes and to release data publicly. No faces or other identifying human characteristics are present in the dataset.

We show images from our dataset in Figure 33, highlighting the category diversity. As highlights: calipers are attacked with text saying "ruler," an artichoke is attacked with text saying "pineapple," and a GPU is attacked with text saying "CPU." Critically, none of the object categories—either in the image or in the text attack—overlap with those in SUN397. This allows us to study typographic attacks in the broad transfer setting (see Section 6 for more details).

**Experimental details.**  We first fine-tune on the attacked version of SUN397, using our standard fine-tuning setup discussed in Section 2. We then evaluate interpolated models on attacked SUN397 and real-world test sets. For SUN397, this amounts to standard multi-class classification as the class space is fixed. For real typographic attacks, the task is binary classification between the image category and the text category. The classification head for each image is generated using the frozen CLIP text encoder.

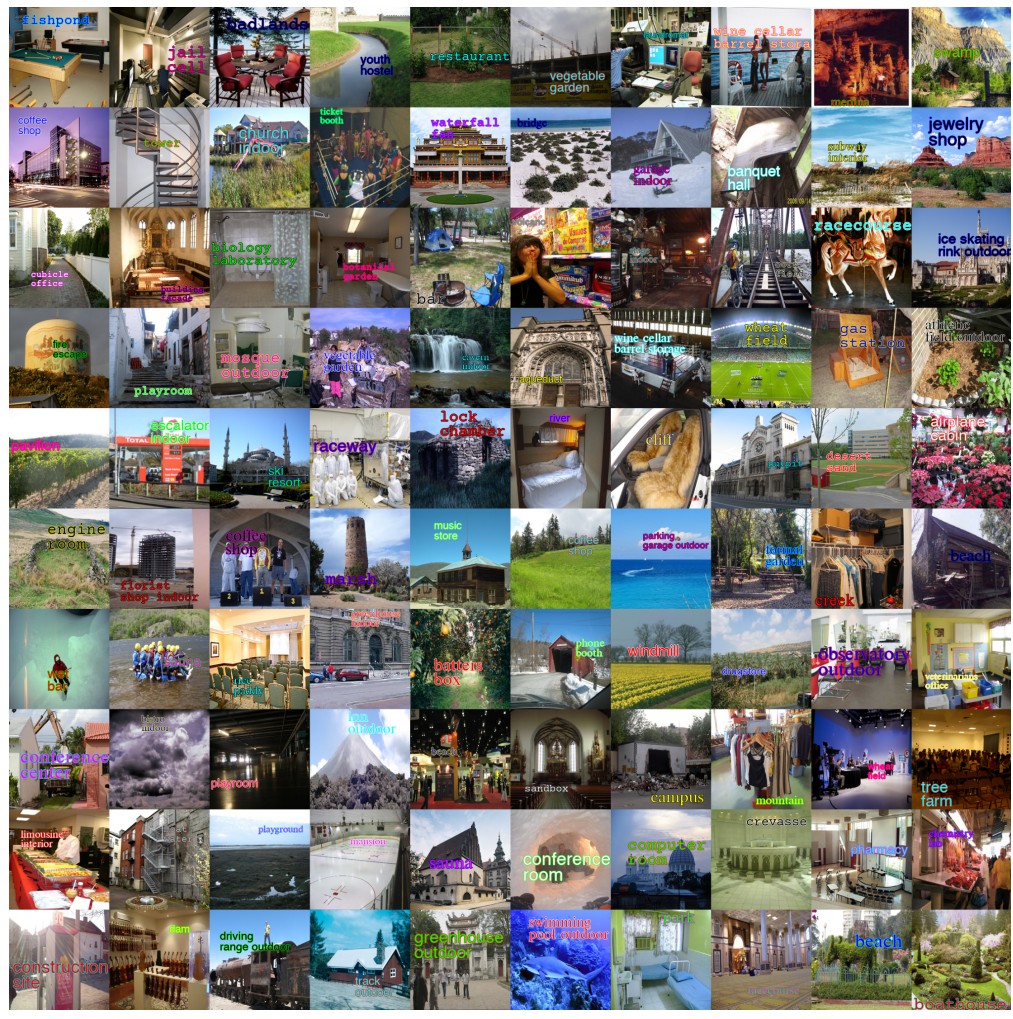

Figure 32: **Synthetically generated typographic attacks used for patching.** Starting with the SUN397 dataset, we procedurally modify images by adding text specifying an incorrect class label for the image. By patching on this data, we observe transfer to real-world classification of images visualized in Figure 33—even with no class-space overlap.

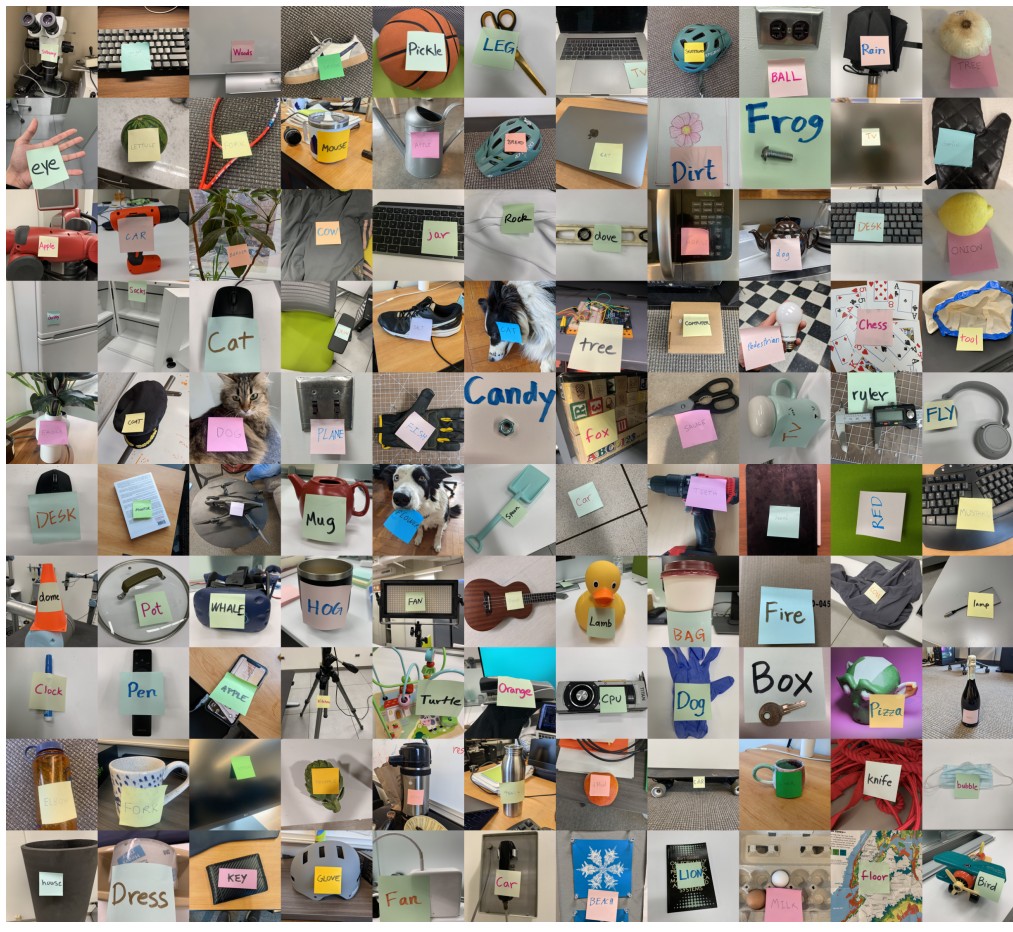

Figure 33: **Real typographic attacks used for testing broad transfer.** Here we present all 110 images from our real-world typographic attacks dataset. Annotators were asked to choose both the target objects and the textual corruptions. Hence, during data collection there were no restrictions on the true object semantic category or the typographic attack semantic category. Note that text is visually dissimilar for the computer generated fonts in Figure 32. Additionally, images in this dataset are *object-centric*, whereas images in Figure 32 capture a distribution of scenes.

## L Counting

We discuss specifics about the dataset and experimental setup for evaluating broad transfer for object counting.

**Dataset details.** We consider the CLEVR dataset [32] with annotations for the number of objects per image. We split the dataset based on its classes into train and test sets as seen in Figure 34. By training on one split and testing on the other with unseen classes, we are able to evaluate broad transfer for this counting task.

**Experimental details.** The patching task is a 5-way classification task. The evaluation task is a 3-way classification task, with a different set of classes. Again the head for the test-time task is created zero-shot using the CLIP text encoder. Full interpolation curves for transfer and the original task are presented in Figure 35.

## M Visual Question Answering

For multiple-choice visual question answering, we use CLIP to contrast images with each candidate answer. For each candidate answer, we construct a prompt that also includes the question following the template "Question: [question text] Answer: [answer text]". This text is fed to CLIP's text encoder, which remains frozen during the patching process.[8] During fine-tuning, the weights of the vision encoder are updated using a contrastive loss: the feature similarity with the text features of the correct prompt are maximized with respect to the other candidates. We patch and evaluate on multiple-choice VQA v1 [4], where each question is associated with 18 candidate answers. Interestingly, 87% of the prompts used for evaluation are not present in the training data. The results are shown in Figure 36. Patching improves VQA performance by 13 to 18 percentage points, while reducing ImageNet accuracy by less than one percentage point.

## N Computational resources

For all of our experiments, we used NVIDIA A-40 GPUs with 46GB of RAM from an internal cluster. We estimate the total amount of compute used for the experiments in this paper is around 10 thousand GPU hours.

---

[8]We found that unfreezing the text encoder improved accuracy, but we present results with an frozen text encoder for experimental consistency.

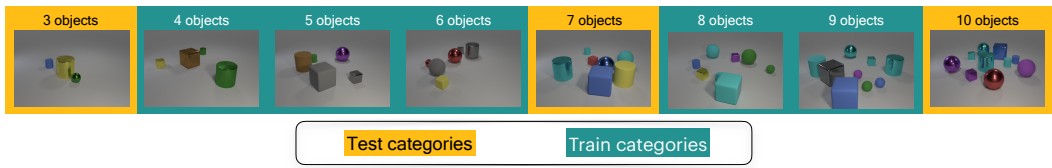

Figure 34: **CLEVR Counting images and splits.** Each image has a number of objects between 3 and 10. For a patching (train) task, models are fine-tuned on images with 4, 5, 6, 8 or 9 objects. We test on images with 3, 7 or 10 objects to evaluate the model's interpolation and extrapolation of the class space.

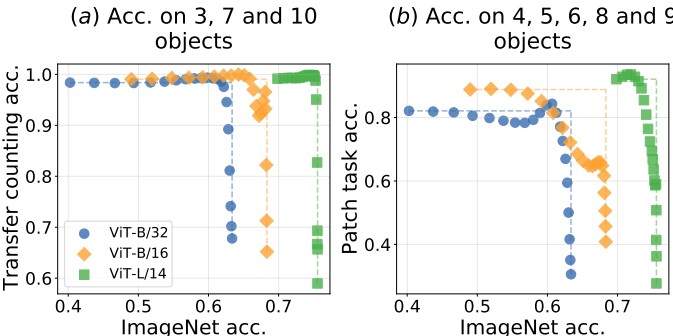

Figure 35: **CLEVR Counting broad transfer interpolation plots.** Curves for different mixing coefficients for (a) the broad transfer task and (b) the patching task. The curves suggest that training on the patching task transfer to the training task, even though the class spaces are different.

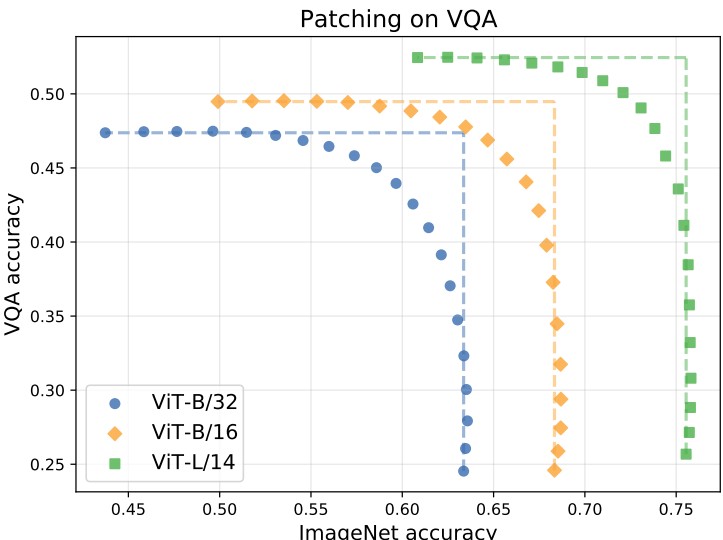

Figure 36: **Patching for visual question answering.** For all models, patching can improve the performance on multiple choice VQA v1 [4] with little loss in ImageNet accuracy.

## O    Tabular results

We now present numerical results for the main experiments presented in this paper. Tables 6 to 10 present results for patching on a single task, for various supported tasks. Table 11 details results for various strategies for patching on multiple tasks.

| | α | Cars S | Cars P | DTD S | DTD P | EuroSAT S | EuroSAT P | GTSRB S | GTSRB P | KITTI S | KITTI P | MNIST S | MNIST P | RESISC45 S | RESISC45 P | SUN397 S | SUN397 P | SVHN S | SVHN P | Avg |
|---|---|---|---|---|---|---|---|---|---|---|---|---|---|---|---|---|---|---|---|---|
| | 0.00 | 63.4 | 59.6 | 63.4 | 44.1 | 63.4 | 45.9 | 63.4 | 32.4 | 63.4 | 22.6 | 63.4 | 48.3 | 63.4 | 60.7 | 63.4 | 63.1 | 63.4 | 31.5 | 54.4 |
| | 0.05 | 63.4 | 61.9 | 63.4 | 47.0 | 63.4 | 63.4 | 63.4 | 39.7 | 63.4 | 25.5 | 63.4 | 60.9 | 63.5 | 65.9 | 63.5 | 64.4 | 63.4 | 39.9 | 57.7 |
| | 0.10 | 63.3 | 63.8 | 63.4 | 50.1 | 63.3 | 74.3 | 63.3 | 48.1 | 63.4 | 35.7 | 63.3 | 77.0 | 63.6 | 71.0 | 63.5 | 65.9 | 63.4 | 50.1 | 61.5 |
| | 0.15 | 63.3 | 65.5 | 63.4 | 53.7 | 63.2 | 82.0 | 63.2 | 57.6 | 63.5 | 47.5 | 63.2 | 87.3 | 63.5 | 75.9 | 63.5 | 67.2 | 63.3 | 60.9 | 64.9 |
| | 0.20 | 63.2 | 67.4 | 63.3 | 56.5 | 63.0 | 89.3 | 63.0 | 68.3 | 63.4 | 55.6 | 62.8 | 92.6 | 63.5 | 79.9 | 63.4 | 68.5 | 63.1 | 70.4 | 67.6 |
| | 0.25 | 63.1 | 69.1 | 63.1 | 59.7 | 62.7 | 93.3 | 62.8 | 78.5 | 63.3 | 60.2 | 62.4 | 95.7 | 63.3 | 83.7 | 63.3 | 69.5 | 62.8 | 78.3 | 69.7 |
| | 0.30 | 63.0 | 70.7 | 62.9 | 62.8 | 62.4 | 95.4 | 62.4 | 85.9 | 63.1 | 66.1 | 62.1 | 97.3 | 63.2 | 86.6 | 63.2 | 70.5 | 62.6 | 84.3 | 71.4 |
| | 0.35 | 62.8 | 72.4 | 62.7 | 65.7 | 62.0 | 96.4 | 62.0 | 90.9 | 62.8 | 69.3 | 61.7 | 98.5 | 63.0 | 89.1 | 63.1 | 71.6 | 62.1 | 88.8 | 72.5 |
| | 0.40 | 62.5 | 73.6 | 62.4 | 68.1 | 61.5 | 97.0 | 61.3 | 93.9 | 62.6 | 72.3 | 61.2 | 99.0 | 62.7 | 91.0 | 62.8 | 72.5 | 61.6 | 92.0 | 73.2 |
| | 0.45 | 62.3 | 74.8 | 62.0 | 70.1 | 61.0 | 97.6 | 60.7 | 95.7 | 62.3 | 73.4 | 60.6 | 99.3 | 62.4 | 92.4 | 62.5 | 73.4 | 61.1 | 94.0 | 73.6 |
| B/32 | 0.50 | 62.0 | 75.8 | 61.5 | 72.3 | 60.3 | 97.9 | 60.0 | 96.8 | 62.0 | 75.1 | 60.0 | 99.4 | 62.0 | 93.3 | 62.1 | 74.2 | 60.5 | 95.2 | 73.9 |
| | 0.55 | 61.6 | 76.4 | 60.9 | 74.2 | 59.5 | 98.1 | 59.0 | 97.5 | 61.5 | 76.9 | 59.0 | 99.5 | 61.6 | 93.7 | 61.7 | 74.7 | 59.6 | 96.1 | 74.0 |
| | 0.60 | 61.2 | 77.3 | 60.2 | 75.3 | 58.7 | 98.2 | 58.0 | 97.8 | 60.9 | 78.1 | 58.1 | 99.6 | 61.1 | 94.3 | 61.3 | 75.2 | 58.7 | 96.5 | 73.9 |
| | 0.65 | 60.6 | 78.0 | 59.3 | 76.2 | 57.7 | 98.4 | 56.7 | 98.1 | 60.2 | 79.2 | 56.9 | 99.6 | 60.4 | 94.7 | 60.6 | 75.6 | 57.7 | 96.7 | 73.7 |
| | 0.70 | 60.1 | 78.4 | 58.3 | 76.8 | 56.6 | 98.5 | 55.1 | 98.2 | 59.6 | 79.5 | 55.7 | 99.6 | 59.8 | 94.9 | 60.0 | 75.9 | 56.4 | 96.9 | 73.4 |
| | 0.75 | 59.5 | 78.9 | 57.3 | 77.7 | 55.4 | 98.6 | 53.3 | 98.4 | 58.7 | 79.5 | 54.1 | 99.6 | 58.9 | 95.1 | 59.3 | 76.1 | 55.0 | 97.0 | 72.9 |
| | 0.80 | 58.8 | 79.1 | 56.1 | 78.3 | 54.0 | 98.5 | 51.1 | 98.6 | 57.8 | 80.2 | 52.3 | 99.6 | 58.1 | 95.1 | 58.5 | 76.1 | 53.4 | 97.1 | 72.4 |
| | 0.85 | 58.1 | 79.6 | 54.7 | 78.6 | 52.6 | 98.5 | 48.6 | 98.6 | 56.8 | 79.2 | 50.3 | 99.6 | 57.0 | 95.0 | 57.7 | 76.0 | 51.6 | 97.1 | 71.6 |
| | 0.90 | 57.3 | 79.5 | 53.0 | 78.7 | 51.0 | 98.5 | 45.8 | 98.6 | 55.6 | 78.9 | 48.1 | 99.6 | 55.9 | 95.0 | 56.8 | 75.8 | 49.5 | 97.2 | 70.8 |
| | 0.95 | 56.2 | 79.4 | 51.3 | 78.9 | 49.1 | 98.5 | 42.7 | 98.5 | 54.3 | 78.2 | 45.4 | 99.6 | 54.6 | 95.1 | 55.8 | 75.4 | 47.2 | 97.2 | 69.9 |
| | 1.00 | 55.2 | 79.2 | 49.3 | 78.7 | 47.2 | 98.6 | 39.1 | 98.5 | 52.9 | 77.9 | 42.5 | 99.6 | 53.2 | 95.0 | 54.6 | 75.1 | 44.7 | 97.2 | 68.8 |
| | 0.00 | 68.3 | 64.7 | 68.3 | 45.0 | 68.3 | 53.9 | 68.3 | 43.4 | 68.3 | 30.5 | 68.3 | 51.3 | 68.3 | 65.8 | 68.3 | 65.5 | 68.3 | 52.0 | 60.4 |
| | 0.05 | 68.4 | 66.9 | 68.4 | 47.4 | 68.4 | 66.8 | 68.4 | 49.0 | 68.4 | 40.5 | 68.3 | 64.4 | 68.5 | 71.2 | 68.5 | 66.9 | 68.4 | 60.6 | 63.9 |
| | 0.10 | 68.4 | 68.9 | 68.4 | 49.7 | 68.5 | 80.0 | 68.4 | 55.3 | 68.4 | 53.9 | 68.3 | 83.0 | 68.6 | 75.9 | 68.5 | 68.4 | 68.4 | 67.8 | 67.7 |
| | 0.15 | 68.4 | 71.1 | 68.5 | 53.0 | 68.4 | 89.5 | 68.4 | 62.8 | 68.5 | 58.5 | 68.1 | 92.1 | 68.6 | 80.0 | 68.6 | 69.7 | 68.4 | 74.5 | 70.4 |
| | 0.20 | 68.4 | 73.3 | 68.5 | 56.5 | 68.2 | 94.1 | 68.4 | 71.1 | 68.5 | 59.8 | 68.0 | 95.8 | 68.7 | 84.1 | 68.6 | 71.0 | 68.2 | 80.4 | 72.3 |
| | 0.25 | 68.3 | 75.4 | 68.4 | 59.5 | 68.0 | 96.1 | 68.4 | 80.5 | 68.4 | 62.2 | 67.9 | 97.6 | 68.7 | 87.4 | 68.6 | 72.3 | 68.1 | 85.6 | 74.0 |
| | 0.30 | 68.2 | 77.2 | 68.3 | 63.6 | 67.8 | 97.1 | 68.2 | 88.0 | 68.4 | 68.4 | 67.7 | 98.4 | 68.6 | 90.3 | 68.5 | 73.4 | 67.9 | 89.8 | 75.5 |
| | 0.35 | 68.1 | 79.1 | 68.1 | 67.0 | 67.5 | 97.6 | 68.0 | 92.7 | 68.3 | 72.3 | 67.5 | 99.0 | 68.5 | 92.1 | 68.5 | 74.5 | 67.7 | 92.9 | 76.6 |
| | 0.40 | 67.8 | 80.9 | 67.8 | 70.4 | 67.2 | 98.0 | 67.8 | 95.5 | 68.2 | 74.0 | 67.2 | 99.3 | 68.4 | 93.5 | 68.3 | 75.4 | 67.5 | 94.9 | 77.3 |
| | 0.45 | 67.6 | 82.0 | 67.5 | 73.5 | 66.7 | 98.3 | 67.5 | 96.9 | 67.9 | 75.9 | 66.8 | 99.4 | 68.1 | 94.5 | 68.1 | 76.2 | 67.1 | 96.1 | 77.8 |
| B/16 | 0.50 | 67.3 | 83.5 | 67.1 | 76.1 | 66.3 | 98.6 | 67.2 | 97.6 | 67.7 | 78.3 | 66.3 | 99.4 | 67.9 | 95.0 | 67.9 | 77.0 | 66.7 | 96.7 | 78.1 |
| | 0.55 | 67.0 | 84.5 | 66.7 | 77.6 | 65.8 | 98.7 | 66.8 | 98.1 | 67.4 | 79.5 | 65.7 | 99.5 | 67.6 | 95.5 | 67.6 | 77.7 | 66.2 | 97.2 | 78.3 |
| | 0.60 | 66.7 | 85.2 | 66.1 | 79.4 | 65.2 | 98.7 | 66.3 | 98.4 | 67.1 | 81.4 | 65.2 | 99.6 | 67.2 | 95.8 | 67.3 | 78.2 | 65.8 | 97.4 | 78.4 |
| | 0.65 | 66.3 | 85.9 | 65.5 | 80.6 | 64.6 | 98.9 | 65.6 | 98.7 | 66.7 | 82.6 | 64.5 | 99.6 | 66.8 | 96.0 | 66.9 | 78.4 | 65.1 | 97.6 | 78.3 |
| | 0.70 | 65.9 | 86.3 | 64.7 | 81.1 | 63.8 | 99.0 | 64.8 | 98.7 | 66.3 | 83.8 | 63.6 | 99.7 | 66.4 | 96.2 | 66.3 | 78.7 | 64.3 | 97.6 | 78.2 |
| | 0.75 | 65.3 | 86.8 | 63.9 | 81.3 | 62.9 | 99.0 | 63.7 | 98.8 | 65.9 | 84.2 | 62.7 | 99.7 | 65.9 | 96.3 | 65.8 | 78.9 | 63.5 | 97.7 | 77.9 |
| | 0.80 | 64.8 | 87.0 | 63.0 | 82.4 | 61.9 | 99.0 | 62.5 | 98.9 | 65.3 | 84.7 | 61.7 | 99.7 | 65.3 | 96.4 | 65.1 | 79.1 | 62.6 | 97.7 | 77.6 |
| | 0.85 | 64.1 | 87.2 | 61.9 | 82.3 | 60.6 | 99.0 | 61.0 | 98.9 | 64.7 | 84.7 | 60.5 | 99.7 | 64.6 | 96.4 | 64.5 | 79.2 | 61.5 | 97.7 | 77.1 |
| | 0.90 | 63.5 | 87.2 | 60.6 | 82.3 | 59.3 | 99.1 | 59.3 | 98.9 | 64.1 | 84.8 | 58.9 | 99.7 | 63.9 | 96.4 | 63.6 | 79.3 | 60.1 | 97.7 | 76.6 |
| | 0.95 | 62.7 | 87.2 | 59.2 | 82.2 | 57.7 | 99.1 | 57.1 | 99.0 | 63.3 | 84.7 | 57.2 | 99.7 | 63.2 | 96.4 | 62.7 | 79.2 | 58.6 | 97.7 | 75.9 |
| | 1.00 | 61.9 | 87.0 | 57.5 | 82.3 | 56.0 | 99.1 | 54.7 | 99.0 | 62.4 | 84.8 | 55.2 | 99.7 | 62.2 | 96.4 | 61.7 | 79.0 | 56.8 | 97.7 | 75.2 |
| | 0.00 | 75.5 | 77.8 | 75.5 | 55.4 | 75.5 | 60.2 | 75.5 | 50.6 | 75.5 | 25.9 | 75.5 | 76.4 | 75.5 | 71.0 | 75.5 | 68.3 | 75.5 | 58.6 | 68.0 |
| | 0.05 | 75.6 | 79.9 | 75.6 | 57.8 | 75.5 | 74.6 | 75.6 | 57.9 | 75.6 | 33.2 | 75.6 | 88.3 | 75.6 | 76.5 | 75.7 | 69.7 | 75.6 | 67.0 | 71.4 |
| | 0.10 | 75.5 | 82.1 | 75.6 | 60.6 | 75.6 | 88.1 | 75.6 | 65.6 | 75.5 | 47.0 | 75.6 | 94.5 | 75.7 | 81.1 | 75.8 | 71.3 | 75.6 | 73.2 | 74.7 |
| | 0.15 | 75.6 | 84.0 | 75.6 | 63.8 | 75.6 | 93.9 | 75.6 | 74.8 | 75.6 | 54.3 | 75.6 | 97.4 | 75.7 | 85.8 | 75.8 | 72.9 | 75.6 | 78.9 | 77.0 |
| | 0.20 | 75.6 | 85.8 | 75.6 | 66.8 | 75.6 | 96.6 | 75.6 | 83.3 | 75.5 | 60.1 | 75.6 | 98.6 | 75.7 | 89.3 | 75.8 | 74.3 | 75.7 | 84.1 | 78.8 |
| | 0.25 | 75.5 | 87.5 | 75.6 | 69.5 | 75.5 | 97.7 | 75.5 | 89.6 | 75.5 | 65.8 | 75.5 | 99.2 | 75.7 | 92.1 | 75.8 | 75.5 | 75.7 | 88.7 | 80.3 |
| | 0.30 | 75.4 | 88.5 | 75.6 | 71.9 | 75.4 | 98.1 | 75.5 | 93.6 | 75.5 | 70.6 | 75.6 | 99.4 | 75.7 | 93.5 | 75.9 | 76.8 | 75.6 | 92.5 | 81.4 |
| | 0.35 | 75.4 | 89.7 | 75.6 | 73.8 | 75.4 | 98.4 | 75.5 | 95.7 | 75.5 | 73.0 | 75.5 | 99.5 | 75.7 | 94.7 | 75.8 | 77.9 | 75.6 | 94.9 | 82.1 |
| | 0.40 | 75.3 | 90.4 | 75.5 | 75.7 | 75.3 | 98.6 | 75.5 | 97.0 | 75.4 | 75.8 | 75.4 | 99.6 | 75.7 | 95.5 | 75.8 | 78.9 | 75.5 | 96.3 | 82.6 |
| | 0.45 | 75.3 | 91.0 | 75.4 | 78.0 | 75.1 | 98.8 | 75.4 | 97.8 | 75.3 | 77.5 | 75.3 | 99.7 | 75.7 | 95.9 | 75.8 | 79.8 | 75.4 | 97.1 | 83.0 |
| L/14 | 0.50 | 75.2 | 91.6 | 75.3 | 79.1 | 74.9 | 98.9 | 75.3 | 98.4 | 75.2 | 80.6 | 75.2 | 99.8 | 75.6 | 96.2 | 75.7 | 80.6 | 75.3 | 97.5 | 83.4 |
| | 0.55 | 75.1 | 92.1 | 75.1 | 80.3 | 74.7 | 98.9 | 75.2 | 98.7 | 75.1 | 82.8 | 75.1 | 99.8 | 75.5 | 96.6 | 75.5 | 81.3 | 75.2 | 97.7 | 83.6 |
| | 0.60 | 75.0 | 92.4 | 74.9 | 81.2 | 74.4 | 99.0 | 75.1 | 98.9 | 75.0 | 84.0 | 74.9 | 99.8 | 75.4 | 96.8 | 75.4 | 81.8 | 75.0 | 97.9 | 83.7 |
| | 0.65 | 74.9 | 92.6 | 74.7 | 82.1 | 74.1 | 99.1 | 75.0 | 99.0 | 75.0 | 84.7 | 74.8 | 99.8 | 75.3 | 96.9 | 75.2 | 82.1 | 74.8 | 97.9 | 83.8 |
| | 0.70 | 74.7 | 92.8 | 74.5 | 82.3 | 73.8 | 99.1 | 74.7 | 99.1 | 74.8 | 85.4 | 74.6 | 99.8 | 75.2 | 97.0 | 75.0 | 82.4 | 74.6 | 98.0 | 83.8 |
| | 0.75 | 74.5 | 92.8 | 74.2 | 82.7 | 73.4 | 99.2 | 74.6 | 99.2 | 74.7 | 85.7 | 74.4 | 99.8 | 75.1 | 97.1 | 74.6 | 82.6 | 74.3 | 98.0 | 83.7 |
| | 0.80 | 74.3 | 92.9 | 73.9 | 82.8 | 73.0 | 99.3 | 74.4 | 99.2 | 74.5 | 85.7 | 74.2 | 99.8 | 74.9 | 97.1 | 74.4 | 82.7 | 74.0 | 98.0 | 83.6 |
| | 0.85 | 74.0 | 92.9 | 73.6 | 83.0 | 72.4 | 99.3 | 74.2 | 99.2 | 74.4 | 85.8 | 73.9 | 99.8 | 74.7 | 97.0 | 74.0 | 82.8 | 73.8 | 98.1 | 83.5 |
| | 0.90 | 73.8 | 92.8 | 73.2 | 83.6 | 71.8 | 99.3 | 73.9 | 99.3 | 74.1 | 86.2 | 73.7 | 99.8 | 74.4 | 97.0 | 73.6 | 82.7 | 73.4 | 98.0 | 83.4 |
| | 0.95 | 73.5 | 92.8 | 72.8 | 83.6 | 71.2 | 99.2 | 73.5 | 99.3 | 73.9 | 86.4 | 73.3 | 99.8 | 74.1 | 97.0 | 73.2 | 82.6 | 73.0 | 98.0 | 83.2 |
| | 1.00 | 73.1 | 92.8 | 72.3 | 83.7 | 70.5 | 99.2 | 73.1 | 99.3 | 73.6 | 86.4 | 72.9 | 99.8 | 73.8 | 96.9 | 72.7 | 82.4 | 72.6 | 98.0 | 82.9 |

Table 6: **Patching on a single new task where ImageNet is the supported task.** *S*: accuracy on the supported task (ImageNet); *P*: accuracy on the patching task; *Avg:* average accuracy on patching and supported tasks.

| | α | Cars S | Cars P | DTD S | DTD P | EuroSAT S | EuroSAT P | GTSRB S | GTSRB P | KITTI S | KITTI P | MNIST S | MNIST P | RESISC45 S | RESISC45 P | SUN397 S | SUN397 P | SVHN S | SVHN P | Avg |
|---|---|---|---|---|---|---|---|---|---|---|---|---|---|---|---|---|---|---|---|---|
| | 0.00 | 89.7 | 59.6 | 89.8 | 44.1 | 89.7 | 45.9 | 89.7 | 32.4 | 89.7 | 22.6 | 89.7 | 48.3 | 89.8 | 60.7 | 89.7 | 63.1 | 89.7 | 31.5 | 67.6 |
| | 0.05 | 89.6 | 61.9 | 89.7 | 47.0 | 90.1 | 63.4 | 90.1 | 39.7 | 89.8 | 25.5 | 90.2 | 60.9 | 89.8 | 65.9 | 89.5 | 64.4 | 90.0 | 39.9 | 71.0 |
| | 0.10 | 89.5 | 63.8 | 89.6 | 50.1 | 90.6 | 74.3 | 90.2 | 48.1 | 89.8 | 35.7 | 90.4 | 77.0 | 89.7 | 71.0 | 89.4 | 65.9 | 90.3 | 50.1 | 74.7 |
| | 0.15 | 89.3 | 65.5 | 89.5 | 53.7 | 90.7 | 82.0 | 90.3 | 57.6 | 89.8 | 47.5 | 90.2 | 87.3 | 89.6 | 75.9 | 89.4 | 67.2 | 90.3 | 60.9 | 78.1 |
| | 0.20 | 89.3 | 67.4 | 89.4 | 56.5 | 90.8 | 89.3 | 90.1 | 68.3 | 89.7 | 55.6 | 90.2 | 92.6 | 89.6 | 79.9 | 89.2 | 68.5 | 90.0 | 70.4 | 80.9 |
| | 0.25 | 89.2 | 69.1 | 89.2 | 59.7 | 90.6 | 93.3 | 89.8 | 78.5 | 89.4 | 60.2 | 89.7 | 95.7 | 89.4 | 83.7 | 89.2 | 69.5 | 89.6 | 78.3 | 83.0 |
| | 0.30 | 89.0 | 70.7 | 88.9 | 62.8 | 90.2 | 95.4 | 89.1 | 85.9 | 89.2 | 66.1 | 89.3 | 97.3 | 88.9 | 86.6 | 89.2 | 70.5 | 89.1 | 84.3 | 84.6 |
| | 0.35 | 88.9 | 72.4 | 88.6 | 65.7 | 90.1 | 96.4 | 88.0 | 90.9 | 88.9 | 69.3 | 88.7 | 98.5 | 88.5 | 89.1 | 89.2 | 71.6 | 88.2 | 88.8 | 85.7 |
| | 0.40 | 88.5 | 73.6 | 88.0 | 68.1 | 89.6 | 97.0 | 86.9 | 93.9 | 88.2 | 72.3 | 87.8 | 99.0 | 87.9 | 91.0 | 89.2 | 72.5 | 87.4 | 92.0 | 86.3 |
| | 0.45 | 88.3 | 74.8 | 87.3 | 70.1 | 89.2 | 97.6 | 85.0 | 95.7 | 87.7 | 73.4 | 86.7 | 99.3 | 87.4 | 92.4 | 89.1 | 73.4 | 86.0 | 94.0 | 86.5 |
| B/32 | 0.50 | 88.1 | 75.8 | 86.8 | 72.3 | 88.6 | 97.9 | 83.0 | 96.8 | 87.2 | 75.1 | 85.2 | 99.4 | 86.9 | 93.3 | 88.8 | 74.2 | 84.4 | 95.2 | 86.6 |
| | 0.55 | 87.6 | 76.4 | 86.4 | 74.2 | 87.7 | 98.1 | 80.4 | 97.5 | 86.5 | 76.9 | 83.3 | 99.5 | 86.2 | 93.7 | 88.5 | 74.7 | 82.3 | 96.1 | 86.5 |
| | 0.60 | 87.2 | 77.3 | 85.7 | 75.3 | 86.9 | 98.2 | 77.2 | 97.8 | 85.8 | 78.1 | 80.8 | 99.6 | 85.8 | 94.3 | 88.2 | 75.2 | 79.9 | 96.5 | 86.1 |
| | 0.65 | 86.7 | 78.0 | 84.8 | 76.2 | 85.7 | 98.4 | 73.3 | 98.1 | 84.9 | 79.2 | 77.8 | 99.6 | 85.1 | 94.7 | 87.9 | 75.6 | 76.9 | 96.7 | 85.5 |
| | 0.70 | 86.2 | 78.4 | 83.9 | 76.8 | 84.5 | 98.5 | 69.3 | 98.2 | 83.9 | 79.5 | 74.5 | 99.6 | 84.0 | 94.9 | 87.5 | 75.9 | 73.7 | 96.9 | 84.8 |
| | 0.75 | 85.3 | 78.9 | 82.8 | 77.7 | 83.0 | 98.6 | 65.3 | 98.4 | 82.5 | 79.5 | 70.6 | 99.6 | 83.0 | 95.1 | 87.2 | 76.1 | 70.1 | 97.0 | 83.9 |
| | 0.80 | 84.2 | 79.1 | 81.6 | 78.3 | 81.0 | 98.5 | 61.0 | 98.6 | 80.7 | 80.2 | 66.1 | 99.6 | 82.0 | 95.1 | 86.6 | 76.1 | 66.2 | 97.1 | 82.9 |
| | 0.85 | 82.8 | 79.6 | 80.3 | 78.6 | 79.2 | 98.5 | 56.2 | 98.6 | 79.0 | 79.2 | 60.5 | 99.6 | 80.3 | 95.0 | 86.1 | 76.0 | 62.2 | 97.1 | 81.6 |
| | 0.90 | 81.4 | 79.5 | 78.6 | 78.7 | 76.9 | 98.5 | 51.9 | 98.6 | 77.0 | 78.9 | 54.6 | 99.6 | 78.9 | 95.0 | 85.6 | 75.8 | 58.4 | 97.2 | 80.3 |
| | 0.95 | 80.0 | 79.4 | 77.0 | 78.9 | 74.1 | 98.5 | 47.3 | 98.5 | 74.7 | 78.2 | 48.9 | 99.6 | 77.3 | 95.1 | 84.9 | 75.4 | 54.5 | 97.2 | 78.9 |
| | 1.00 | 78.2 | 79.2 | 75.0 | 78.7 | 71.5 | 98.6 | 42.6 | 98.5 | 72.1 | 77.9 | 42.9 | 99.6 | 75.2 | 95.0 | 84.2 | 75.1 | 50.4 | 97.2 | 77.3 |
| | 0.00 | 90.8 | 64.7 | 90.8 | 45.0 | 90.8 | 53.9 | 90.8 | 43.4 | 90.8 | 30.5 | 90.8 | 51.3 | 90.8 | 65.8 | 90.8 | 65.5 | 90.8 | 52.0 | 71.6 |
| | 0.05 | 90.8 | 66.9 | 90.7 | 47.4 | 91.4 | 66.8 | 91.1 | 49.0 | 90.8 | 40.5 | 91.2 | 64.4 | 91.0 | 71.2 | 90.6 | 66.9 | 91.3 | 60.6 | 75.1 |
| | 0.10 | 90.8 | 68.9 | 90.5 | 49.7 | 91.6 | 80.0 | 91.3 | 55.3 | 90.7 | 53.9 | 91.6 | 83.0 | 91.0 | 75.9 | 90.5 | 68.4 | 91.5 | 67.8 | 79.0 |
| | 0.15 | 90.7 | 71.1 | 90.4 | 53.0 | 91.9 | 89.5 | 91.3 | 62.8 | 90.6 | 58.5 | 91.8 | 92.1 | 91.0 | 80.0 | 90.2 | 69.7 | 91.6 | 74.5 | 81.7 |
| | 0.20 | 90.8 | 73.3 | 90.2 | 56.5 | 92.0 | 94.1 | 91.0 | 71.1 | 90.6 | 59.8 | 91.8 | 95.8 | 91.2 | 84.1 | 90.0 | 71.0 | 91.7 | 80.4 | 83.6 |
| | 0.25 | 90.7 | 75.4 | 90.0 | 59.5 | 92.0 | 96.1 | 90.6 | 80.5 | 90.5 | 62.2 | 91.5 | 97.6 | 91.0 | 87.4 | 89.7 | 72.3 | 91.6 | 85.6 | 85.2 |
| | 0.30 | 90.6 | 77.2 | 89.9 | 63.6 | 91.8 | 97.1 | 90.1 | 88.0 | 90.6 | 68.4 | 91.5 | 98.4 | 90.9 | 90.3 | 89.5 | 73.4 | 91.2 | 89.8 | 86.8 |
| | 0.35 | 90.5 | 79.1 | 89.8 | 67.0 | 91.7 | 97.6 | 89.7 | 92.7 | 90.5 | 72.3 | 91.0 | 99.0 | 90.7 | 92.1 | 89.3 | 74.5 | 90.6 | 92.9 | 87.8 |
| | 0.40 | 90.3 | 80.9 | 89.6 | 70.4 | 91.4 | 98.0 | 88.9 | 95.5 | 90.4 | 74.0 | 90.6 | 99.3 | 90.6 | 93.5 | 89.2 | 75.4 | 90.2 | 94.9 | 88.5 |
| | 0.45 | 90.0 | 82.0 | 89.5 | 73.5 | 90.8 | 98.3 | 87.8 | 96.9 | 90.3 | 75.9 | 90.1 | 99.4 | 90.5 | 94.5 | 89.0 | 76.2 | 89.6 | 96.1 | 88.9 |
| B/16 | 0.50 | 89.9 | 83.5 | 89.3 | 76.1 | 90.0 | 98.6 | 86.6 | 97.6 | 90.1 | 78.3 | 89.7 | 99.4 | 90.1 | 95.0 | 88.7 | 77.0 | 88.8 | 96.7 | 89.2 |
| | 0.55 | 89.4 | 84.5 | 89.1 | 77.6 | 89.2 | 98.7 | 84.8 | 98.1 | 89.9 | 79.5 | 88.9 | 99.5 | 89.8 | 95.5 | 88.5 | 77.7 | 87.9 | 97.2 | 89.2 |
| | 0.60 | 89.1 | 85.2 | 88.8 | 79.4 | 88.2 | 98.7 | 82.9 | 98.4 | 89.5 | 81.4 | 87.9 | 99.6 | 89.5 | 95.8 | 88.4 | 78.2 | 86.4 | 97.4 | 89.2 |
| | 0.65 | 88.4 | 85.9 | 88.4 | 80.6 | 86.9 | 98.9 | 80.2 | 98.7 | 89.0 | 82.6 | 86.9 | 99.6 | 88.9 | 96.0 | 88.1 | 78.4 | 84.6 | 97.6 | 88.9 |
| | 0.70 | 87.4 | 86.3 | 87.9 | 81.1 | 85.5 | 99.0 | 77.1 | 98.7 | 88.6 | 83.8 | 85.7 | 99.7 | 88.4 | 96.2 | 87.7 | 78.7 | 82.3 | 97.6 | 88.4 |
| | 0.75 | 86.1 | 86.8 | 87.5 | 81.3 | 83.8 | 99.0 | 73.6 | 98.8 | 88.2 | 84.2 | 84.3 | 99.7 | 87.8 | 96.3 | 87.5 | 78.9 | 79.9 | 97.7 | 87.9 |
| | 0.80 | 84.3 | 87.0 | 86.9 | 82.4 | 82.0 | 99.0 | 69.5 | 98.9 | 87.5 | 84.7 | 82.7 | 99.7 | 87.2 | 96.4 | 87.2 | 79.1 | 77.1 | 97.7 | 87.2 |
| | 0.85 | 82.5 | 87.2 | 86.3 | 82.3 | 80.2 | 99.0 | 65.4 | 98.9 | 86.7 | 84.7 | 80.8 | 99.7 | 86.4 | 96.4 | 86.8 | 79.2 | 74.1 | 97.7 | 86.3 |
| | 0.90 | 80.7 | 87.2 | 85.3 | 82.3 | 78.0 | 99.1 | 60.8 | 98.9 | 85.9 | 84.8 | 78.4 | 99.7 | 85.6 | 96.4 | 86.3 | 79.3 | 70.8 | 97.7 | 85.4 |
| | 0.95 | 79.0 | 87.2 | 84.5 | 82.2 | 75.6 | 99.1 | 56.0 | 99.0 | 85.1 | 84.7 | 75.4 | 99.7 | 84.5 | 96.4 | 85.7 | 79.2 | 67.0 | 97.7 | 84.3 |
| | 1.00 | 77.9 | 87.0 | 83.5 | 82.3 | 72.9 | 99.1 | 51.5 | 99.0 | 83.9 | 84.8 | 71.7 | 99.7 | 83.5 | 96.4 | 85.2 | 79.0 | 63.1 | 97.7 | 83.2 |
| | 0.00 | 95.7 | 77.8 | 95.7 | 55.4 | 95.7 | 60.2 | 95.7 | 50.6 | 95.7 | 25.9 | 95.7 | 76.4 | 95.7 | 71.0 | 95.7 | 68.3 | 95.7 | 58.6 | 78.1 |
| | 0.05 | 95.7 | 79.9 | 95.7 | 57.8 | 96.0 | 74.6 | 95.8 | 57.9 | 95.7 | 33.2 | 95.9 | 88.3 | 95.8 | 76.5 | 95.7 | 69.7 | 95.9 | 67.0 | 81.5 |
| | 0.10 | 95.7 | 82.1 | 95.7 | 60.6 | 96.2 | 88.1 | 95.8 | 65.6 | 95.6 | 47.0 | 96.0 | 94.5 | 95.9 | 81.1 | 95.7 | 71.3 | 96.0 | 73.2 | 84.8 |
| | 0.15 | 95.6 | 84.0 | 95.6 | 63.8 | 96.3 | 93.9 | 95.6 | 74.8 | 95.5 | 54.3 | 96.0 | 97.4 | 96.0 | 85.8 | 95.6 | 72.9 | 95.9 | 78.9 | 87.1 |
| | 0.20 | 95.5 | 85.8 | 95.6 | 66.8 | 96.4 | 96.6 | 95.5 | 83.3 | 95.5 | 60.1 | 96.0 | 98.6 | 96.0 | 89.3 | 95.5 | 74.3 | 95.8 | 84.1 | 88.9 |
| | 0.25 | 95.5 | 87.5 | 95.5 | 69.5 | 96.1 | 97.7 | 95.2 | 89.6 | 95.3 | 65.8 | 96.0 | 99.2 | 96.0 | 92.1 | 95.5 | 75.5 | 95.5 | 88.7 | 90.3 |
| | 0.30 | 95.4 | 88.5 | 95.5 | 71.9 | 96.0 | 98.1 | 94.9 | 93.6 | 95.2 | 70.6 | 96.0 | 99.4 | 95.9 | 93.5 | 95.4 | 76.8 | 95.4 | 92.5 | 91.4 |
| | 0.35 | 95.3 | 89.7 | 95.5 | 73.8 | 95.9 | 98.4 | 94.5 | 95.7 | 95.1 | 73.0 | 95.9 | 99.5 | 95.8 | 94.7 | 95.3 | 77.9 | 95.2 | 94.9 | 92.0 |
| | 0.40 | 95.2 | 90.4 | 95.4 | 75.7 | 95.8 | 98.6 | 94.0 | 97.0 | 95.0 | 75.8 | 95.7 | 99.6 | 95.8 | 95.5 | 95.2 | 78.9 | 94.8 | 96.3 | 92.5 |
| | 0.45 | 95.0 | 91.0 | 95.3 | 78.0 | 95.7 | 98.8 | 93.4 | 97.8 | 94.9 | 77.5 | 95.5 | 99.7 | 95.7 | 95.9 | 95.2 | 79.8 | 94.4 | 97.1 | 92.8 |
| L/14 | 0.50 | 94.8 | 91.6 | 95.2 | 79.1 | 95.5 | 98.9 | 92.5 | 98.4 | 94.8 | 80.6 | 95.1 | 99.8 | 95.6 | 96.2 | 95.1 | 80.6 | 94.1 | 97.5 | 93.1 |
| | 0.55 | 94.5 | 92.1 | 95.1 | 80.3 | 95.0 | 98.9 | 91.2 | 98.7 | 94.7 | 82.8 | 94.7 | 99.8 | 95.6 | 96.6 | 95.0 | 81.3 | 93.6 | 97.7 | 93.2 |
| | 0.60 | 94.3 | 92.4 | 95.0 | 81.2 | 94.6 | 99.0 | 89.8 | 98.9 | 94.5 | 84.0 | 94.4 | 99.8 | 95.5 | 96.8 | 95.0 | 81.8 | 93.1 | 97.9 | 93.2 |
| | 0.65 | 94.1 | 92.6 | 94.7 | 82.1 | 94.2 | 99.1 | 88.3 | 99.0 | 94.3 | 84.7 | 93.9 | 99.8 | 95.4 | 96.9 | 95.0 | 82.1 | 92.4 | 97.9 | 93.1 |
| | 0.70 | 94.0 | 92.8 | 94.6 | 82.3 | 93.7 | 99.1 | 86.2 | 99.1 | 94.1 | 85.4 | 93.4 | 99.8 | 95.2 | 97.0 | 94.9 | 82.4 | 91.5 | 98.0 | 93.0 |
| | 0.75 | 93.8 | 92.8 | 94.4 | 82.7 | 93.1 | 99.2 | 84.1 | 99.2 | 93.8 | 85.7 | 92.8 | 99.8 | 94.7 | 97.1 | 94.8 | 82.6 | 90.3 | 98.0 | 92.7 |
| | 0.80 | 93.5 | 92.9 | 94.2 | 82.8 | 92.3 | 99.3 | 81.9 | 99.2 | 93.4 | 85.7 | 92.1 | 99.8 | 94.4 | 97.1 | 94.7 | 82.7 | 89.2 | 98.0 | 92.4 |
| | 0.85 | 93.3 | 92.9 | 94.0 | 83.0 | 91.4 | 99.3 | 79.6 | 99.2 | 93.0 | 85.3 | 91.3 | 99.8 | 94.0 | 97.0 | 94.5 | 82.8 | 88.2 | 98.1 | 92.1 |
| | 0.90 | 93.0 | 92.8 | 93.9 | 83.6 | 90.6 | 99.3 | 77.2 | 99.3 | 92.6 | 86.2 | 90.5 | 99.8 | 93.5 | 97.0 | 94.3 | 82.7 | 86.7 | 98.0 | 91.7 |
| | 0.95 | 92.6 | 92.8 | 93.5 | 83.6 | 89.5 | 99.2 | 74.3 | 99.3 | 92.2 | 86.4 | 89.5 | 99.8 | 93.1 | 97.0 | 94.2 | 82.6 | 85.0 | 98.0 | 91.3 |
| | 1.00 | 92.1 | 92.8 | 93.2 | 83.7 | 87.9 | 99.2 | 71.5 | 99.3 | 91.7 | 86.4 | 88.1 | 99.8 | 92.5 | 96.9 | 94.1 | 82.4 | 83.4 | 98.0 | 90.7 |

Table 7: **Patching on a single new task where CIFAR10 is the supported task.** *S*: accuracy on the supported task (CIFAR10); *P*: accuracy on the patching task; *Avg:* average accuracy on patching and supported tasks.

| | Cars | | DTD | | EuroSAT | | GTSRB | | KITTI | | MNIST | | RESISC45 | | SUN397 | | SVHN | | |
|---|---|---|---|---|---|---|---|---|---|---|---|---|---|---|---|---|---|---|---|
| α | S | P | S | P | S | P | S | P | S | P | S | P | S | P | S | P | S | P | Avg |
| 0.00 | 65.1 | 59.6 | 65.1 | 44.1 | 65.1 | 45.9 | 65.1 | 32.4 | 65.1 | 22.6 | 65.1 | 48.3 | 65.1 | 60.7 | 65.1 | 63.1 | 65.1 | 31.5 | 55.2 |
| 0.05 | 64.6 | 61.9 | 64.9 | 47.0 | 66.5 | 63.4 | 65.8 | 39.7 | 65.1 | 25.5 | 66.1 | 60.9 | 65.2 | 65.9 | 65.1 | 64.4 | 65.7 | 39.9 | 58.8 |
| 0.10 | 64.3 | 63.8 | 64.9 | 50.1 | 67.7 | 74.3 | 66.5 | 48.1 | 65.5 | 35.7 | 67.1 | 77.0 | 65.7 | 71.0 | 64.9 | 65.9 | 66.0 | 50.1 | 62.7 |
| 0.15 | 64.2 | 65.5 | 65.1 | 53.7 | 68.2 | 82.0 | 66.7 | 57.6 | 66.0 | 47.5 | 67.2 | 87.3 | 66.2 | 75.9 | 64.7 | 67.2 | 66.2 | 60.9 | 66.2 |
| 0.20 | 64.2 | 67.4 | 65.2 | 56.5 | 68.3 | 89.3 | 66.1 | 68.3 | 66.5 | 55.6 | 66.9 | 92.6 | 66.8 | 79.9 | 64.8 | 68.5 | 66.0 | 70.4 | 69.1 |
| 0.25 | 64.2 | 69.1 | 65.4 | 59.7 | 68.3 | 93.3 | 65.3 | 78.5 | 66.6 | 60.2 | 66.1 | 95.7 | 66.7 | 83.7 | 64.9 | 69.5 | 65.4 | 78.3 | 71.2 |
| 0.30 | 64.2 | 70.7 | 65.2 | 62.8 | 67.8 | 95.4 | 64.0 | 85.9 | 66.4 | 66.1 | 64.7 | 97.3 | 66.2 | 86.6 | 64.7 | 70.5 | 64.4 | 84.3 | 72.6 |
| 0.35 | 64.2 | 72.4 | 65.2 | 65.7 | 66.7 | 96.4 | 62.3 | 90.9 | 66.3 | 69.3 | 63.0 | 98.5 | 65.9 | 89.1 | 64.7 | 71.6 | 62.7 | 88.8 | 73.5 |
| 0.40 | 64.4 | 73.6 | 64.7 | 68.1 | 65.6 | 97.0 | 60.3 | 93.9 | 65.8 | 72.3 | 61.1 | 99.0 | 65.3 | 91.0 | 64.5 | 72.5 | 60.6 | 92.0 | 74.0 |
| 0.45 | 64.3 | 74.8 | 64.1 | 70.1 | 64.1 | 97.6 | 57.7 | 95.7 | 64.9 | 73.4 | 58.6 | 99.3 | 64.5 | 92.4 | 64.3 | 73.4 | 58.3 | 94.0 | 74.0 |
| B/32 0.50 | 64.2 | 75.8 | 63.1 | 72.3 | 62.5 | 97.9 | 54.6 | 96.8 | 63.8 | 75.1 | 56.1 | 99.4 | 63.8 | 93.3 | 64.3 | 74.2 | 55.4 | 95.2 | 73.8 |
| 0.55 | 63.9 | 76.4 | 61.9 | 74.2 | 60.9 | 98.1 | 50.9 | 97.5 | 62.8 | 76.9 | 53.0 | 99.5 | 62.9 | 93.7 | 63.9 | 74.7 | 52.4 | 96.1 | 73.3 |
| 0.60 | 63.7 | 77.3 | 60.2 | 75.3 | 58.6 | 98.2 | 47.2 | 97.8 | 61.5 | 78.1 | 49.6 | 99.6 | 61.5 | 94.3 | 63.3 | 75.2 | 49.3 | 96.5 | 72.6 |
| 0.65 | 63.2 | 78.0 | 58.7 | 76.2 | 56.3 | 98.4 | 43.4 | 98.1 | 60.2 | 79.2 | 45.9 | 99.6 | 59.9 | 94.7 | 62.7 | 75.6 | 46.2 | 96.7 | 71.8 |
| 0.70 | 62.5 | 78.4 | 56.7 | 76.8 | 53.8 | 98.5 | 39.4 | 98.2 | 58.5 | 79.5 | 42.2 | 99.6 | 58.6 | 94.9 | 62.0 | 75.9 | 43.1 | 96.9 | 70.9 |
| 0.75 | 61.8 | 78.9 | 54.9 | 77.7 | 51.3 | 98.6 | 35.6 | 98.4 | 56.8 | 79.5 | 38.4 | 99.6 | 56.8 | 95.1 | 61.2 | 76.1 | 40.0 | 97.0 | 69.9 |
| 0.80 | 61.0 | 79.1 | 52.9 | 78.3 | 48.5 | 98.5 | 32.0 | 98.6 | 54.9 | 80.2 | 34.1 | 99.6 | 54.9 | 95.1 | 60.4 | 76.1 | 36.6 | 97.1 | 68.8 |
| 0.85 | 60.2 | 79.6 | 50.4 | 78.6 | 45.3 | 98.5 | 28.4 | 98.6 | 53.1 | 79.2 | 29.9 | 99.6 | 52.9 | 95.0 | 59.3 | 76.0 | 33.5 | 97.1 | 67.5 |
| 0.90 | 59.0 | 79.5 | 48.0 | 78.7 | 42.5 | 98.5 | 24.9 | 98.6 | 51.0 | 78.9 | 26.0 | 99.6 | 51.1 | 95.0 | 58.4 | 75.8 | 30.6 | 97.2 | 66.3 |
| 0.95 | 57.9 | 79.4 | 45.4 | 78.9 | 39.8 | 98.5 | 21.9 | 98.5 | 48.6 | 78.2 | 22.2 | 99.6 | 48.9 | 95.1 | 56.9 | 75.4 | 27.4 | 97.2 | 65.0 |
| 1.00 | 56.5 | 79.2 | 42.4 | 78.7 | 36.7 | 98.6 | 18.8 | 98.5 | 45.9 | 77.9 | 18.9 | 99.6 | 46.6 | 95.0 | 55.8 | 75.1 | 24.3 | 97.2 | 63.6 |
| 0.00 | 68.2 | 64.7 | 68.2 | 45.0 | 68.2 | 53.9 | 68.2 | 43.4 | 68.2 | 30.5 | 68.2 | 51.3 | 68.2 | 65.8 | 68.2 | 65.5 | 68.2 | 52.0 | 60.3 |
| 0.05 | 68.3 | 66.9 | 68.3 | 47.4 | 69.5 | 66.8 | 68.9 | 49.0 | 68.3 | 40.5 | 69.0 | 64.4 | 68.5 | 71.2 | 67.9 | 66.9 | 68.8 | 60.6 | 64.0 |
| 0.10 | 68.0 | 68.9 | 68.0 | 49.7 | 70.3 | 80.0 | 69.4 | 55.3 | 68.3 | 53.9 | 69.6 | 83.0 | 68.6 | 75.9 | 67.6 | 68.4 | 69.3 | 67.8 | 67.9 |
| 0.15 | 67.9 | 71.1 | 67.8 | 53.0 | 70.7 | 89.5 | 69.4 | 62.8 | 68.3 | 58.5 | 70.0 | 92.1 | 68.7 | 80.0 | 67.2 | 69.7 | 69.4 | 74.5 | 70.6 |
| 0.20 | 67.8 | 73.3 | 67.7 | 56.5 | 71.0 | 94.1 | 69.2 | 71.1 | 68.3 | 59.8 | 70.1 | 95.8 | 68.9 | 84.1 | 66.9 | 71.0 | 69.3 | 80.4 | 72.5 |
| 0.25 | 67.5 | 75.4 | 67.5 | 59.5 | 70.9 | 96.1 | 68.8 | 80.5 | 68.3 | 62.2 | 69.9 | 97.6 | 69.2 | 87.4 | 66.5 | 72.3 | 68.8 | 85.6 | 74.1 |
| 0.30 | 67.3 | 77.2 | 67.3 | 63.6 | 70.6 | 97.1 | 68.1 | 88.0 | 68.2 | 68.4 | 69.5 | 98.4 | 69.2 | 90.3 | 66.1 | 73.4 | 67.8 | 89.8 | 75.6 |
| 0.35 | 66.9 | 79.1 | 67.1 | 67.0 | 70.1 | 97.6 | 67.0 | 92.7 | 68.2 | 72.3 | 69.0 | 99.0 | 69.4 | 92.1 | 65.9 | 74.5 | 66.7 | 92.9 | 76.5 |
| 0.40 | 66.4 | 80.9 | 66.7 | 70.4 | 69.3 | 98.0 | 65.5 | 95.5 | 68.0 | 74.0 | 68.2 | 99.3 | 69.2 | 93.5 | 65.5 | 75.4 | 65.5 | 94.9 | 77.0 |
| 0.45 | 66.1 | 82.0 | 66.4 | 73.5 | 68.2 | 98.3 | 63.6 | 96.9 | 67.9 | 75.9 | 66.8 | 99.4 | 69.1 | 94.5 | 65.1 | 76.2 | 63.7 | 96.1 | 77.2 |
| B/16 0.50 | 65.5 | 83.5 | 65.8 | 76.1 | 67.0 | 98.6 | 61.2 | 97.6 | 67.8 | 78.3 | 65.1 | 99.4 | 68.6 | 95.0 | 64.6 | 77.0 | 61.6 | 96.7 | 77.2 |
| 0.55 | 64.9 | 84.5 | 65.5 | 77.6 | 65.7 | 98.7 | 57.9 | 98.1 | 67.4 | 79.5 | 63.5 | 99.5 | 68.2 | 95.5 | 64.1 | 77.7 | 59.6 | 97.2 | 76.9 |
| 0.60 | 64.5 | 85.2 | 65.1 | 79.4 | 64.1 | 98.7 | 53.8 | 98.4 | 66.7 | 81.4 | 61.3 | 99.6 | 67.5 | 95.8 | 63.7 | 78.2 | 57.1 | 97.4 | 76.6 |
| 0.65 | 63.8 | 85.9 | 64.3 | 80.6 | 62.4 | 98.9 | 49.9 | 98.7 | 65.9 | 82.6 | 59.1 | 99.6 | 66.7 | 96.0 | 63.0 | 78.4 | 53.9 | 97.6 | 76.0 |
| 0.70 | 63.3 | 86.3 | 63.5 | 81.1 | 60.0 | 99.0 | 45.8 | 98.7 | 65.1 | 83.8 | 56.7 | 99.7 | 65.8 | 96.2 | 62.5 | 78.7 | 50.6 | 97.6 | 75.2 |
| 0.75 | 62.6 | 86.8 | 62.7 | 81.3 | 57.9 | 99.0 | 41.4 | 98.8 | 63.8 | 84.2 | 53.6 | 99.7 | 64.6 | 96.3 | 61.8 | 78.9 | 47.2 | 97.7 | 74.3 |
| 0.80 | 61.8 | 87.0 | 61.8 | 82.4 | 55.3 | 99.0 | 37.2 | 98.9 | 62.9 | 84.7 | 50.3 | 99.7 | 63.3 | 96.4 | 61.2 | 79.1 | 44.0 | 97.7 | 73.5 |
| 0.85 | 60.7 | 87.2 | 60.4 | 82.3 | 53.1 | 99.0 | 33.1 | 98.9 | 61.5 | 84.7 | 46.4 | 99.7 | 62.1 | 96.4 | 60.3 | 79.2 | 40.7 | 97.7 | 72.4 |
| 0.90 | 59.6 | 87.2 | 59.0 | 82.3 | 50.7 | 99.1 | 28.9 | 98.9 | 59.9 | 84.8 | 42.1 | 99.7 | 60.7 | 96.4 | 59.6 | 79.3 | 37.1 | 97.7 | 71.3 |
| 0.95 | 58.5 | 87.2 | 56.9 | 82.2 | 47.8 | 99.1 | 25.1 | 99.0 | 58.5 | 84.7 | 37.7 | 99.7 | 59.4 | 96.4 | 58.8 | 79.2 | 33.1 | 97.7 | 70.1 |
| 1.00 | 57.0 | 87.0 | 55.0 | 82.3 | 44.8 | 99.1 | 22.0 | 99.0 | 56.9 | 84.8 | 32.6 | 99.7 | 57.3 | 96.4 | 57.8 | 79.0 | 29.9 | 97.7 | 68.8 |
| 0.00 | 78.3 | 77.8 | 78.3 | 55.4 | 78.3 | 60.2 | 78.3 | 50.6 | 78.3 | 25.9 | 78.3 | 76.4 | 78.3 | 71.0 | 78.3 | 68.3 | 78.3 | 58.6 | 69.4 |
| 0.05 | 78.0 | 79.9 | 78.2 | 57.8 | 79.3 | 74.6 | 78.5 | 57.9 | 78.3 | 33.2 | 78.9 | 88.3 | 78.5 | 76.5 | 78.1 | 69.7 | 78.9 | 67.0 | 72.9 |
| 0.10 | 77.8 | 82.1 | 78.2 | 60.6 | 79.7 | 88.1 | 78.6 | 65.6 | 78.2 | 47.0 | 79.4 | 94.5 | 78.7 | 81.1 | 78.0 | 71.3 | 79.2 | 73.2 | 76.2 |
| 0.15 | 77.3 | 84.0 | 78.1 | 63.8 | 80.2 | 93.9 | 78.5 | 74.8 | 78.0 | 54.3 | 79.8 | 97.4 | 78.8 | 85.8 | 77.8 | 72.9 | 79.4 | 78.9 | 78.5 |
| 0.20 | 76.9 | 85.8 | 77.9 | 66.8 | 80.5 | 96.6 | 78.3 | 83.3 | 77.9 | 60.1 | 80.0 | 98.6 | 79.0 | 89.3 | 77.4 | 74.3 | 79.5 | 84.1 | 80.3 |
| 0.25 | 76.7 | 87.5 | 77.5 | 69.5 | 80.6 | 97.7 | 78.0 | 89.6 | 77.7 | 65.8 | 80.1 | 99.2 | 79.1 | 92.1 | 77.4 | 75.5 | 79.2 | 88.7 | 81.8 |
| 0.30 | 76.5 | 88.5 | 77.4 | 71.9 | 80.4 | 98.1 | 77.5 | 93.6 | 77.6 | 70.6 | 79.9 | 99.4 | 79.3 | 93.5 | 77.3 | 76.8 | 78.8 | 92.5 | 82.7 |
| 0.35 | 76.1 | 89.7 | 77.3 | 73.8 | 80.2 | 98.4 | 76.8 | 95.7 | 77.4 | 73.0 | 79.8 | 99.5 | 79.3 | 94.7 | 77.1 | 77.9 | 78.3 | 94.9 | 83.3 |
| 0.40 | 75.8 | 90.4 | 77.2 | 75.7 | 79.9 | 98.6 | 75.8 | 97.0 | 77.3 | 75.8 | 79.5 | 99.6 | 79.3 | 95.5 | 76.8 | 78.9 | 77.6 | 96.3 | 83.7 |
| 0.45 | 75.3 | 91.0 | 77.0 | 78.0 | 79.2 | 98.8 | 74.7 | 97.8 | 77.1 | 77.5 | 78.9 | 99.7 | 79.2 | 95.9 | 76.7 | 79.8 | 76.4 | 97.1 | 83.9 |
| L/14 0.50 | 74.9 | 91.6 | 76.7 | 79.1 | 78.5 | 98.9 | 73.2 | 98.4 | 77.1 | 80.6 | 78.2 | 99.8 | 78.9 | 96.2 | 76.5 | 80.6 | 75.1 | 97.5 | 84.0 |
| 0.55 | 74.4 | 92.1 | 76.4 | 80.3 | 77.8 | 98.9 | 71.7 | 98.7 | 77.1 | 82.8 | 77.4 | 99.8 | 78.8 | 96.6 | 76.2 | 81.3 | 73.6 | 97.7 | 84.0 |
| 0.60 | 74.2 | 92.4 | 76.2 | 81.2 | 76.5 | 99.0 | 69.7 | 98.9 | 77.0 | 84.0 | 76.4 | 99.8 | 78.7 | 96.8 | 75.9 | 81.8 | 72.1 | 97.9 | 83.8 |
| 0.65 | 74.1 | 92.6 | 75.9 | 82.1 | 75.4 | 99.1 | 67.3 | 99.0 | 76.9 | 84.7 | 75.5 | 99.8 | 78.4 | 96.9 | 75.6 | 82.1 | 70.4 | 97.9 | 83.5 |
| 0.70 | 73.7 | 92.8 | 75.8 | 82.3 | 73.9 | 99.1 | 64.9 | 99.1 | 76.6 | 85.4 | 74.2 | 99.8 | 78.1 | 97.0 | 75.3 | 82.4 | 68.5 | 98.0 | 83.1 |
| 0.75 | 73.4 | 92.8 | 75.6 | 82.7 | 72.4 | 99.2 | 62.3 | 99.2 | 76.4 | 85.7 | 72.8 | 99.8 | 77.7 | 97.1 | 75.1 | 82.6 | 66.5 | 98.0 | 82.7 |
| 0.80 | 73.2 | 92.9 | 75.1 | 82.8 | 70.7 | 99.3 | 59.5 | 99.2 | 76.2 | 85.7 | 71.5 | 99.8 | 77.2 | 97.1 | 75.0 | 82.7 | 64.6 | 98.0 | 82.3 |
| 0.85 | 73.1 | 92.9 | 74.7 | 83.0 | 68.9 | 99.3 | 56.4 | 99.2 | 75.7 | 85.8 | 69.8 | 99.8 | 76.8 | 97.0 | 74.9 | 82.8 | 62.3 | 98.1 | 81.7 |
| 0.90 | 73.0 | 92.8 | 74.2 | 83.6 | 66.6 | 99.3 | 53.3 | 99.3 | 75.5 | 86.2 | 67.6 | 99.8 | 76.2 | 97.0 | 74.6 | 82.7 | 59.8 | 98.0 | 81.1 |
| 0.95 | 72.6 | 92.8 | 73.5 | 83.6 | 64.3 | 99.2 | 50.0 | 99.3 | 75.2 | 86.4 | 65.4 | 99.8 | 75.1 | 97.0 | 74.3 | 82.6 | 57.4 | 98.0 | 80.4 |
| 1.00 | 72.4 | 92.8 | 72.9 | 83.7 | 62.0 | 99.2 | 46.5 | 99.3 | 74.8 | 86.4 | 63.4 | 99.8 | 74.1 | 96.9 | 73.7 | 82.4 | 54.6 | 98.0 | 79.6 |

Table 8: **Patching on a single new task where CIFAR100 is the supported task.** *S*: accuracy on the supported task (CIFAR100); *P*: accuracy on the patching task; *Avg:* average accuracy on patching and supported tasks.

| | Cars | | DTD | | EuroSAT | | GTSRB | | KITTI | | MNIST | | RESISC45 | | SUN397 | | SVHN | | |
| α | S | P | S | P | S | P | S | P | S | P | S | P | S | P | S | P | S | P | Avg |
|---|---|---|---|---|---|---|---|---|---|---|---|---|---|---|---|---|---|---|---|
| 0.00 | 84.0 | 59.6 | 84.0 | 44.1 | 84.0 | 45.9 | 84.0 | 32.4 | 84.0 | 22.6 | 84.0 | 48.3 | 84.0 | 60.7 | 84.0 | 63.1 | 84.0 | 31.5 | 64.7 |
| 0.05 | 84.0 | 61.9 | 84.0 | 47.0 | 83.9 | 63.4 | 84.0 | 39.7 | 83.9 | 25.5 | 84.0 | 60.9 | 84.0 | 65.9 | 84.0 | 64.4 | 83.9 | 39.9 | 68.0 |
| 0.10 | 83.9 | 63.8 | 83.9 | 50.1 | 83.9 | 74.3 | 83.9 | 48.1 | 83.9 | 35.7 | 83.9 | 77.0 | 83.9 | 71.0 | 84.0 | 65.9 | 83.9 | 50.1 | 71.7 |
| 0.15 | 83.9 | 65.5 | 83.7 | 53.7 | 83.6 | 82.0 | 83.8 | 57.6 | 83.8 | 47.5 | 83.7 | 87.3 | 83.9 | 75.9 | 84.0 | 67.2 | 83.8 | 60.9 | 75.1 |
| 0.20 | 83.9 | 67.4 | 83.5 | 56.5 | 83.3 | 89.3 | 83.6 | 68.3 | 83.7 | 55.6 | 83.3 | 92.6 | 83.7 | 79.9 | 84.0 | 68.5 | 83.6 | 70.4 | 77.8 |
| 0.25 | 83.9 | 69.1 | 83.3 | 59.7 | 82.7 | 93.3 | 83.3 | 78.5 | 83.6 | 60.2 | 82.9 | 95.7 | 83.5 | 83.7 | 84.0 | 69.5 | 83.1 | 78.3 | 79.9 |
| 0.30 | 83.8 | 70.7 | 83.0 | 62.8 | 82.1 | 95.4 | 82.8 | 85.9 | 83.4 | 66.1 | 82.4 | 97.3 | 83.2 | 86.6 | 83.9 | 70.5 | 82.7 | 84.3 | 81.5 |
| 0.35 | 83.7 | 72.4 | 82.5 | 65.7 | 81.3 | 96.4 | 82.1 | 90.9 | 83.1 | 69.3 | 81.8 | 98.5 | 82.8 | 89.1 | 83.7 | 71.6 | 82.1 | 88.8 | 82.6 |
| 0.40 | 83.6 | 73.6 | 82.1 | 68.1 | 80.4 | 97.0 | 81.3 | 93.9 | 83.0 | 72.3 | 81.1 | 99.0 | 82.5 | 91.0 | 83.6 | 72.5 | 81.3 | 92.0 | 83.2 |
| 0.45 | 83.5 | 74.8 | 81.6 | 70.1 | 79.5 | 97.6 | 80.6 | 95.7 | 82.6 | 73.4 | 80.1 | 99.3 | 81.9 | 92.4 | 83.4 | 73.4 | 80.2 | 94.0 | 83.6 |
| B/32 0.50 | 83.3 | 75.8 | 81.0 | 72.3 | 78.5 | 97.9 | 79.5 | 96.8 | 82.2 | 75.1 | 78.9 | 99.4 | 81.4 | 93.3 | 83.1 | 74.2 | 79.2 | 95.2 | 83.7 |
| 0.55 | 83.0 | 76.4 | 80.2 | 74.2 | 77.2 | 98.1 | 78.0 | 97.5 | 81.7 | 76.9 | 77.5 | 99.5 | 80.7 | 93.7 | 82.9 | 74.7 | 77.6 | 96.1 | 83.7 |
| 0.60 | 82.8 | 77.3 | 79.3 | 75.3 | 75.8 | 98.2 | 76.3 | 97.8 | 81.2 | 78.1 | 75.7 | 99.6 | 79.9 | 94.3 | 82.5 | 75.2 | 75.9 | 96.5 | 83.4 |
| 0.65 | 82.4 | 78.0 | 78.4 | 76.2 | 74.2 | 98.4 | 74.4 | 98.1 | 80.6 | 79.2 | 73.2 | 99.6 | 79.0 | 94.7 | 82.0 | 75.6 | 73.9 | 96.7 | 83.0 |
| 0.70 | 82.1 | 78.4 | 77.3 | 76.8 | 72.4 | 98.5 | 71.9 | 98.2 | 79.8 | 79.5 | 70.6 | 99.6 | 77.9 | 94.9 | 81.6 | 75.9 | 71.4 | 96.9 | 82.4 |
| 0.75 | 81.8 | 78.9 | 76.0 | 77.7 | 70.6 | 98.6 | 68.8 | 98.4 | 78.9 | 79.5 | 67.5 | 99.6 | 76.7 | 95.1 | 81.1 | 76.1 | 68.6 | 97.0 | 81.7 |
| 0.80 | 81.3 | 79.1 | 74.7 | 78.3 | 68.4 | 98.5 | 64.7 | 98.6 | 78.0 | 80.2 | 63.7 | 99.6 | 75.3 | 95.1 | 80.4 | 76.1 | 65.2 | 97.1 | 80.8 |
| 0.85 | 80.8 | 79.6 | 73.2 | 78.6 | 66.1 | 98.5 | 59.7 | 98.6 | 76.9 | 79.2 | 59.3 | 99.6 | 73.7 | 95.0 | 79.8 | 76.0 | 61.3 | 97.1 | 79.6 |
| 0.90 | 80.3 | 79.5 | 71.4 | 78.7 | 63.4 | 98.5 | 54.3 | 98.6 | 75.6 | 78.9 | 54.0 | 99.6 | 71.7 | 95.0 | 79.0 | 75.8 | 56.7 | 97.2 | 78.2 |
| 0.95 | 79.7 | 79.4 | 69.2 | 78.9 | 60.3 | 98.5 | 48.2 | 98.5 | 74.2 | 78.2 | 47.9 | 99.6 | 69.5 | 95.1 | 78.0 | 75.4 | 51.6 | 97.2 | 76.6 |
| 1.00 | 79.0 | 79.2 | 67.0 | 78.7 | 57.2 | 98.6 | 41.7 | 98.5 | 72.6 | 77.9 | 41.9 | 99.6 | 67.3 | 95.0 | 77.0 | 75.1 | 46.4 | 97.2 | 75.0 |
| 0.00 | 88.8 | 64.7 | 88.8 | 45.0 | 88.8 | 53.9 | 88.9 | 43.4 | 88.9 | 30.5 | 88.9 | 51.3 | 88.8 | 65.8 | 88.9 | 65.5 | 88.9 | 52.0 | 70.7 |
| 0.05 | 88.8 | 66.9 | 88.8 | 47.4 | 88.7 | 66.8 | 88.8 | 49.0 | 88.8 | 40.5 | 88.8 | 64.4 | 88.8 | 71.2 | 88.9 | 66.9 | 88.9 | 60.6 | 74.1 |
| 0.10 | 88.8 | 68.9 | 88.8 | 49.7 | 88.6 | 80.0 | 88.8 | 55.3 | 88.8 | 53.9 | 88.7 | 83.0 | 88.7 | 75.9 | 88.9 | 68.4 | 88.9 | 67.8 | 77.9 |
| 0.15 | 88.8 | 71.1 | 88.7 | 53.0 | 88.5 | 89.5 | 88.6 | 62.8 | 88.7 | 58.5 | 88.5 | 92.1 | 88.7 | 80.0 | 88.9 | 69.7 | 88.8 | 74.5 | 80.5 |
| 0.20 | 88.8 | 73.3 | 88.6 | 56.5 | 88.2 | 94.1 | 88.5 | 71.1 | 88.6 | 59.8 | 88.3 | 95.8 | 88.5 | 84.1 | 88.9 | 71.0 | 88.7 | 80.4 | 82.4 |
| 0.25 | 88.7 | 75.4 | 88.4 | 59.5 | 88.0 | 96.1 | 88.3 | 80.5 | 88.5 | 62.2 | 88.1 | 97.6 | 88.3 | 87.4 | 88.8 | 72.3 | 88.6 | 85.6 | 84.0 |
| 0.30 | 88.7 | 77.2 | 88.3 | 63.6 | 87.6 | 97.1 | 88.1 | 88.0 | 88.3 | 68.4 | 87.8 | 98.4 | 88.1 | 90.3 | 88.8 | 73.4 | 88.3 | 89.8 | 85.6 |
| 0.35 | 88.5 | 79.1 | 88.0 | 67.0 | 87.3 | 97.6 | 87.8 | 92.7 | 88.1 | 72.3 | 87.6 | 99.0 | 88.0 | 92.1 | 88.7 | 74.5 | 88.1 | 92.9 | 86.6 |
| 0.40 | 88.3 | 80.9 | 87.9 | 70.4 | 86.8 | 98.0 | 87.6 | 95.5 | 88.0 | 74.0 | 87.2 | 99.3 | 87.8 | 93.5 | 88.5 | 75.4 | 87.8 | 94.9 | 87.3 |
| 0.45 | 88.2 | 82.0 | 87.6 | 73.5 | 86.3 | 98.3 | 87.1 | 96.9 | 87.8 | 75.9 | 86.7 | 99.4 | 87.5 | 94.5 | 88.3 | 76.2 | 87.4 | 96.1 | 87.8 |
| B/16 0.50 | 87.9 | 83.5 | 87.3 | 76.1 | 85.8 | 98.6 | 86.6 | 97.6 | 87.6 | 78.3 | 86.1 | 99.4 | 87.3 | 95.0 | 88.1 | 77.0 | 86.9 | 96.7 | 88.1 |
| 0.55 | 87.7 | 84.5 | 87.0 | 77.6 | 85.1 | 98.7 | 86.0 | 98.1 | 87.3 | 79.5 | 85.5 | 99.5 | 86.9 | 95.5 | 88.0 | 77.7 | 86.3 | 97.2 | 88.2 |
| 0.60 | 87.6 | 85.2 | 86.6 | 79.4 | 84.4 | 98.7 | 85.2 | 98.4 | 87.1 | 81.4 | 84.6 | 99.6 | 86.4 | 95.8 | 87.7 | 78.2 | 85.5 | 97.4 | 88.3 |
| 0.65 | 87.3 | 85.9 | 86.1 | 80.6 | 83.4 | 98.9 | 84.1 | 98.7 | 86.8 | 82.6 | 83.7 | 99.6 | 85.8 | 96.0 | 87.4 | 78.4 | 84.6 | 97.6 | 88.2 |
| 0.70 | 87.0 | 86.3 | 85.5 | 81.1 | 82.3 | 99.0 | 82.6 | 98.7 | 86.5 | 83.8 | 82.7 | 99.7 | 85.2 | 96.2 | 87.1 | 78.7 | 83.4 | 97.6 | 88.0 |
| 0.75 | 86.6 | 86.8 | 84.9 | 81.3 | 81.2 | 99.0 | 80.7 | 98.8 | 86.0 | 84.2 | 81.2 | 99.7 | 84.5 | 96.3 | 86.7 | 78.9 | 82.0 | 97.7 | 87.6 |
| 0.80 | 86.3 | 87.0 | 84.1 | 82.4 | 79.8 | 99.0 | 78.1 | 98.9 | 85.5 | 84.7 | 79.5 | 99.7 | 83.6 | 96.4 | 86.3 | 79.1 | 80.5 | 97.7 | 87.1 |
| 0.85 | 85.8 | 87.2 | 83.0 | 82.3 | 78.3 | 99.0 | 75.2 | 98.9 | 84.8 | 84.7 | 77.4 | 99.7 | 82.6 | 96.4 | 85.8 | 79.2 | 78.3 | 97.7 | 86.5 |
| 0.90 | 85.3 | 87.2 | 82.0 | 82.3 | 76.5 | 99.1 | 71.1 | 98.9 | 84.3 | 84.8 | 74.6 | 99.7 | 81.6 | 96.4 | 85.2 | 79.3 | 75.8 | 97.7 | 85.7 |
| 0.95 | 84.9 | 87.2 | 80.7 | 82.2 | 74.5 | 99.1 | 66.8 | 99.0 | 83.5 | 84.7 | 71.4 | 99.7 | 80.5 | 96.4 | 84.5 | 79.2 | 72.9 | 97.7 | 84.7 |
| 1.00 | 84.3 | 87.0 | 79.4 | 82.3 | 72.4 | 99.1 | 61.3 | 99.0 | 82.6 | 84.8 | 67.4 | 99.7 | 79.4 | 96.4 | 83.7 | 79.0 | 69.0 | 97.7 | 83.6 |
| 0.00 | 93.1 | 77.8 | 93.1 | 55.4 | 93.1 | 60.2 | 93.1 | 50.6 | 93.1 | 25.9 | 93.1 | 76.4 | 93.1 | 71.0 | 93.1 | 68.3 | 93.1 | 58.6 | 76.8 |
| 0.05 | 93.1 | 79.9 | 93.1 | 57.8 | 93.1 | 74.6 | 93.0 | 57.9 | 93.0 | 33.2 | 93.0 | 88.3 | 93.1 | 76.5 | 93.1 | 69.7 | 93.1 | 67.0 | 80.1 |
| 0.10 | 93.1 | 82.1 | 93.1 | 60.6 | 93.0 | 88.1 | 93.0 | 65.6 | 93.0 | 47.0 | 93.0 | 94.5 | 93.1 | 81.1 | 93.1 | 71.3 | 93.1 | 73.2 | 83.4 |
| 0.15 | 93.1 | 84.0 | 93.0 | 63.8 | 93.0 | 93.9 | 93.0 | 74.8 | 93.0 | 54.3 | 93.0 | 97.4 | 93.1 | 85.8 | 93.1 | 72.9 | 93.0 | 78.9 | 85.7 |
| 0.20 | 93.1 | 85.8 | 93.0 | 66.8 | 92.9 | 96.6 | 93.0 | 83.3 | 92.9 | 60.1 | 93.0 | 98.6 | 93.0 | 89.3 | 93.1 | 74.3 | 93.0 | 84.1 | 87.5 |
| 0.25 | 93.0 | 87.5 | 93.0 | 69.5 | 92.8 | 97.7 | 92.9 | 89.6 | 92.9 | 65.8 | 92.9 | 99.2 | 93.0 | 92.1 | 93.0 | 75.5 | 92.9 | 88.7 | 89.0 |
| 0.30 | 93.0 | 88.5 | 92.9 | 71.9 | 92.7 | 98.1 | 92.8 | 93.6 | 92.8 | 70.6 | 92.8 | 99.4 | 92.9 | 93.5 | 93.0 | 76.8 | 92.8 | 92.5 | 90.0 |
| 0.35 | 92.9 | 89.7 | 92.8 | 73.8 | 92.6 | 98.4 | 92.8 | 95.7 | 92.8 | 73.0 | 92.7 | 99.5 | 92.9 | 94.7 | 93.1 | 77.9 | 92.7 | 94.9 | 90.7 |
| 0.40 | 92.8 | 90.4 | 92.7 | 75.7 | 92.4 | 98.6 | 92.8 | 97.0 | 92.8 | 75.8 | 92.6 | 99.6 | 92.9 | 95.5 | 93.1 | 78.9 | 92.6 | 96.3 | 91.2 |
| 0.45 | 92.7 | 91.0 | 92.6 | 78.0 | 92.2 | 98.8 | 92.6 | 97.8 | 92.7 | 77.5 | 92.5 | 99.7 | 92.7 | 95.9 | 93.0 | 79.8 | 92.5 | 97.1 | 91.6 |
| L/14 0.50 | 92.7 | 91.6 | 92.5 | 79.1 | 92.0 | 98.9 | 92.6 | 98.4 | 92.6 | 80.6 | 92.4 | 99.8 | 92.6 | 96.2 | 93.0 | 80.6 | 92.4 | 97.5 | 92.0 |
| 0.55 | 92.6 | 92.1 | 92.3 | 80.3 | 91.8 | 98.9 | 92.5 | 98.7 | 92.5 | 82.8 | 92.2 | 99.8 | 92.5 | 96.6 | 92.9 | 81.3 | 92.2 | 97.7 | 92.2 |
| 0.60 | 92.6 | 92.4 | 92.2 | 81.2 | 91.5 | 99.0 | 92.3 | 98.9 | 92.5 | 84.0 | 92.1 | 99.8 | 92.3 | 96.8 | 92.8 | 81.8 | 92.1 | 97.9 | 92.3 |
| 0.65 | 92.5 | 92.6 | 92.0 | 82.1 | 91.2 | 99.1 | 92.2 | 99.0 | 92.4 | 84.7 | 91.9 | 99.8 | 92.1 | 96.9 | 92.7 | 82.1 | 92.0 | 97.9 | 92.4 |
| 0.70 | 92.4 | 92.8 | 91.8 | 82.3 | 90.8 | 99.1 | 92.0 | 99.1 | 92.3 | 85.4 | 91.7 | 99.8 | 92.0 | 97.0 | 92.7 | 82.4 | 91.7 | 98.0 | 92.4 |
| 0.75 | 92.3 | 92.8 | 91.6 | 82.7 | 90.4 | 99.2 | 91.8 | 99.2 | 92.2 | 85.7 | 91.5 | 99.8 | 91.8 | 97.1 | 92.5 | 82.6 | 91.5 | 98.0 | 92.4 |
| 0.80 | 92.1 | 92.9 | 91.4 | 82.8 | 89.9 | 99.3 | 91.5 | 99.2 | 92.0 | 85.7 | 91.2 | 99.8 | 91.5 | 97.1 | 92.5 | 82.7 | 91.2 | 98.0 | 92.3 |
| 0.85 | 91.9 | 92.9 | 91.0 | 83.0 | 89.3 | 99.3 | 91.2 | 99.2 | 91.9 | 85.8 | 90.8 | 99.8 | 91.2 | 97.0 | 92.2 | 82.8 | 90.9 | 98.1 | 92.1 |
| 0.90 | 91.6 | 92.8 | 90.7 | 83.6 | 88.7 | 99.3 | 90.8 | 99.3 | 91.8 | 86.2 | 90.4 | 99.8 | 91.0 | 97.0 | 92.1 | 82.7 | 90.6 | 98.0 | 92.0 |
| 0.95 | 91.5 | 92.8 | 90.3 | 83.6 | 87.9 | 99.2 | 90.4 | 99.3 | 91.7 | 86.4 | 89.9 | 99.8 | 90.6 | 97.0 | 91.8 | 82.6 | 90.2 | 98.0 | 91.8 |
| 1.00 | 91.2 | 92.8 | 89.8 | 83.7 | 86.9 | 99.2 | 89.8 | 99.3 | 91.5 | 86.4 | 89.5 | 99.8 | 90.2 | 96.9 | 91.5 | 82.4 | 89.6 | 98.0 | 91.6 |

Table 9: **Patching on a single new task where Food101 is the supported task.** *S*: accuracy on the supported task (Food101); *P*: accuracy on the patching task; *Avg:* average accuracy on patching and supported tasks.

| | α | Cars S | Cars P | DTD S | DTD P | EuroSAT S | EuroSAT P | GTSRB S | GTSRB P | KITTI S | KITTI P | MNIST S | MNIST P | RESISC45 S | RESISC45 P | SUN397 S | SUN397 P | SVHN S | SVHN P | Avg |
|---|---|---|---|---|---|---|---|---|---|---|---|---|---|---|---|---|---|---|---|---|
| | 0.00 | 97.1 | 59.6 | 97.1 | 44.1 | 97.1 | 45.9 | 97.1 | 32.4 | 97.1 | 22.6 | 97.1 | 48.3 | 97.1 | 60.7 | 97.1 | 63.1 | 97.1 | 31.5 | 71.3 |
| | 0.05 | 97.2 | 61.9 | 97.2 | 47.0 | 97.0 | 63.4 | 97.2 | 39.7 | 97.1 | 25.5 | 97.2 | 60.9 | 97.2 | 65.9 | 97.2 | 64.4 | 97.2 | 39.9 | 74.6 |
| | 0.10 | 97.1 | 63.8 | 97.2 | 50.1 | 97.0 | 74.3 | 97.2 | 48.1 | 97.1 | 35.7 | 97.2 | 77.0 | 97.2 | 71.0 | 97.2 | 65.9 | 97.1 | 50.1 | 78.3 |
| | 0.15 | 97.1 | 65.5 | 97.2 | 53.7 | 96.9 | 82.0 | 97.2 | 57.6 | 97.1 | 47.5 | 97.1 | 87.3 | 97.1 | 75.9 | 97.2 | 67.2 | 97.1 | 60.9 | 81.8 |
| | 0.20 | 97.1 | 67.4 | 97.1 | 56.5 | 96.5 | 89.3 | 97.1 | 68.3 | 97.0 | 55.6 | 96.9 | 92.6 | 97.1 | 79.9 | 97.1 | 68.5 | 97.1 | 70.4 | 84.5 |
| | 0.25 | 97.1 | 69.1 | 97.1 | 59.7 | 96.4 | 93.3 | 97.0 | 78.5 | 97.0 | 60.2 | 96.8 | 95.7 | 97.1 | 83.7 | 97.1 | 69.5 | 96.9 | 78.3 | 86.7 |
| | 0.30 | 97.0 | 70.7 | 97.0 | 62.8 | 96.2 | 95.4 | 96.9 | 85.9 | 96.9 | 66.1 | 96.7 | 97.3 | 97.0 | 86.6 | 97.0 | 70.5 | 96.8 | 84.3 | 88.4 |
| | 0.35 | 97.0 | 72.4 | 97.0 | 65.7 | 95.9 | 96.4 | 96.7 | 90.9 | 96.8 | 69.3 | 96.5 | 98.5 | 97.0 | 89.1 | 97.0 | 71.6 | 96.7 | 88.8 | 89.6 |
| | 0.40 | 97.0 | 73.6 | 96.9 | 68.1 | 95.7 | 97.0 | 96.4 | 93.9 | 96.8 | 72.3 | 96.2 | 99.0 | 96.9 | 91.0 | 96.9 | 72.5 | 96.4 | 92.0 | 90.5 |
| | 0.45 | 96.9 | 74.8 | 96.8 | 70.1 | 95.4 | 97.6 | 96.2 | 95.7 | 96.5 | 73.4 | 96.0 | 99.3 | 96.8 | 92.4 | 96.9 | 73.4 | 96.3 | 94.0 | 91.0 |
| B/32 | 0.50 | 96.6 | 75.8 | 96.8 | 72.3 | 95.0 | 97.9 | 95.8 | 96.8 | 96.3 | 75.1 | 95.5 | 99.4 | 96.6 | 93.3 | 96.8 | 74.2 | 96.1 | 95.2 | 91.4 |
| | 0.55 | 96.3 | 76.4 | 96.6 | 74.2 | 94.5 | 98.1 | 95.3 | 97.5 | 96.2 | 76.9 | 95.2 | 99.5 | 96.4 | 93.7 | 96.7 | 74.7 | 95.7 | 96.1 | 91.7 |
| | 0.60 | 96.0 | 77.3 | 96.4 | 75.3 | 94.1 | 98.2 | 94.6 | 97.8 | 96.0 | 78.1 | 94.7 | 99.6 | 96.2 | 94.3 | 96.7 | 75.2 | 95.3 | 96.5 | 91.8 |
| | 0.65 | 95.8 | 78.0 | 96.2 | 76.2 | 93.6 | 98.4 | 93.8 | 98.1 | 95.7 | 79.2 | 94.2 | 99.6 | 96.0 | 94.7 | 96.5 | 75.6 | 94.5 | 96.7 | 91.8 |
| | 0.70 | 95.4 | 78.4 | 96.2 | 76.8 | 92.7 | 98.5 | 92.7 | 98.2 | 95.3 | 79.5 | 93.4 | 99.6 | 95.8 | 94.9 | 96.4 | 75.9 | 93.6 | 96.9 | 91.7 |
| | 0.75 | 95.0 | 78.9 | 96.0 | 77.7 | 91.7 | 98.6 | 91.5 | 98.4 | 95.0 | 79.5 | 92.6 | 99.6 | 95.3 | 95.1 | 96.2 | 76.1 | 92.8 | 97.0 | 91.5 |
| | 0.80 | 94.5 | 79.1 | 95.8 | 78.3 | 90.8 | 98.5 | 90.1 | 98.6 | 94.8 | 80.2 | 91.6 | 99.6 | 94.9 | 95.1 | 96.2 | 76.1 | 91.7 | 97.1 | 91.3 |
| | 0.85 | 93.6 | 79.6 | 95.4 | 78.6 | 89.9 | 98.5 | 88.3 | 98.6 | 94.3 | 79.2 | 90.4 | 99.6 | 94.5 | 95.0 | 96.0 | 76.0 | 90.5 | 97.1 | 90.8 |
| | 0.90 | 92.6 | 79.5 | 95.0 | 78.7 | 88.6 | 98.5 | 86.2 | 98.6 | 93.6 | 78.9 | 89.1 | 99.6 | 93.9 | 95.0 | 95.8 | 75.8 | 88.9 | 97.2 | 90.3 |
| | 0.95 | 91.4 | 79.4 | 94.5 | 78.9 | 87.3 | 98.5 | 83.6 | 98.5 | 92.9 | 78.2 | 87.3 | 99.6 | 93.1 | 95.1 | 95.6 | 75.4 | 87.0 | 97.2 | 89.6 |
| | 1.00 | 90.2 | 79.2 | 94.2 | 78.7 | 86.2 | 98.6 | 80.4 | 98.5 | 91.9 | 77.9 | 84.6 | 99.6 | 92.5 | 95.0 | 95.3 | 75.1 | 84.8 | 97.2 | 88.9 |
| | 0.00 | 98.2 | 64.7 | 98.2 | 45.0 | 98.2 | 53.9 | 98.2 | 43.4 | 98.2 | 30.5 | 98.2 | 51.3 | 98.2 | 65.8 | 98.2 | 65.5 | 98.2 | 52.0 | 75.4 |
| | 0.05 | 98.3 | 66.9 | 98.3 | 47.4 | 98.2 | 66.8 | 98.2 | 49.0 | 98.3 | 40.5 | 98.2 | 64.4 | 98.3 | 71.2 | 98.2 | 66.9 | 98.3 | 60.6 | 78.8 |
| | 0.10 | 98.2 | 68.9 | 98.3 | 49.7 | 98.2 | 80.0 | 98.3 | 55.3 | 98.2 | 53.9 | 98.2 | 83.0 | 98.3 | 75.9 | 98.2 | 68.4 | 98.2 | 67.8 | 82.6 |
| | 0.15 | 98.2 | 71.1 | 98.3 | 53.0 | 98.2 | 89.5 | 98.2 | 62.8 | 98.2 | 58.5 | 98.1 | 92.1 | 98.2 | 80.0 | 98.3 | 69.7 | 98.2 | 74.5 | 85.3 |
| | 0.20 | 98.2 | 73.3 | 98.2 | 56.5 | 98.1 | 94.1 | 98.1 | 71.1 | 98.2 | 59.8 | 98.1 | 95.8 | 98.2 | 84.1 | 98.3 | 71.0 | 98.2 | 80.4 | 87.2 |
| | 0.25 | 98.1 | 75.4 | 98.2 | 59.5 | 98.1 | 96.1 | 98.1 | 80.5 | 98.2 | 62.2 | 98.0 | 97.6 | 98.2 | 87.4 | 98.2 | 72.3 | 98.1 | 85.6 | 88.9 |
| | 0.30 | 98.1 | 77.2 | 98.2 | 63.6 | 98.0 | 97.1 | 97.9 | 88.0 | 98.1 | 68.4 | 97.8 | 98.4 | 98.2 | 90.3 | 98.1 | 73.4 | 97.9 | 89.8 | 90.5 |
| | 0.35 | 98.0 | 79.1 | 98.2 | 67.0 | 97.9 | 97.6 | 97.8 | 92.7 | 98.0 | 72.3 | 97.7 | 99.0 | 98.1 | 92.1 | 98.1 | 74.5 | 97.7 | 92.9 | 91.6 |
| | 0.40 | 97.9 | 80.9 | 98.1 | 70.4 | 97.8 | 98.0 | 97.6 | 95.5 | 97.9 | 74.0 | 97.5 | 99.3 | 98.1 | 93.5 | 98.1 | 75.4 | 97.6 | 94.9 | 92.4 |
| | 0.45 | 97.8 | 82.0 | 98.0 | 73.5 | 97.7 | 98.3 | 97.4 | 96.9 | 97.9 | 75.9 | 97.3 | 99.4 | 98.0 | 94.5 | 98.0 | 76.2 | 97.5 | 96.1 | 92.9 |
| B/16 | 0.50 | 97.7 | 83.5 | 98.0 | 76.1 | 97.5 | 98.6 | 97.2 | 97.6 | 97.8 | 78.3 | 97.1 | 99.4 | 98.0 | 95.0 | 98.0 | 77.0 | 97.4 | 96.7 | 93.4 |
| | 0.55 | 97.5 | 84.5 | 98.0 | 77.6 | 97.3 | 98.7 | 97.0 | 98.1 | 97.8 | 79.5 | 96.9 | 99.5 | 97.9 | 95.5 | 97.9 | 77.7 | 97.2 | 97.2 | 93.6 |
| | 0.60 | 97.3 | 85.2 | 98.0 | 79.4 | 96.8 | 98.7 | 96.5 | 98.4 | 97.6 | 81.4 | 96.5 | 99.6 | 97.8 | 95.8 | 97.8 | 78.2 | 96.7 | 97.4 | 93.8 |
| | 0.65 | 96.8 | 85.9 | 98.0 | 80.6 | 96.4 | 98.9 | 96.0 | 98.7 | 97.5 | 82.6 | 96.0 | 99.6 | 97.7 | 96.0 | 97.7 | 78.4 | 96.4 | 97.6 | 93.9 |
| | 0.70 | 96.2 | 86.3 | 98.0 | 81.1 | 96.0 | 99.0 | 95.5 | 98.7 | 97.3 | 83.8 | 95.6 | 99.7 | 97.6 | 96.2 | 97.6 | 78.7 | 96.0 | 97.6 | 93.9 |
| | 0.75 | 95.4 | 86.8 | 97.9 | 81.3 | 95.2 | 99.0 | 94.7 | 98.8 | 97.1 | 84.2 | 94.8 | 99.7 | 97.5 | 96.3 | 97.5 | 78.9 | 95.3 | 97.7 | 93.8 |
| | 0.80 | 94.2 | 87.0 | 97.9 | 82.4 | 94.4 | 99.0 | 93.7 | 98.9 | 96.9 | 84.7 | 94.1 | 99.7 | 97.3 | 96.4 | 97.3 | 79.1 | 94.6 | 97.7 | 93.6 |
| | 0.85 | 93.1 | 87.2 | 97.7 | 82.3 | 93.4 | 99.0 | 92.4 | 98.9 | 96.6 | 84.7 | 93.0 | 99.7 | 97.1 | 96.4 | 97.2 | 79.2 | 93.5 | 97.7 | 93.3 |
| | 0.90 | 92.0 | 87.2 | 97.7 | 82.3 | 92.5 | 99.1 | 90.9 | 98.9 | 96.3 | 84.8 | 91.9 | 99.7 | 96.9 | 96.4 | 96.9 | 79.3 | 92.5 | 97.7 | 92.9 |
| | 0.95 | 90.8 | 87.2 | 97.5 | 82.2 | 91.2 | 99.1 | 88.9 | 99.0 | 95.9 | 84.7 | 90.6 | 99.7 | 96.7 | 96.4 | 96.7 | 79.2 | 91.5 | 97.7 | 92.5 |
| | 1.00 | 89.9 | 87.0 | 97.3 | 82.3 | 89.5 | 99.1 | 86.8 | 99.0 | 95.4 | 84.8 | 89.0 | 99.7 | 96.5 | 96.4 | 96.4 | 79.0 | 90.1 | 97.7 | 92.0 |
| | 0.00 | 99.4 | 77.8 | 99.4 | 55.4 | 99.4 | 60.2 | 99.4 | 50.6 | 99.4 | 25.9 | 99.4 | 76.4 | 99.4 | 71.0 | 99.4 | 68.3 | 99.4 | 58.6 | 79.9 |
| | 0.05 | 99.4 | 79.9 | 99.4 | 57.8 | 99.4 | 74.6 | 99.4 | 57.9 | 99.4 | 33.2 | 99.4 | 88.3 | 99.4 | 76.5 | 99.4 | 69.7 | 99.4 | 67.0 | 83.3 |
| | 0.10 | 99.4 | 82.1 | 99.4 | 60.6 | 99.4 | 88.1 | 99.4 | 65.6 | 99.4 | 47.0 | 99.4 | 94.5 | 99.4 | 81.1 | 99.4 | 71.3 | 99.4 | 73.2 | 86.6 |
| | 0.15 | 99.4 | 84.0 | 99.4 | 63.8 | 99.3 | 93.9 | 99.4 | 74.8 | 99.4 | 54.3 | 99.4 | 97.4 | 99.3 | 85.8 | 99.4 | 72.9 | 99.3 | 78.9 | 88.9 |
| | 0.20 | 99.4 | 85.8 | 99.4 | 66.8 | 99.2 | 96.6 | 99.4 | 83.3 | 99.4 | 60.1 | 99.4 | 98.6 | 99.4 | 89.3 | 99.3 | 74.3 | 99.3 | 84.1 | 90.7 |
| | 0.25 | 99.4 | 87.5 | 99.3 | 69.5 | 99.2 | 97.7 | 99.4 | 89.6 | 99.4 | 65.8 | 99.4 | 99.2 | 99.4 | 92.1 | 99.3 | 75.5 | 99.3 | 88.7 | 92.2 |
| | 0.30 | 99.3 | 88.5 | 99.3 | 71.9 | 99.2 | 98.1 | 99.3 | 93.6 | 99.4 | 70.6 | 99.3 | 99.4 | 99.4 | 93.5 | 99.3 | 76.8 | 99.2 | 92.5 | 93.3 |
| | 0.35 | 99.3 | 89.7 | 99.3 | 73.8 | 99.2 | 98.4 | 99.2 | 95.7 | 99.4 | 73.0 | 99.3 | 99.5 | 99.3 | 94.7 | 99.3 | 77.9 | 99.2 | 94.9 | 94.0 |
| | 0.40 | 99.2 | 90.4 | 99.3 | 75.7 | 99.1 | 98.6 | 99.2 | 97.0 | 99.4 | 75.8 | 99.2 | 99.6 | 99.3 | 95.5 | 99.4 | 78.9 | 99.2 | 96.3 | 94.5 |
| | 0.45 | 99.2 | 91.0 | 99.2 | 78.0 | 99.0 | 98.8 | 99.2 | 97.8 | 99.3 | 77.5 | 99.2 | 99.7 | 99.3 | 95.9 | 99.3 | 79.8 | 99.1 | 97.1 | 94.9 |
| L/14 | 0.50 | 99.1 | 91.6 | 99.2 | 79.1 | 99.0 | 98.9 | 99.2 | 98.4 | 99.3 | 80.6 | 99.1 | 99.8 | 99.2 | 96.2 | 99.4 | 80.6 | 99.0 | 97.5 | 95.3 |
| | 0.55 | 99.1 | 92.1 | 99.2 | 80.3 | 98.9 | 98.9 | 99.1 | 98.7 | 99.3 | 82.8 | 99.1 | 99.8 | 99.2 | 96.6 | 99.3 | 81.3 | 98.9 | 97.7 | 95.6 |
| | 0.60 | 99.0 | 92.4 | 99.2 | 81.2 | 98.8 | 99.0 | 99.1 | 98.9 | 99.3 | 84.0 | 99.0 | 99.8 | 99.2 | 96.8 | 99.3 | 81.8 | 98.9 | 97.9 | 95.7 |
| | 0.65 | 98.9 | 92.6 | 99.1 | 82.1 | 98.7 | 99.1 | 99.0 | 99.0 | 99.2 | 84.7 | 99.0 | 99.8 | 99.1 | 96.9 | 99.3 | 82.1 | 98.8 | 97.9 | 95.8 |
| | 0.70 | 98.8 | 92.8 | 99.1 | 82.3 | 98.6 | 99.1 | 99.0 | 99.1 | 99.2 | 85.4 | 98.9 | 99.8 | 99.1 | 97.0 | 99.3 | 82.4 | 98.7 | 98.0 | 95.9 |
| | 0.75 | 98.6 | 92.8 | 99.0 | 82.7 | 98.5 | 99.2 | 98.8 | 99.2 | 99.2 | 85.7 | 98.8 | 99.8 | 99.1 | 97.1 | 99.2 | 82.6 | 98.6 | 98.0 | 95.9 |
| | 0.80 | 98.3 | 92.9 | 99.0 | 82.8 | 98.3 | 99.3 | 98.6 | 99.2 | 99.1 | 85.7 | 98.7 | 99.8 | 99.1 | 97.1 | 99.2 | 82.7 | 98.4 | 98.0 | 95.9 |
| | 0.85 | 98.0 | 92.9 | 99.0 | 83.0 | 98.1 | 99.3 | 98.5 | 99.2 | 99.0 | 85.8 | 98.6 | 99.8 | 99.0 | 97.0 | 99.2 | 82.8 | 98.2 | 98.1 | 95.8 |
| | 0.90 | 97.5 | 92.8 | 98.9 | 83.6 | 97.9 | 99.3 | 98.2 | 99.3 | 99.0 | 86.2 | 98.4 | 99.8 | 99.0 | 97.0 | 99.2 | 82.7 | 97.9 | 98.0 | 95.8 |
| | 0.95 | 96.9 | 92.8 | 98.8 | 83.6 | 97.5 | 99.2 | 97.9 | 99.3 | 98.9 | 86.4 | 98.3 | 99.8 | 98.9 | 97.0 | 99.1 | 82.6 | 97.5 | 98.0 | 95.7 |
| | 1.00 | 95.9 | 92.8 | 98.8 | 83.7 | 97.2 | 99.2 | 97.4 | 99.3 | 98.8 | 86.4 | 98.1 | 99.8 | 98.8 | 96.9 | 99.1 | 82.4 | 97.2 | 98.0 | 95.5 |

Table 10: **Patching on a single new task where STL10 is the supported task.** *S*: accuracy on the supported task (STL10); *P*: accuracy on the patching task; *Avg:* average accuracy on patching and supported tasks.

| Model | Method | ImageNet | Cars | DTD | EuroSAT | GTSRB | KITTI | MNIST | RESISC45 | SUN397 | SVHN | Average |
|-------|--------|----------|------|-----|---------|-------|-------|-------|----------|--------|------|---------|
| | Unpatched | 63.4 | 59.6 | 44.1 | 45.9 | 32.4 | 22.6 | 48.7 | 60.7 | 63.1 | 31.5 | 54.4 |
| | Parallel patching | 60.1 | 63.4 | 50.2 | 75.6 | 58.3 | 42.1 | 90.8 | 71.7 | 64.8 | 66.7 | 62.5 |
| B/32 | Seq. patching (seed 0) | 55.0 | 67.4 | 65.0 | 86.3 | 83.5 | 63.3 | 96.6 | 77.1 | 68.0 | 75.7 | 65.4 |
| | Seq. patching (seed 1) | 55.0 | 60.1 | 60.9 | 86.5 | 82.5 | 58.1 | 98.0 | 85.9 | 67.2 | 87.0 | 65.6 |
| | Seq. patching (seed 2) | 56.3 | 61.1 | 58.8 | 85.4 | 87.1 | 57.0 | 96.9 | 84.7 | 69.5 | 88.5 | 66.4 |
| | Joint patching | 61.5 | 75.2 | 76.0 | 98.0 | 96.9 | 76.2 | 99.2 | 92.6 | 72.6 | 94.2 | 74.1 |
| | Multiple models | 63.4 | 79.2 | 78.7 | 98.6 | 98.5 | 77.9 | 99.6 | 95.0 | 75.1 | 97.2 | 76.1 |
| | Unpatched | 68.3 | 64.6 | 44.9 | 53.9 | 43.3 | 30.5 | 51.6 | 65.8 | 65.5 | 52.0 | 60.4 |
| | Parallel patching | 67.0 | 68.9 | 51.1 | 81.3 | 65.7 | 48.0 | 96.2 | 74.4 | 68.6 | 75.9 | 68.5 |
| B/16 | Seq. patching (seed 0) | 61.6 | 70.2 | 61.8 | 95.0 | 86.4 | 64.1 | 97.8 | 83.9 | 73.4 | 86.2 | 70.7 |
| | Seq. patching (seed 1) | 62.7 | 73.8 | 64.9 | 88.4 | 87.6 | 59.6 | 98.5 | 89.0 | 72.4 | 82.0 | 71.1 |
| | Seq. patching (seed 2) | 64.4 | 70.9 | 59.4 | 83.1 | 91.3 | 64.4 | 97.5 | 87.4 | 70.9 | 87.4 | 71.8 |
| | Joint patching | 67.1 | 82.7 | 79.5 | 98.3 | 97.7 | 80.5 | 99.2 | 94.2 | 75.9 | 95.7 | 78.2 |
| | Multiple models | 68.3 | 87.0 | 82.3 | 99.1 | 99.0 | 84.8 | 99.7 | 96.4 | 79.0 | 97.7 | 80.0 |
| | Unpatched | 75.5 | 77.8 | 55.4 | 60.3 | 50.6 | 26.0 | 76.4 | 71.0 | 68.3 | 58.6 | 68.0 |
| | Parallel patching | 75.1 | 83.1 | 63.7 | 94.0 | 82.1 | 49.6 | 98.4 | 85.8 | 73.4 | 83.4 | 77.2 |
| L/14 | Seq. patching (seed 0) | 73.3 | 85.5 | 75.4 | 95.5 | 91.8 | 71.2 | 98.6 | 90.8 | 80.0 | 93.6 | 80.1 |
| | Seq. patching (seed 1) | 73.1 | 87.7 | 73.6 | 95.3 | 93.5 | 67.4 | 98.7 | 94.3 | 78.8 | 95.1 | 80.1 |
| | Seq. patching (seed 2) | 73.5 | 88.3 | 76.5 | 94.6 | 96.2 | 75.4 | 98.7 | 93.7 | 78.4 | 92.0 | 80.9 |
| | Joint patching | 75.2 | 92.1 | 83.6 | 98.7 | 98.9 | 85.1 | 99.7 | 96.3 | 80.9 | 97.3 | 83.8 |
| | Multiple models | 75.5 | 92.8 | 83.7 | 99.2 | 99.3 | 86.4 | 99.8 | 96.9 | 82.4 | 98.0 | 84.4 |

Table 11: **Contrasting strategies for patching on multiple tasks.** ImageNet is used as the supported task while the other nine datasets are used for patching.