# OpenReview forum: "Patching open-vocabulary models by interpolating weights"
_NeurIPS.cc/2022/Conference — NeurIPS 2022 Accept_

### Official Review · Reviewer_i385 · 2022-07-10

**Rating:** 5
**Confidence:** 3
**Soundness:** 3 good
**Presentation:** 3 good
**Contribution:** 3 good

**Summary:**

This paper proposes a simple yet effective way to do model patching, by interpolating the weights before and after fine-tuning. Experiments are performed with the image encoder from a recent dual-stream vision-language model CLIP. Empirical results show that the proposed method improves performance on other tasks while preserving the performance on ImageNet. Moreover, the proposed method can make CLIP more robust and more powerful for tasks like VQA.

**Questions:**

Please address the questions above.

**Ethics Review Area:**

["I don’t know"]

**Limitations:**

Yes, there is slight discussion about the limitation at the end of the paper.

**Strengths And Weaknesses:**

The paper is overall well structured and easy to follow. The method is simple and effective.

One of my main concerns is about the motivation of "patching models on a single new task". More discussions on the real applications of this scenario would be necessary to better show the motivation of this research. Yet another concern is that this work keeps saying the "open-vocabulary" model. However, it is not clear why the proposed method only works for open-vocabulary models, or whether can be extended to other open-vocabulary models beyond CLIP.

The experiments also need some more clarifications. Though the authors describe some connections and differences with some continual learning methods in the related work section, it would better show the efficacy of the proposed method by empirically comparing with them. In particular, in lines 264-266, the authors said that "in contrast to regularization or replay-based methods, patching requires no extra computational cost during training". Since the proposed method also requires training (i.e., fine-tuning on the data
of the target task), it is not clear to me what the "training" means here. Can the authors provide an exact comparison of the performance VS training time/data between the proposed method and regularization methods like EWC and EWC++?

---

> ### Author Response · Authors · 2022-08-02
> **Response to Reviewer i385 (2/2)**
>
> ### Comparison with EWC
>
> We thank the reviewer for the comment, and clarify the comparison with EWC in terms of the computational requirements. Accuracy comparisons can be found in Appendix E.2: EWC provides high accuracy on the patching and supported tasks for EuroSAT and MNIST, but not for SUN397. Moreover, we can also interpolate between the zero-shot model and the model obtained via EWC, which can lead to further improvements. These results highlight that **patching is complementary with EWC, rather than mutually exclusive alternatives**.
>
> With respect to computational requirements, note that EWC first needs to compute the Fisher information matrix using pre-training data, which is not necessary for our patching procedure. Moreover, during fine-tuning, EWC needs to compute an additional regularization loss, which increases the time required to fine-tune. In contrast, our patching method does not modify the fine-tuning procedure, and thus requires no extra computational cost during fine-tuning.
>
> Concretely, for the experiments shown in Appendix E using a ViT-B/32, computing the Fisher information matrix took 700 seconds, and fine-tuning on MNIST took 748 seconds, a total of 1448 seconds for EWC. Fine-tuning for our patching procedure took 703 seconds, a **2x speed-up compared to EWC, and 6% faster than the fine-tuning portion of EWC**. Note that EWC costs can be alleviated if fewer steps are used for computing the Fisher information matrix, or through multiple experiments, since the Fisher information matrix only needs to be computed once. However, unlike with interpolations where the accuracy trade-off between supported and patching tasks can be navigated without any additional training, EWC requires fine-tuning a new model for every choice of the coefficient for the regularization loss.
>
> EWC++ (Chaudhry et al., 2018) provides efficiency improvements over EWC by keeping a single diagonal Fisher matrix that is updated at every batch. This modification improves runtime when fine-tuning on multiple tasks sequentially, since the Fisher information matrix is updated on the fly during fine-tuning. However, it does not improve runtime when fine-tuning only on a single task---in fact, because the exponential moving average needs to be computed at every batch, runtime may even increase compared to EWC in this setting.
>
> &nbsp;
>
> ### References
> Brown, Tom, et al. "Language models are few-shot learners." Advances in neural information processing systems 33 (2020): 1877-1901.
>
> Ramesh, Aditya, et al. "Hierarchical text-conditional image generation with clip latents." arXiv preprint arXiv:2204.06125 (2022).
>
> Chaudhry, Arslan, et al. "Riemannian walk for incremental learning: Understanding forgetting and intransigence." Proceedings of the European Conference on Computer Vision (ECCV). 2018.

---

> > ### Comment · Reviewer_i385 · 2022-08-08
> > **response**
> >
> > I thank the authors for the detailed reply. The further explanation does let me become more aware of the motivation and scenario of the problem studied, i.e., make models better in certain domains, without making the model worse in other domains. One following-up concern is that in this work, experiment results mostly show that patching helps some tasks while maintaining only imagenet zero-shot performance. It would be more convincing to show performances on more tasks, to further tell if the generalization of a pretrained model is still kept. It would also be interesting if in the future, similar patching ideas can be verified on more pretrained models like GPT-3 as mentioned by the authors. The extra experiments on patching non open-vocabulary models are interesting, and please also consider adding them to the paper.

---

> > > ### Author Response · Authors · 2022-08-08
> > > **Thank you for the response**
> > >
> > > Thank you so much for your response.
> > > We note that, in addition to ImageNet, we explore five other zero-shot datasets (CIFAR-10, CIFAR-100, Food-101 and STL-10) in Section 4.2. We find that our method is stable with respect to the choice of these tasks (Fig. 4).
> > > We agree that exploring similar ideas with large language models is an interesting and timely research direction, and hope that our work fosters further research in this direction.
> > > Once again, we thank you for the valuable suggestion regarding closed-vocabulary models, and we will include the new results in the paper.
> > > We would be very grateful if you considered increasing our score in light of the new experiments and clarifications.
> > > Thanks again for engaging with us to improve our paper.

---

> ### Author Response · Authors · 2022-08-02
> **Response to Reviewer i385 (1/2)**
>
> Thank you for your valuable comments and suggestions, and thank you for engaging with us to to make our work stronger. We address individual questions and concerns below. As mentioned in the comment to all reviewers, we ran new experiments in response to your review. In particular, we extended our method to closed-vocabulary models, and showed that patching is also effective in this setting. We would greatly appreciate it if you took those experiments and the clarifications below into account during discussions, and if there is anything else we can do to further improve our work, please let us know.
>
> &nbsp;
>
> ### Motivation of the paper
> In recent years, it has become increasingly popular to release large-scale models through APIs. Some notable examples are GPT-3 and DALLE-2 from OpenAI (Brown et al., 2020, Ramesh et al., 2022). Other examples include Azure Cognitive Services from Microsoft, Google Cloud AI and AWS AI services.
>
> Many of these models are open-ended, and the APIs can be used for diverse purposes (for instance, using GPT-3 for sentiment analysis, a text classification task, or for text summarization). From the point of view of the organization behind the API, it is desirable that the models they serve cover the widest range possible of cases requested by users. Ideally, this should be achieved using a single model, so that all users have access to the same API and the engineering pipeline required to serve the model is simple. As we discuss in the paper, it is unlikely that after pre-training, the model has high accuracy on all domains, especially for open-ended models---as a practical example, consider how even the largest zero-shot CLIP model has lower accuracy on MNIST than simple logistic regression in pixel space.
>
> Therefore, we desire computationally cheap ways to make models better in certain domains, and we wish these modifications do not make the model worse on other domains, so no user is negatively impacted. This paper tackles this very problem. We also note that these post-hoc updates have been done in practice with both DALL-E 2 (https://openai.com/blog/reducing-bias-and-improving-safety-in-dall-e-2/) and GPT-3 (that now versions their models much like software releases, https://beta.openai.com/docs/models/overview).
>
> &nbsp;
>
> ### Beyond open-vocabulary models
>
> Open-vocabulary models are a natural candidate for patching, since they can perform any classification task without the need to introduce new parameters. For this reason, this class of models is the focus of this work.
>
> Beyond open-vocabulary models, we ran new experiments to investigate whether patching is effective for closed-vocabulary image classifiers. Our experimental setting is as follows: we start with a (closed-vocabulary) model trained on ImageNet from scratch, from the Pytorch ImageNet Models library (https://github.com/rwightman/pytorch-image-models). Our goal is to expand the set of categories known by the model to improve its performance on MNIST, without hurting accuracy on ImageNet. In other words, the goal is to build a model that is competent at classifying an image both amongst the 1000 categories from ImageNet, and amongst 10 digit categories from MNIST. For such, we expand the classification head from the original model by adding 10 new classes, and initialize the corresponding weights and biases to zero. We then fine-tune the model on MNIST without any frozen weights, and interpolate with the model before fine-tuning. Results for patching closed-vocabulary ViT-B/32, ViT-B/16 and ViT-L/16 models that are trained from scratch on ImageNet are shown at https://i.postimg.cc/p2JRtMhN/closed-vocab.png For all models, accuracy on MNISTimproves to over 99%, while accuracy on ImageNet decreases by less than one percentage point. These experiments show that **patching is effective beyond open-vocabulary models**.

---

### Official Review · Reviewer_xjCv · 2022-07-11

**Rating:** 5
**Confidence:** 3
**Soundness:** 3 good
**Presentation:** 2 fair
**Contribution:** 3 good

**Summary:**

In this submission, the authors attempt to improve the CLIP model on the tasks that it performs poorly. They propose to fine-tune pre-trained CLIP model on the target task with frozen classification layer and derive the final model with a linear interpolation between pre-trained and fine-tuned model. The mixing coefficient is decided by validation. This task-patching procedure could be adopted for patching multiple tasks. In the experiments, the authors show that the proposed method retains good performance on the task where the pre-trained model is already good at while improves the performance on the target tasks. In addition to classification tasks, the authors also show that their patching approach could improve CLIP model on 1) typographic attacks, 2) object counting, and 3) visual question answering.

**Questions:**

Would appreciate if authors can address the questions mentioned in the weaknesses.

**Limitations:**

The authors discussed the limitations in the submission.

**Strengths And Weaknesses:**

Strengths
- Improving CLIP model is an active research problem. The authors propose a method to improve the model performance on wider range of tasks for open-vocabulary classification.
- The proposed approach is simple yet effective. To improve CLIP model on poorly performed task, the authors propose to freeze the classifier weights derived fro text encoder and fine-tune the model on given tasks. This preserve the ability of open-vocabulary classification while improve model performance on new tasks.
- The experiments show decent improvement (1% ~ 20%) over the tasks for which the model is fine-tuned while preserving the performance for the task that CLIP model is already good at.
- In addition to classification tasks, the authors show that their patching approach can improve object counting and VQA comparing to the pre-trained CLIP model


Weaknesses
- The writing could improve. The design of proposed method to retain open-vocabulary ability is based on freezing classification layer derived from text encoder. This only mentioned in one sentence in experimental setup. It was pretty confusing to me how the proposed method retain open vocabulary ability until I locate that single statement in experimental setup. I would recommend the authors to make it more clear through out the paper.
- The proposed method is simple. However, in practice, it requires non-trivial effort. To patch the model, the proposed method requires supervised annotations and fine-tuning model with hyper-parameter search and validating the mixing coefficients. The procedure may not be simpler than adding downstream data to the pre-training dataset and then re-train image-text model.
- The authors only compare the performance against pre-trained CLIP model. I am curious how the proposed method compare with a simple baseline: adding downstream data to the CLIP training dataset and re-train image-text model.

---

> ### Author Response · Authors · 2022-08-02
> **Response to Reviewer xjCv**
>
> Thank you for your valuable comments and suggestions. We agree that improving CLIP is an active research problem, and hope our method and empirical results show a solid step towards that goal. Please find our responses to your individual concerns below, and please let us know if there is anything else we can do to further strengthen our submission.
>
> &nbsp;
>
> ### Freezing the classification layer and writing improvements
>
> We kindly thank you for the suggestion, we will add the requested details in more places throughout the paper, including the introduction and when describing our method in Section 2.
>
> We also highlight that the decision to freeze the classification layer does not substantially impact performance, as shown in Appendix C. Moreover, we note that our new experiments with closed-vocabulary models do not freeze the classification layer during fine-tuning, and patching is still effective in that setting (see https://i.postimg.cc/p2JRtMhN/closed-vocab.png for results and the response to reviewer i385 for details).
>
> &nbsp;
>
> ### Patching requires non-trivial effort
>
> We agree with the reviewer that adding downstream data to the pre-training dataset and re-training is simpler. Although it can involve new hyper-parameter choices (like whether the downstream data should be upsampled and by how much), many of the hyper-parameter choices for pre-training are already known. Despite the simplicity, re-training is substantially more expensive than our method. For instance, the ViT-L/14 CLIP model was pre-trained on **256 V100 GPUs for 288 hours**; while Flamingo pre-trained on 1536 TPUv4 chips for 360 hours (Radford et al., 2021, Alayrac et al., 2022). In contrast, patching a ViT-L/14 CLIP model in our experiments takes less than **2 hours on 4 A40 GPUs**, which is around **10,000x** faster than training from scratch, and requires significantly less engineering effort.
>
> Finally, we note that many hyper-parameter configurations work well when weight interpolations are used, which also simplifies our patching process. This was first observed by Wortsman et al., 2022, who showed that many hyper-parameter configurations lead to similar accuracy trade-offs when fine-tuning with a task-specific classification head. This flexibility with respect to hyper-parameter choices reduces the effort required to patch a model, since *no extensive hyper-parameter search is needed*. In our experiments, we found that using the same hyper-parameters works well for all tasks: in all experiments, our method succeeds to improve accuracy on the patching task without substantially harming accuracy on the supported task. Moreover, the evaluations required to choose the mixing coefficient for patching are even cheaper than fine-tuning, since no weight updates are required and validation sets are typically smaller than training sets. Additionally, evaluations with different mixing coefficients can also be run in parallel when multiple GPUs are available.
>
> &nbsp;
>
> ### Adding downstream data to the pre-training set
>
> As suggested, we added data from the MNIST training set to YFCC-15M from Radford et al., and trained a ViT-B/32 from scratch for 32 epochs with a cosine learning rate schedule with linear warmup of 5000 steps and learning rate of 0.001, AdamW optimizer with weight decay 0.1 and batch size 128 using the OpenCLIP codebase (Ilharco et al., 2021). We note that YFCC-15M is 27x smaller than the dataset used by Radford et al. to pre-train their CLIP models, which contains 400M pairs of images and text. We also use the smallest ViT model to make this experiment viable.
>
> We find that **re-training the model with downstream data is an effective way to improve accuracy** on that task. After pre-training, the model achieved over 99% accuracy on MNIST, in contrast to an accuracy of 10% for the model pre-trained without MNIST data (i.e. only YFCC-15M). Re-training with MNIST data yields the same accuracy on MNIST as fine-tuning on MNIST after pre-training on the original dataset. On ImageNet, re-training with MNIST data only slightly decreases accuracy (0.4 percentage points). While effective, re-training is also prohibitively expensive in most practical cases. For instance, retraining a ViT-L/14 with the same dataset used by Radford et al., 2021, would take around **10,000 times more compute** than our patching procedure, which is unrealistic in most settings.
>
> &nbsp;
> ### References:
>
> Alayrac, Jean-Baptiste, et al. "Flamingo: a visual language model for few-shot learning." arXiv preprint arXiv:2204.14198 (2022).
>
> Radford, Alec, et al. "Learning transferable visual models from natural language supervision." International Conference on Machine Learning. PMLR, 2021.
>
> Wortsman, Mitchell, et al. "Model soups: averaging weights of multiple fine-tuned models improves accuracy without increasing inference time." International Conference on Machine Learning. PMLR, 2022
> Ilharco, Gabriel, et al., “OpenCLIP”.

---

> ### Author Response · Authors · 2022-08-08
> **Follow up**
>
> Thanks again for your detailed feedback on our work.
> We are reaching out to check if there are any additional questions or concerns regarding our rebuttal that we can address before the discussion period ends on August 9.
> If there is anything else we can do to further improve our manuscript, please let us know.

---

### Official Review · Reviewer_G7Ms · 2022-07-14

**Rating:** 4
**Confidence:** 4
**Soundness:** 3 good
**Presentation:** 2 fair
**Contribution:** 3 good

**Summary:**

The paper performs an exhaustive empirical study to propose model patching where the goal is to improve accuracy for open-vocabulary models on specific tasks (i.e., patching tasks) without degrading accuracy on  tasks where performance is already adequate. Model patching refers to interpolation of model weights between a finetuned model and the original model. Among other experiments, the paper shows results of model patching on nine tasks where zero-shot CLIP performs poorly, and obtains improvement of over 15-60 percentage points while not losing performance on ImageNet itself. The paper also talks about broad transfer across multiple tasks, and notes that the proposed approach becomes more effective with increasing scale of datasets.

**Questions:**

* What is the summary of takeaways from the paper? The conclusion states “...it is possible to expand the set of tasks on which open-vocabulary models achieve high accuracy”. But a succinct summary of *how* this can be achieved is missing.

* One could argue that interpolating between model weights and finetuned weights is like parameter regularization techniques in continual learning, which impose a certain constraint on how far the weights can be from the original weights. This comparison is glaringly lacking in the main paper. While the Appendix presents a result with Elastic Weight Consolidation, it mentions that one needs access to the pre-training data for a complete experiment. However, there are many continual learning methods that do not need replay buffers (e.g. Gradient Projection Memory for Continual Learning, ICLR 2021 – this is only a representational example), and it may have been useful to see how such a method compares against them.

* Considering the recent focus on OOD generalization performance and robustness to corruption, how would this method work on OOD corruption benchmarks? It may be interesting to see the performance.

I will be happy to revisit the rating after the discussion with/response from authors.

**Limitations:**

Please see the weaknesses listed above.

**Strengths And Weaknesses:**

Strengths:
—-----------
+ The methodology tested and proposed is very simple, and easy-to-implement.
+ The results are interesting, and of broad relevance to the community, especially to those in large-scale ML practice.
+ The results are comprehensive, and cover experiments across various settings, including ones such as typographic attacks, counting and visual question-answering.
+ The Appendix is loaded with even more results, which make this an elaborate empirical effort.

Weaknesses:
—---------------
- The primary weakness of the paper is the limitation mentioned in the paper itself (L315-316): “....our method provides no guarantees on which data the model performance might change…”. This limits the robustness of the takeaways from the paper, and where one may be able to use them – especially considering the contributions are largely empirical.

- Considering \alpha is the key hyperparameter for the interpolation, it would have been nice to see how the performance changes with different \alphas. This, to me, is another weakness of the paper - amid the large number of results, it lacks a clearer perspective on how a reader can take away lessons that can be used in practice (especially considering practice is the focus of this work).

- In continuation to the above comment, there is no evident trend in the results (or a summary of it in the paper) to see when this method works best.

- Considering the large number of results presented in the work (including the Appendix), the paper definitely needs a summary or discussion summarizing the lessons and takeaways for the work to be useful to the readers. It becomes the reader’s burden otherwise to sift and find the takeaways.

- The paper doesn’t compare with other papers that linearly interpolate neural network weights as mentioned in the baselines section.

---

> ### Author Response · Authors · 2022-08-02
> **Response to Reviewer G7Ms (4/4)**
>
> ### References
>
>
> Kornblith, Simon, et al. "Similarity of neural network representations revisited." International Conference on Machine Learning. PMLR, 2019.
>
> Ramasesh, Vinay Venkatesh, Aitor Lewkowycz, and Ethan Dyer. "Effect of scale on catastrophic forgetting in neural networks." International Conference on Learning Representations. 2021.
>
> Kirkpatrick, James, et al. "Overcoming catastrophic forgetting in neural networks." Proceedings of the national academy of sciences 114.13 (2017): 3521-3526.
>
> Wortsman, Mitchell, et al. "Robust fine-tuning of zero-shot models." Proceedings of the IEEE/CVF Conference on Computer Vision and Pattern Recognition. 2022.
>
> Li, Zhizhong, and Derek Hoiem. "Learning without forgetting." IEEE transactions on pattern analysis and machine intelligence 40.12 (2017): 2935-2947.
>
> Lubana, Ekdeep Singh, et al. "How do Quadratic Regularizers Prevent Catastrophic Forgetting: The Role of Interpolation." arXiv preprint arXiv:2102.02805 (2021).
>
> Gobinda Saha et al. “Gradient Projection Memory for Continual Learning.” International Conference on Learning Representations. 2021.
>
> Recht, Benjamin, et al. "Do imagenet classifiers generalize to imagenet?." International Conference on Machine Learning. PMLR, 2019.
>
> Hendrycks, Dan, et al. "The many faces of robustness: A critical analysis of out-of-distribution generalization." Proceedings of the IEEE/CVF International Conference on Computer Vision. 2021.
>
> Wang, Haohan, et al. "Learning robust global representations by penalizing local predictive power." Advances in Neural Information Processing Systems 32 (2019).
>
> Barbu, Andrei, et al. "Objectnet: A large-scale bias-controlled dataset for pushing the limits of object recognition models." Advances in Neural Information Processing Systems 32 (2019).
>
> Hendrycks, Dan, and Thomas Dietterich. "Benchmarking neural network robustness to common corruptions and perturbations." arXiv preprint arXiv:1903.12261 (2019).
>
> Szegedy, Christian, et al. "Rethinking the inception architecture for computer vision." Proceedings of the IEEE conference on computer vision and pattern recognition. 2016.
>
> Taori, Rohan, et al. "Measuring robustness to natural distribution shifts in image classification." Advances in Neural Information Processing Systems 33 (2020): 18583-18599.

---

> > ### Comment · Reviewer_G7Ms · 2022-08-09
> > **Response to rebuttal**
> >
> > Thank you for your detailed responses to each of the concerns, and the efforts for all the additional results.
> >
> > To me, the most useful part of the response was the summary of trends, and the response to the question on "On which data does the method work best?" Considering the significant number of results across the paper (which is also a strength of the paper), it may have been nice to pivot the full paper on such take-aways, organize the paper in terms of them and show empirical proof for the take-aways. This may be more useful for a reader. This still remains a bit of a concern for me however, given the large number of results across the paper. I do not have further questions for the authors at this time, and will consider these responses in discussing with other reviewers and the AC in the finalization stage.

---

> > > ### Author Response · Authors · 2022-08-09
> > > **Thank you for the response**
> > >
> > > Thank you for engaging with us to improve our work. We offer to revisit the paper organization if the paper is accepted, and thank you again for considering our responses.

---

> ### Author Response · Authors · 2022-08-02
> **Response to Reviewer G7Ms  (3/4)**
>
> ### OOD generalization
> We investigated how our method performs on several OOD benchmarks. First, we studied natural distribution shifts (Taori et a., 2020) from ImageNet: ImageNet-V2 (Recht et al., 2019), ImageNet-R (Hendrycks et al., 2021), ImageNet-Sketch (Wang et al., 2019), and ObjectNet (Barbu et al., 2019). We use ImageNet as our patching task, CIFAR100 as the supported task and the same hyper-parameters described in Section 3 for fine-tuning.
>
> Results for natural distribution shifts can be found in https://i.postimg.cc/1y6yJqy3/ood-evals.png. We find that **patching consistently improves accuracy on natural distribution shifts** and on ImageNet, compared to the zero-shot model. Moreover, patching consistently improves the performance on the OOD benchmarks compared to the fine-tuned model, with the exception of ImageNet-V2, which was collected to closely reproduce ImageNet. These results are aligned with those of Wortsman et al., 2021, who fine-tuned models with trainable, task-specific classification heads. The table below shows the accuracy before and after patching for our experiments, using a mixing coefficient of 0.5.
>
> |  | ImageNet | ImageNet-V2 | ImageNet-R | ImageNet-Sketch | ObjectNet | CIFAR100 |
> | -------------------------- | :---------: | :---------: | :--------: | :-------------: | :-------: | :------: |
> | ViT-B/32 before patching   | 0.634  | 0.557       | 0.693      | 0.424           | 0.434     | 0.651    |
> | ViT-B/32 after fine-tuning | **0.765** | **0.656** | 0.632 | 0.431 | 0.424 | 0.578 |
> | ViT-B/32 after patching  |   0.754 | 0.654 | **0.712** | **0.470** | **0.462** | **0.660** |
> ||
> | ViT-B/16 before patching   |  0.683 | 0.619 | 0.776 | 0.483 | 0.537 | 0.682 |
> | ViT-B/16 after fine-tuning  | **0.816** | **0.725** | 0.719 | 0.498 | 0.525 | 0.620 |
> | ViT-B/16 after patching   |  0.804 | 0.716 | **0.792** | **0.537** | **0.575** | **0.699** |
> | |
> | ViT-L/14 before patching |   0.755 | 0.697 | 0.878 | 0.595 | 0.666 | 0.783 |
> | ViT-L/14 after fine-tuning  | **0.855** | 0.774 | 0.843 | 0.598 | 0.645 | 0.713 |
> | ViT-L/14 after patching   |  0.852 | **0.778** | **0.891** | **0.638** | **0.696** | **0.785** |
> ||
>
> Moreover, we study how our method performs on synthetic distribution shifts, where images are artificially corrupted. Specifically, we explore corruptions from ImageNet-C (Hendrycks et al., 2019). Similarly to the experiments with natural distribution shifts, we use ImageNet as the patching task and CIFAR100 as the supported task. We further evaluate models on seven corruption types from ImageNet-C, “gaussian noise”, “impulse noise”, “shot noise”, “defocus blur”, “glass blur”, “motion blur” and “zoom blur”, averaging results over all corruptions severities (1,...,5).
>
> Results can be found in https://i.postimg.cc/KZ2xV8Yr/ood-evals-corruptions.png. We find that **patching consistently improves accuracy on synthetic out-of-distribution benchmarks** and on ImageNet compared to the zero-shot model. Moreover, patching consistently improves the performance on these OOD benchmarks compared to the fine-tuned model. The table below shows the accuracy before and after patching, for a mixing coefficient of 0.5, averaging over the “blur” and “noise” categories.
>
> | | ImageNet | ImageNet-C Blur | ImageNet-C Noise | CIFAR100 |
> | -------------------------- | :---------: | :---------: | :--------: | :-------------: |
> | ViT-B/32 before patching |   0.634 |   0.357   |    0.354   |   0.651 |
> | ViT-B/32 after fine-tuning  | **0.765** |   0.429   |    0.444   |   0.578 |
> | ViT-B/32 after patching    |  0.754 |   **0.448**   |    **0.460**   |   **0.660** |
> ||
> | ViT-B/16 before patching   |  0.683 |   0.392   |    0.374   |   0.682 |
> | ViT-B/16 after fine-tuning  | **0.816** |   0.469   |    0.429   |   0.620 |
> | ViT-B/16 after patching    |  0.804 |   **0.489**   |   **0.475**   |   **0.699** |
> ||
> | ViT-L/14 before patching   |  0.755 |   0.524   |    0.517   |   0.783 |
> | ViT-L/14 after fine-tuning  | **0.855** |   0.578   |    0.569   |   0.713 |
> | ViT-L/14 after patching    |  0.852 |   **0.617**   |   **0.616**   |   **0.785** |
> ||

---

> ### Author Response · Authors · 2022-08-02
> **Response to Reviewer G7Ms (2/4)**
>
>
> ### How performance changes with different $\alpha$s
>
> We explore how accuracy changes by varying the mixing coefficient in several experiments. Some examples are Tables 6-10 in the Appendix. We provide an additional plot showing how accuracy on the patching and supported tasks vary *as a function of the mixing coefficient* at https://i.postimg.cc/92Kr543B/alphas-normalized.png. To account for the difference in difficulty of the tasks, we normalize accuracies by dividing them by the maximum accuracy of all weight interpolations for a given model and task. As expected, accuracy on the patching task increases for larger mixing coefficients, accuracy on the supported task decreases for larger mixing coefficients. The average accuracy peaks for intermediary values of the mixing coefficient, and becomes more stable with larger models.
>
> In Section 2, we propose a simple criterion for choosing the mixing coefficient that can be used by practitioners: choose the $\alpha$ that maximizes the average validation accuracy on the patching and supported tasks. Moreover, we believe that it is a strength of our method that, in addition to the above criterion, practitioners can efficiently control the accuracy trade-off between patching and supported tasks, since using different mixing coefficients do not require new models to be trained. This is in contrast with traditional continual learning methods like EWC or LwF (Kirkpatrick et al., 2017; Li et al., 2017), which require multiple training runs to obtain models with different accuracy trade-offs.
>
> &nbsp;
>
> ### Comparisons with other papers that interpolate between networks
> We compare our method with Exponential Moving averages (EMA, Szegedy et al., 2016) in Appendix E.1. As shown in Figure 9, patching achieves a similar trade-off curve as EMA. In contrast with EMA, we can get a continuous range of solutions betweens the patching and fine-tuned models. Moreover, our method can be applied to existing fine-tuned checkpoints, since it does not modify the fine-tuning procedure.
>
> Additionally, our investigation differs from that of Wortsman et al., 2022 in that we keep the model open-vocabulary when fine-tuning. By contrast, their method WiSE-FT fine-tunes models with a task-specific classification head. In WiSE-FT, the fine-tuned model is not evaluated on tasks which have different classes than the fine-tuning task, which is an important aspect of our experiments.
>
> &nbsp;
>
> ### Comparison with parameter regularization techniques and other CL methods
> In Figure 3, we show comparisons with regularizing the model towards initialization, a popular technique for preventing catastrophic forgetting, along with other baselines. A zoomed-in version of this figure can be found in https://i.postimg.cc/BbXt12q0/baselines.png. Our method matches or surpasses the baselines we investigate. In contrast to regularization towards initialization, our method can be applied to any existing fine-tuned checkpoint, since it does not modify or add new hyper-parameters at fine-tuning time. This also makes the process of exploring the accuracy trade-off between patching and supported tasks easier and more efficient, since it is not necessary to fine-tune multiple models.
>
> Moreover, we have added a new comparison against another continual learning method, Learning without Forgetting (LwF, Li et al., 2017). We fine-tune with various distillation coefficients, (0.1, 0.2, …, 0.9), using the hyper-parameters described in Section 3. Following comparisons on Appendix E.2, we use a ViT-B/16 model and fine-tune on SUN-397, EuroSAT and MNIST. We further interpolate between the weights of the model obtained using LwF and the weights of the zero-shot model. We show results in https://i.postimg.cc/q4NJ6FYn/lwf.png, finding that **our method achieves comparable or better accuracy than LwF on all three datasets**.
>
> We chose to investigate the additional LwF baseline instead of the Gradient Projection Memory (GPM) method from Saha et al., 2021 mentioned as a representative example. Our concern was that nontrivial engineering time may be required to get GPM working at the scale we consider. Our experiments primarily consider CLIP ViT networks, the smallest of which (ViT-B/32, 87M parameters) is much larger than the largest network in GPM (ResNet18, 11M parameters). Moreover, while GPM analyzes fully connected and convolutional layers, the ViT networks we use in most experiments use self-attention.
>
> Finally, as noted by the reviewer, we also compare against EWC in Appendix E.2. We view all of these techniques as **complementary with our method, rather than mutually exclusive alternatives**. This is because the models obtained via these techniques can also be interpolated with the zero-shot model, which can lead to further improvements. This is illustrated both in Appendix E.2 and in our plots showing LwF results.

---

> ### Author Response · Authors · 2022-08-02
> **Response to Reviewer G7Ms (1/4)**
>
> Thank you for your very useful suggestions, and for engaging with us to improve our work. As mentioned in the comment to all reviewers, we ran several new experiments in response to your constructive comments. We would greatly appreciate it if you took those experiments and the clarifications below into account. Please let us know if there is anything else we can do to further improve our paper.
>
> Since it was suggested that we include a summary of the main takeaways and trends in our work, we will begin this response by addressing these points in order to clarify our contributions.
>
> &nbsp;
>
> ### Summary of the lessons in this work
>
> We highlight the following lessons from our work:
> - Even the best pre-trained models are not perfect. We show that it is possible to “patch” models, improving their accuracy on tasks where their performance is not satisfactory, without harming accuracy elsewhere.
> - Our method for how to achieve this has two steps: first, fine-tune on one or more tasks where we wish to improve performance; then, linearly interpolate between the weights of the model before fine-tuning and the weights of the model after fine-tuning. The mixing coefficient can be chosen based on the average accuracy across the tasks of interest, or other criteria practitioners might choose.
> - Patching incurs no extra computational cost compared to standard fine-tuning, neither during fine-tuning itself or at inference time.
> - Patching provides improvements on a wide range of tasks, ranging from image classification to visual question answering. Our method can be applied either to a single task or to multiple tasks. Overall, patching leads to a single model that has competitive performance compared to using multiple specialized models.
>
> We also highlight the following **trends** from our experimental results:
>
> - Patching works best for larger models (Section 4.1).
> - Patching is stable across supported tasks (Section 4.2).
> - Patching is more effective for ViT models than it is for ResNets (Section 4.2, Appendix G).
> - When patching on multiple tasks, we show clear accuracy trends across the different strategies: joint patching outperforms other strategies, followed by sequential patching, and finally parallel patching (Section 5). All methods improve with scale.
>
> &nbsp;
>
> ### On which data does the method work best?
>
> Recall that we distinguish between *patching tasks*, where accuracy is low and we wish to improve performance with additional training, and *supported tasks*, where accuracy is already satisfactory, and we wish to maintain performance.
> Our method is stable with respect to different choices of supported tasks, as shown in Figure 4 in our paper, and further elaborated in Appendix F. As shown in Figure 4 (right), the mixing coefficient that provides the optimal performance is stable across different supported tasks. Choosing the optimal mixing coefficient based on a different supporting task results in negligible accuracy drops compared to using a fixed supporting task.
>
> Moreover, our method is also stable with respect to the choice of patching tasks. As seen in Figure 3 (left), the accuracy distance to optimal for a given model varies by at most two percentage points between different patching tasks. We further investigated for which dataset patching is more effective, as measured by accuracy distance to optimal (see Section 4.1). As shown in https://i.postimg.cc/wHfxvzWg/corr-centered-kernel-alignment.png, we found that the representational similarity (CKA, Kornblith et al., 2019) between the zero-shot and fine-tuned models is strongly correlated with accuracy distance to optimal (Spearman rank correlation 0.72, p < 0.01). Based on this finding, we speculate that patching is more effective when the patching task is closer to the pre-training task, so the representations from the model change less during fine-tuning.

---

> ### Author Response · Authors · 2022-08-08
> **Follow up**
>
> Thank you again for taking the time to provide valuable comments about our work.
> As we are approaching the end of the reviewer-author discussion period on August 9, we are reaching out to check if there are any additional questions or concerns about our rebuttal.
> Please let us know if there is anything we can do to further improve our paper, and thanks again for your comments so far.

---

### Author Response · Authors · 2022-08-02
**General response**

We kindly thank the reviewers for their time and thoughtful feedback. We are grateful for reviewers pointing out that our experiments are “comprehensive” and form an “elaborate empirical effort” (G7Ms), that our methodology is simple, easy to implement, and effective (G7Ms, xjCv, i385), that our paper is “well structured and easy to follow” (i385), and that our results are “interesting, and of broad relevance to the community” (G7Ms).

In response to reviewers, we **ran several new experiments**, which we hope make our paper stronger. In particular, we thank the reviewers for the following suggestions:


- We compare our method with additional regularization-based continual learning baselines, as suggested by reviewer G7Ms. We find that **our method is competitive or outperforms Learning without Forgetting** (LwF, Li et al., 2017), and that linear interpolations between the model learned with LwF and the zero-shot model provide further improvements.
- We ran several new evaluations to understand how patching affects performance out-of-distribution, as suggested by reviewer G7Ms. More specifically, we evaluate how patching on ImageNet affects performance on natural distribution shifts (ImageNet-V2, ImageNet-R, ImageNet-Sketch, ObjectNet) and synthetic distributions shifts, exploring corruptions from ImageNet-C. Our findings show that **patching also improves accuracy under distribution shift**, in line with earlier work on model interpolation.
- Beyond open-vocabulary models, we explore patching closed-vocabulary (e.g., standard ImageNet pretrained models) as suggested by reviewer i385. To do so, we expand the classification head of a trained closed-vocabulary model to incorporate new classes, fine-tune on a new dataset, and interpolate with the model before fine-tuning. For multiple closed-vocabulary models trained on ImageNet, patching on MNIST improves accuracy on that task without substantially degrading accuracy on ImageNet, showing that **patching is also effective for closed-vocabulary models**.
- Finally, as suggested by reviewer xjCv, we train a CLIP model from scratch, adding data from MNIST at pre-training time. To make this experiment feasible, we use YFCC-15M as our main data source for pre-training, which is 27x smaller than the dataset used by Radford et al., 2021. We find that retraining is highly effective, improving MNIST accuracy to over 99%, with negligible difference in zero-shot accuracy on ImageNet compared to the model trained from scratch without MNIST data. Despite its effectiveness, re-training is prohibitively expensive in most practical scenarios. For instance, considering the ViT-L/14 model trained by Radford et al., **training requires around 10,000 times more compute than our patching procedure**.

We hope these new experiments, in addition to the clarifications detailed in the individual responses, address the concerns raised by the reviewers. We would greatly appreciate it if the reviewers would consider taking the clarifications and experiments into consideration during discussions.

---

> ### Author Response · Authors · 2022-08-08
> **Follow up to general response**
>
> Hello Reviewers,
>
> We’d like to reach out again to check if there were any additional questions or concerns about our rebuttal that we can address before the reviewer-author discussion period ends on August 9. Thanks again for taking the time to read our work and provide helpful feedback!
>
> Paper Authors

---

### Meta-Review · Area_Chair_YTE2 · 2022-08-30

**Recommendation:** Accept
**Confidence:** Less certain

**Metareview:**

The reviewers had some concerns about clarity of motivation and baselines.  My own opinion is that this work is valuable for the community because of the simplicity of the method and depth of experiments.

**Award:**

No

---

### Decision · Program_Chairs · 2022-09-14

Accept